# Optimal Top-$k$ Identification from Pairwise Comparisons

**Motti Goldberger** [1]   **Nils Rudi** [1]

## Abstract

We study the active learning problem of fixed-confidence top-$k$ identification from noisy pairwise comparisons. In this problem, an algorithm sequentially chooses pairs of items to compare, observes the outcomes, and stops when it can return the set of top-$k$ items with error probability at most $\delta$. The objective is to design such a $\delta$-*correct* procedure that minimizes the expected number of comparisons (the sample complexity). This problem falls within the broader literature on fixed-confidence pure exploration in bandit models, where a common target is asymptotic optimality: the algorithm's expected sample complexity matches the information theoretic lower bound as $\delta \to 0$. Asymptotically optimal procedures have been developed for a range of fixed-confidence pure-exploration problems, however to the best of our knowledge, for top-1, or more generally top-$k$ identification from pairwise comparisons under latent utility models an asymptotically optimal algorithm has not been established. In this setting, we develop such an algorithm. We characterize the structure of the lower bound and formulate it as a saddle-point problem. This structure enables a computationally efficient primal–dual procedure that learns the asymptotically optimal comparison allocation online. We then construct an adaptive comparison-allocation algorithm that tracks the allocation learned by the primal–dual procedure and prove it is asymptotically optimal.

## 1. Introduction

The objective of identifying the top-$k$ items from a set of candidates using noisy pairwise comparisons arises in many settings. Consider a practitioner who wants to determine which large language model is best for their task. They open

Arena (Chiang et al., 2024) (formerly Chatbot Arena), a public platform for evaluating LLMs through crowdsourced pairwise human preferences, and scan the leaderboard. The leaderboard ranks all relevant models, but the practitioner only wants to consider the top few on the list.

Additional settings that use pairwise comparisons for top-$k$ identification include: crowdsourced pairwise comparisons to identify the best photographs, translations, or annotations (Narimanzadeh et al., 2023; Kou et al., 2017; Chen et al., 2013); recommender systems that elicit pairwise feedback to identify a small set of items to display to a user (Kalloori et al., 2018); and sports tournaments.

Top-$k$ identification serves as the natural objective in two distinct settings. In the first setting, the top-$k$ set is the desired output itself. For example, the single best LLM is selected ($k = 1$), a funding program supports the top five proposals, or a streaming service displays a top ten list. In the second setting, top-$k$ identification is the first stage of a two-stage process, with the second stage using a different and/or more expensive signal. For example, Arena could be used to narrow the field of candidate models to $k$, after which a more careful application-specific evaluation is conducted on the finalists. The top-$k$ identification in the first stage can be essential when the time and resources required for a second stage with all items is prohibitively large.

In both settings, comparisons cost money, time, and/or scarce human attention, which makes adaptivity valuable. Rather than fixing a static sampling plan up front, an experiment designer can adaptively choose which pair to compare next based on outcomes observed so far, allocating effort on comparisons that are most informative for distinguishing the top-$k$ items from the rest. We study this problem in the fixed-confidence setting: an experiment designer adaptively chooses which pair to compare next, observes the outcome, and stops only once they can return the correct top-$k$ set with probability at least $1 - \delta$. The objective is to design such a procedure that minimizes the expected number of comparisons.

A key modeling decision is what to assume about the relationship between the relative ordering of items and their pairwise win probabilities. Some work makes minimal assumptions, allowing situations where $i \succ j$ while

[1]Yale University, New Haven, CT, USA. Correspondence to: Motti Goldberger <motti.goldberger@yale.edu>.

*Proceedings of the 43$^{rd}$ International Conference on Machine Learning*, Seoul, South Korea. PMLR 306, 2026. Copyright 2026 by the author(s).

$\mathbb{P}(i$ beats $j) < \frac{1}{2}$; see, e.g., Haddenhorst et al. (2021); Xu et al. (2019). Other work assumes some form of stochastic transitivity, which rules out such reversals; see, e.g., Braverman et al. (2019); Shah et al. (2017). Finally, latent-utility models assume that items are ordered by unobserved utilities $\boldsymbol{\theta} = (\theta_1, \ldots, \theta_n)$. Under this model there is a non-decreasing link function $f$ such that, for every pair $(i, j)$, $\mathbb{P}(i$ beats $j) = f(\theta_i - \theta_j)$; see, e.g., Saha & Gopalan (2020; 2019).

In this paper, we assume a latent-utility model. Unlike much of the pairwise-comparison and dueling bandits literature which only considers binary outcomes, our algorithm also applies when comparison outcomes are cardinal. In this case, a comparison returns a numerical value indicating how strongly one item is preferred, for example, a score difference or a difference in measured performance.

Prior work on fixed-confidence best-arm (item) identification establishes information-theoretic lower bounds on the asymptotic sample complexity (Kaufmann et al., 2016; Garivier & Kaufmann, 2016), which is easily extended to our setting. Algorithms that match this lower bound as $\delta \to 0$ are called asymptotically optimal. The lower bound is governed by a max–min oracle allocation problem. If the true parameter vector $\boldsymbol{\theta}$ were known, one could allocate pairwise comparisons so that the resulting data would rule out all parameter vectors whose top-$k$ set differs from the true top-$k$ set—so-called "alternative" parameters. The optimal oracle allocation maximizes the rate at which the most difficult alternative can be ruled out.

**Contributions.** For top-$k$ identification with pairwise comparisons, we characterize the structure of this oracle problem. Any alternative parameter vector $\boldsymbol{\theta}'$ must reverse the ordering of at least one pair $(i, j)$ with $i$ in the true top-$k$ and $j$ outside it. We call such a pair a *boundary pair*. We show that the oracle problem can be formulated as a two-player game. A designer allocates comparisons across all $\binom{n}{2}$ pairs, while an adversary chooses which of the $k(n-k)$ boundary pairs to "target". The equilibrium characterizes both the asymptotically optimal sampling strategy and the sample complexity. We construct a comparison allocation procedure based on a primal–dual algorithm that learns this equilibrium online, and prove it is asymptotically optimal.

**Organization.** Section 2 reviews the literature on top-$k$ identification from pairwise comparisons as well as related work on optimal pure exploration in bandit models. Section 3 formalizes the model and objective, and states the information-theoretic lower bound. Section 4 analyzes the oracle allocation problem. Section 5 presents the online comparison allocation algorithm, and Section 6 proves its guarantees. Section 7 demonstrates the algorithm's performance on simulated instances.

## 2. Related Work

### 2.1. Top-$k$ Identification from Pairwise Comparisons

There is a large body of work on sequential learning from noisy pairwise comparisons; see Bengs et al. (2021) for a survey. This includes the extensive literature on top-1 identification, an important special case of fixed confidence top-$k$ identification, which is our focus. We will refer to two standard notions of fixed-confidence guarantees: $\delta$-*correct*, meaning the exact target is returned with probability at least $1 - \delta$, and *probably approximately correct* (PAC), meaning that with probability at least $1 - \delta$ the algorithm returns a near-optimal target.

Early work on adaptive top-$k$ selection from comparisons includes Busa-Fekete et al. (2013), who propose a preference-based racing algorithm for selecting the top-$k$ items using pairwise comparisons. They do not make assumptions about the structure of pairwise preferences, and the "top-$k$" set is defined through a chosen ranking rule (e.g., Copeland's ranking).

In contrast, and closer to our work, Ren et al. (2020) studies $\delta$-*correct* top-$k$ selection assuming strong stochastic transitivity (SST) together with a stochastic triangle inequality (STI); see Ren et al. (2020) for formal definitions. They develop an algorithm, SEEKS, that uses an elimination scheme for top-$k$ selection. In each round, it first chooses a pivot item that is "close" to the current $k$-th best item, compares items to that pivot, and then assigns them into three groups: clearly above the pivot, ambiguous, and clearly below it. Items in the first group are "accepted" into the top-$k$, and items in the third group are eliminated. Items in both of these groups are no longer considered for future comparisons. The algorithm then repeats on the unresolved items in the second group and stops once $k$ items have been accepted, or $n - k$ items have been eliminated. They prove that SEEKS has a sample complexity of $O\big(\sum_{i=1}^{n} \Delta_i^{-2}\big(\log(n/\delta) + \log\log \Delta_i^{-1}\big)\big)$ where $\Delta_i$ is a measure of distance of item $i$ to the $k$-th (or $(k+1)$-th) best item. They also derive a lower bound on the expected number of samples required for any algorithm to be $\delta$-correct, $\Omega\big(\sum_{i=1}^{n} \Delta_i^{-2} \log(1/\delta) + \log\log \Delta_{r_k}^{-1}\big)$. However, while their model is formulated under SST and STI, the lower bound is proven under the following assumptions: (i) comparisons are generated from a Thurstone model i.e., $\mathbb{P}(i$ beats $j) = \mathbb{P}(\theta_i + Z_1 > \theta_j + Z_2)$ with $Z_1$ and $Z_2$ independent Gaussian noise with variance 1, (ii) $\delta \in (0, 1/100)$, (iii) $\theta_1, \ldots, \theta_n \in [0, 1]$. Thus, under these conditions, they show SEEKS is optimal up to a $\log n$ factor, but under more general conditions, the optimality gap may be larger. In contrast, our optimality guarantee is only in the asymptotic regime $\delta \to 0$, but we match the lower bound exactly (including multiplicative constants).

Top-$k$ identification has also been studied under parametric latent-utility models, specifically the multinomial logit (MNL) model, where queries are listwise. At each round, the learner selects a set of $\ell \geq 2$ items and observes which item is most preferred in that set. Chen et al. (2018) give an algorithm that is optimal up to polylog factors, however, their algorithm guarantees correctness with "high probability", but the risk level $\delta$ is not an input parameter. Ren et al. (2018) study PAC top-$k$ selection with both pairwise and listwise queries.

Beyond top-$k$ selection, there is also work that studies adaptive comparison strategies for learning rankings, with top-$k$ identification as a special case; see, e.g., Heckel et al. (2019); Mohajer et al. (2017). Additionally, there is a large literature on non-adaptive top-$k$ recovery from pairwise data under Bradley–Terry / Plackett–Luce models (see, e.g., Chen et al. (2019); Negahban et al. (2017); Jang et al. (2016); Khetan & Oh (2016); Chen & Suh (2015); Hajek et al. (2014)). These works study the accuracy of computationally efficient estimators of the top-$k$ items under prespecified sampling designs.

### 2.2. Pure Exploration and Oracle-Tracking Methods

In the bandit literature, *pure exploration* problems are those in which samples are collected to inform a terminal decision, rather than to maximize the rewards obtained during sampling. For the most studied pure exploration task, best-arm identification, Garivier & Kaufmann (2016) propose the Track-and-Stop framework: a sampling rule that tracks the optimal allocation that governs the lower bound on the sample complexity, together with a stopping rule, and prove optimality as $\delta \to 0$. Degenne et al. (2019) formulate this same max–min lower-bound program as a two-player game between an experiment designer and nature, and develop online tracking schemes that learn the saddle point without solving the full max–min program at every round.

These ideas have been extended to structured bandit models. For linear bandits, Degenne et al. (2020) obtain asymptotically optimal fixed-confidence algorithms by this same "gamification" idea, and Jedra & Proutière (2020) show that a Track-and-Stop approach can be made computationally scalable via lazy updates of the optimal allocation.

Wang et al. (2021) propose a general Frank-Wolfe oracle-tracking method that can be applied to several pure-exploration settings. Their appendix includes an application to top-$k$ identification for dueling bandits under a nonparametric preference-matrix model. However, instantiating the Frank–Wolfe updates requires identifying the (near-)most confusing alternative parameters, which in their case is combinatorial in the number of items. In our latent-utility model, the alternative parameters reduce to parameters that reverse the ranking of boundary pairs, yielding an online learner

with computationally efficient updates.

## 3. Model and Lower Bound

In this section, we formalize the sequential comparison model and state an instance-dependent lower bound on the expected sample complexity, which follows from Kaufmann et al. (2016); Garivier & Kaufmann (2016).

### 3.1. Pairwise Comparison Model

Consider $n$ items indexed $i \in \{1, \ldots, n\}$ each with an unknown utility $\theta_i$. Comparison outcomes depend on utility differences, so $\boldsymbol{\theta}$ is identifiable only up to an additive constant. We assume that the set of parameters is bounded by some constant $R$ and define the parameter space

$$\Theta := \{\boldsymbol{\theta} \in \mathbb{R}^n : \sum_{i=1}^n \theta_i = 0, \|\boldsymbol{\theta}\|_\infty \leq R\}.$$

Let $\mathcal{P} := \{(i,j) : 1 \leq i < j \leq n\}$ denote the set of pairs. For each $(i,j) \in \mathcal{P}$, define $\boldsymbol{x}_{ij} := \boldsymbol{e}_i - \boldsymbol{e}_j$ where $\boldsymbol{e}_i$ is the $i$-th standard basis vector and the natural parameter $\eta_{ij}(\boldsymbol{\theta}) := \boldsymbol{x}_{ij}^\top \boldsymbol{\theta} = \theta_i - \theta_j$. Comparison outcomes follow a one-parameter exponential family with parameter $\eta_{ij}(\boldsymbol{\theta})$: there exist functions $T : \mathcal{Y} \to \mathbb{R}$, $A : \mathbb{R} \to \mathbb{R}$, and $B : \mathcal{Y} \to \mathbb{R}$ such that, for any comparison outcome $y \in \mathcal{Y}$,

$$p(y \mid (i,j), \boldsymbol{\theta}) = \exp\{\eta_{ij}(\boldsymbol{\theta})\, T(y) - A(\eta_{ij}(\boldsymbol{\theta})) + B(y)\}.$$

**Assumption 1.** *$A$ is twice continuously differentiable on $\mathbb{R}$, and*

$$0 < A''(\eta) \leq \overline{\sigma}^2 \quad \text{for all } \eta \in \mathbb{R}$$

*where $\overline{\sigma}^2 < \infty$. Define $\underline{a} := \inf_{|\eta| \leq 2R} A''(\eta) > 0$.*

**Assumption 2.** *There exists $\sigma^2 < \infty$ such that for all $\eta \in [-2R, 2R]$ and all $\lambda \in \mathbb{R}$,*

$$\mathbb{E}_\eta[\exp\{\lambda(T(Y) - A'(\eta))\}] \leq \exp\left(\frac{\sigma^2 \lambda^2}{2}\right).$$

We highlight two standard models that satisfy our assumptions on $\Theta$ and illustrate the binary and cardinal observation regimes.

**Example 3.1** (Bradley–Terry). Let $Y \in \{0, 1\}$ be the random variable indicating whether item $i$ beats item $j$, then given $\boldsymbol{\theta}$,

$$Y \sim \text{Bernoulli}\left(\frac{1}{1 + e^{-(\theta_i - \theta_j)}}\right).$$

**Example 3.2** (Gaussian differences). Let $Y \in \mathbb{R}$ be the random variable denoting the observed difference between $i$ and $j$, then given $\boldsymbol{\theta}$,

$$Y \sim N(\theta_i - \theta_j, \sigma_0^2)$$

where $\sigma_0^2$ is known.

## 3.2. Fixed-Confidence top-$k$ Identification

Fix $k \in \{1, \ldots, n-1\}$ and let $\theta_{(1)} \geq \cdots \geq \theta_{(n)}$ denote the order statistics of $\boldsymbol{\theta}$. Assume a positive top-$k$ gap, $\theta_{(k)} > \theta_{(k+1)}$ and define $\Theta^{\text{gap}} := \{\boldsymbol{\theta} \in \Theta : \theta_{(k)} > \theta_{(k+1)}\}$. On $\Theta^{\text{gap}}$, the top-$k$ set $S^*(\boldsymbol{\theta}) := \{i : \theta_i \geq \theta_{(k)}\}$ is unique.

**Assumption 3.** *The true utility parameter satisfies* $\boldsymbol{\theta} \in \text{int}(\Theta) \cap \Theta^{\text{gap}}$.

At each round $t = 1, 2, \ldots$ the algorithm selects a pair $(i_t, j_t) \in \mathcal{P}$ and observes $Y_t \sim p(\cdot \mid (i_t, j_t), \boldsymbol{\theta})$. Conditioned on the selected pairs, outcomes are independent. Formally, a sequential strategy consists of:

- a *sampling rule*: a sequence $\{(i_t, j_t)\}_{t \geq 1}$ where $(i_t, j_t)$ is $\mathcal{F}_{t-1}$-measurable and $\mathcal{F}_t := \sigma((i_s, j_s, Y_s) : s \leq t)$;
- a *stopping rule*: a stopping time $\tau$ with respect to $(\mathcal{F}_t)$;
- a *decision rule*: an $\mathcal{F}_\tau$-measurable subset $\hat{S}_\tau \subset [n]$ with $|\hat{S}_\tau| = k$.

**Definition 3.3** ($\delta$-correct). A strategy is $\delta$-correct if for every $\boldsymbol{\theta} \in \Theta^{\text{gap}}$,

$$\mathbb{P}_{\boldsymbol{\theta}}(\hat{S}_\tau \neq S^*(\boldsymbol{\theta})) \leq \delta \quad \text{and} \quad \mathbb{P}_{\boldsymbol{\theta}}(\tau < \infty) = 1.$$

### 3.3. Information-Theoretic Lower Bound

For a single comparison on pair $(i, j)$, let $d_{ij}(\boldsymbol{\theta}, \boldsymbol{\theta}') := \text{KL}(P_{\eta_{ij}(\boldsymbol{\theta})} \| P_{\eta_{ij}(\boldsymbol{\theta}')})$ denote the KL divergence between outcome distributions under $\boldsymbol{\theta}$ and $\boldsymbol{\theta}'$. A *sampling design* is a distribution $\boldsymbol{w} \in \Delta_{\mathcal{P}}$ over pairs, where $\Delta_{\mathcal{P}} := \{\boldsymbol{w} \in \mathbb{R}_+^{|\mathcal{P}|} : \sum_{(i,j)} w_{ij} = 1\}$ ($w_{ij}$ is the proportion of comparisons allocated to $(i, j)$). Under design $\boldsymbol{w}$, define

$$D_{\boldsymbol{w}}(\boldsymbol{\theta} \| \boldsymbol{\theta}') := \sum_{(i,j) \in \mathcal{P}} w_{ij} \, d_{ij}(\boldsymbol{\theta}, \boldsymbol{\theta}').$$

Ultimately, our goal is to distinguish $\boldsymbol{\theta}$ from alternative parameters $\boldsymbol{\theta}'$ whose top-$k$ set is different. Accordingly, for $\boldsymbol{\theta} \in \Theta^{\text{gap}}$ define the alternative region

$$\text{Alt}(\boldsymbol{\theta}) := \{\boldsymbol{\theta}' \in \Theta : \exists u \in S^*(\boldsymbol{\theta}), v \notin S^*(\boldsymbol{\theta}) \text{ s.t. } \theta'_v \geq \theta'_u\}$$

and the information rate

$$\Gamma^*(\boldsymbol{\theta}) := \sup_{\boldsymbol{w} \in \Delta_{\mathcal{P}}} \inf_{\boldsymbol{\theta}' \in \text{Alt}(\boldsymbol{\theta})} D_{\boldsymbol{w}}(\boldsymbol{\theta} \| \boldsymbol{\theta}').$$

Intuitively, $\Gamma^*(\boldsymbol{\theta})$ is the most per-sample information we can learn for distinguishing $\boldsymbol{\theta}$ from the hardest alternatives. We now present the lower bound on the expected stopping time given by Garivier & Kaufmann (2016).

**Theorem 3.4** (Garivier & Kaufmann (2016)). *Let* $\delta \in (0, 1)$ *and let* $(\{(i_t, j_t)\}, \tau_\delta, \hat{S}_\tau)$ *be a $\delta$-correct strategy. For any* $\boldsymbol{\theta} \in \Theta^{\text{gap}}$,

$$\liminf_{\delta \to 0} \frac{\mathbb{E}_{\boldsymbol{\theta}}[\tau_\delta]}{\log(1/\delta)} \geq \frac{1}{\Gamma^*(\boldsymbol{\theta})}.$$

The Track-and-Stop algorithm of Garivier & Kaufmann (2016) gives a framework for sequential algorithms to track the optimal oracle allocation $\boldsymbol{w}^*(\boldsymbol{\theta})$. In the next section, we will characterize this oracle allocation for the pairwise top-$k$ setting. Then, building on the Track-and-Stop framework, we will develop a sequential algorithm that tracks the oracle allocation and prove that it is asymptotically optimal.

## 4. Structure of the Oracle Problem

The oracle's allocation problem is, given $\boldsymbol{\theta}$, to find a design $\boldsymbol{w}^* \in \text{argmax}_{\boldsymbol{w} \in \Delta_{\mathcal{P}}} \inf_{\boldsymbol{\theta}' \in \text{Alt}(\boldsymbol{\theta})} D_{\boldsymbol{w}}(\boldsymbol{\theta} \| \boldsymbol{\theta}')$. In this section, we (i) formulate this problem as a two-player game, (ii) show that a saddle-point exists, and (iii) calculate the gradients that will be used in the online algorithm, when, in contrast to the oracle problem, $\boldsymbol{\theta}$ is not known.

### 4.1. Boundary Reduction

We start by characterizing the alternative region $\text{Alt}(\boldsymbol{\theta})$ for top-$k$ identification. Define the set of boundary pairs

$$B(\boldsymbol{\theta}) := \{(i, j) \in \mathcal{P} : \mathbf{1}\{i \in S^*(\boldsymbol{\theta})\} \neq \mathbf{1}\{j \in S^*(\boldsymbol{\theta})\}\},$$

so $|B(\boldsymbol{\theta})| = k(n - k)$. For each $(i, j) \in B(\boldsymbol{\theta})$, let $u_{ij} \in S^*(\boldsymbol{\theta})$ and $v_{ij} \notin S^*(\boldsymbol{\theta})$ denote the endpoints lying inside and outside the true top-$k$ set, respectively. Define the set of parameters that invert the true ordering, $\Theta_{ij} := \{\boldsymbol{\theta}' \in \Theta : \theta'_{v_{ij}} \geq \theta'_{u_{ij}}\}$. It follows from the definition of $\text{Alt}(\boldsymbol{\theta})$ that if $\boldsymbol{\theta}' \in \text{Alt}(\boldsymbol{\theta})$, there is some boundary pair $(i, j) \in B(\boldsymbol{\theta})$ such that $\boldsymbol{\theta}' \in \Theta_{ij}$. Thus, $\text{Alt}(\boldsymbol{\theta}) = \bigcup_{(i,j) \in B(\boldsymbol{\theta})} \Theta_{ij}$, and the oracle objective can be written as

$$\Gamma^*(\boldsymbol{\theta}) = \sup_{\boldsymbol{w} \in \Delta_{\mathcal{P}}} \min_{(i,j) \in B(\boldsymbol{\theta})} \inf_{\boldsymbol{\theta}' \in \Theta_{ij}} D_{\boldsymbol{w}}(\boldsymbol{\theta} \| \boldsymbol{\theta}').$$

### 4.2. Saddle Point Formulation

Let $\gamma_{ij}(\boldsymbol{w}; \boldsymbol{\theta}) := \inf_{\boldsymbol{\theta}' \in \Theta_{ij}} D_{\boldsymbol{w}}(\boldsymbol{\theta} \| \boldsymbol{\theta}')$ and define the information rate under $\boldsymbol{w}$, $\Gamma(\boldsymbol{w}; \boldsymbol{\theta}) := \min_{(i,j) \in B(\boldsymbol{\theta})} \gamma_{ij}(\boldsymbol{w}; \boldsymbol{\theta})$. The minimum over boundary pairs is nonsmooth in general, so we write $\Gamma(\boldsymbol{w}; \boldsymbol{\theta}) = \min_{\boldsymbol{q} \in \Delta_{B(\boldsymbol{\theta})}} F(\boldsymbol{w}, \boldsymbol{q}; \boldsymbol{\theta})$, where $F(\boldsymbol{w}, \boldsymbol{q}; \boldsymbol{\theta}) := \sum_{(i,j) \in B(\boldsymbol{\theta})} q_{ij} \gamma_{ij}(\boldsymbol{w}; \boldsymbol{\theta})$. Consequently,

$$\Gamma^*(\boldsymbol{\theta}) = \max_{\boldsymbol{w} \in \Delta_{\mathcal{P}}} \min_{\boldsymbol{q} \in \Delta_{B(\boldsymbol{\theta})}} F(\boldsymbol{w}, \boldsymbol{q}; \boldsymbol{\theta}).$$

We view this as a two-player game. The designer chooses allocation proportions $\boldsymbol{w}$, and an adversary chooses $\boldsymbol{q} \in \Delta_{B(\boldsymbol{\theta})}$, a distribution over boundary pairs. For any fixed $\boldsymbol{w}$, the inner minimization over $\boldsymbol{q}$ is equivalent to taking the minimum over $(i, j) \in B(\boldsymbol{\theta})$: an optimal $\boldsymbol{q}$ allocates its mass on the boundary pair(s) attaining $\min_{(i,j) \in B(\boldsymbol{\theta})} \gamma_{ij}(\boldsymbol{w}; \boldsymbol{\theta})$. Informally, $\boldsymbol{q}$ identifies the boundary pairs under design $\boldsymbol{w}$ that are most "vulnerable" to being misordered.

**Lemma 4.1** (Basic properties of $\gamma_{ij}$). *Fix* $\boldsymbol{\theta} \in \Theta$ *and* $(i, j) \in B(\boldsymbol{\theta})$. *The map* $\boldsymbol{w} \mapsto \gamma_{ij}(\boldsymbol{w}; \boldsymbol{\theta})$ *is concave and continuous on* $\Delta_{\mathcal{P}}$.

By Lemma 4.1, $F(\cdot, \boldsymbol{q}; \boldsymbol{\theta}) = \sum_{(i,j) \in B(\boldsymbol{\theta})} q_{ij} \gamma_{ij}(\cdot; \boldsymbol{\theta})$ is concave and continuous in $\boldsymbol{w}$ for each $\boldsymbol{q}$, and $F(\boldsymbol{w}, \cdot; \boldsymbol{\theta})$ is linear (hence convex and continuous) in $\boldsymbol{q}$ for each $\boldsymbol{w}$. Since $\Delta_{\mathcal{P}}$ and $\Delta_{B(\boldsymbol{\theta})}$ are nonempty compact convex sets, Sion's minimax theorem gives

$$\max_{\boldsymbol{w} \in \Delta_{\mathcal{P}}} \min_{\boldsymbol{q} \in \Delta_{B(\boldsymbol{\theta})}} F(\boldsymbol{w}, \boldsymbol{q}; \boldsymbol{\theta}) = \min_{\boldsymbol{q} \in \Delta_{B(\boldsymbol{\theta})}} \max_{\boldsymbol{w} \in \Delta_{\mathcal{P}}} F(\boldsymbol{w}, \boldsymbol{q}; \boldsymbol{\theta}).$$

Moreover, by compactness and continuity the max and min are attained, so there exists $(\boldsymbol{w}^*, \boldsymbol{q}^*) \in \Delta_{\mathcal{P}} \times \Delta_{B(\boldsymbol{\theta})}$ such that $\Gamma^*(\boldsymbol{\theta}) = F(\boldsymbol{w}^*, \boldsymbol{q}^*; \boldsymbol{\theta})$ and

$$F(\boldsymbol{w}, \boldsymbol{q}^*; \boldsymbol{\theta}) \leq F(\boldsymbol{w}^*, \boldsymbol{q}^*; \boldsymbol{\theta}) \leq F(\boldsymbol{w}^*, \boldsymbol{q}; \boldsymbol{\theta})$$
$$\forall (\boldsymbol{w}, \boldsymbol{q}) \in \Delta_{\mathcal{P}} \times \Delta_{B(\boldsymbol{\theta})}.$$

To compute gradients of $F$ we need the KL projection $\inf_{\boldsymbol{\theta}' \in \Theta_{ij}} D_{\boldsymbol{w}}(\boldsymbol{\theta} \| \boldsymbol{\theta}')$ to exist and be unique. This is ensured by strict convexity of $D_{\boldsymbol{w}}(\boldsymbol{\theta} \| \cdot)$ over $\Theta$, which is guaranteed when the sampling design has connected support. Define the support graph

$$G(\boldsymbol{w}) := ([n], \{\{i, j\} : w_{ij} > 0\}).$$

$G(\boldsymbol{w})$ is connected if for any $u, v \in [n]$ there exists a path $u = v_0, \ldots, v_m = v$ in $G(\boldsymbol{w})$.

**Lemma 4.2** (Well-posedness of KL projection on $\Theta$). *Fix $\boldsymbol{\theta} \in \Theta$ and $\boldsymbol{w} \in \Delta_{\mathcal{P}}$ with $G(\boldsymbol{w})$ connected. For each $(i, j) \in B(\boldsymbol{\theta})$:*

*(a) The map $\boldsymbol{\theta}' \mapsto D_{\boldsymbol{w}}(\boldsymbol{\theta} \| \boldsymbol{\theta}')$ is strictly convex on $\Theta$*
*(b) $\boldsymbol{\theta}_{ij}^*(\boldsymbol{w}; \boldsymbol{\theta}) := \arg\min_{\boldsymbol{\theta}' \in \Theta_{ij}} D_{\boldsymbol{w}}(\boldsymbol{\theta} \| \boldsymbol{\theta}')$ is unique*
*(c) Either $(\boldsymbol{\theta}_{ij}^*)_{u_{ij}} = (\boldsymbol{\theta}_{ij}^*)_{v_{ij}}$, or $\boldsymbol{\theta}_{ij}^*(\boldsymbol{w}; \boldsymbol{\theta}) \in \partial\Theta$.*

It follows by Danskin's theorem that $\gamma_{ij}(\cdot; \boldsymbol{\theta})$ is differentiable at $\boldsymbol{w}$ such that $G(\boldsymbol{w})$ is connected and

$$\frac{\partial \gamma_{ij}}{\partial w_{ab}}(\boldsymbol{w}; \boldsymbol{\theta}) = d_{ab}(\boldsymbol{\theta}, \boldsymbol{\theta}_{ij}^*(\boldsymbol{w}; \boldsymbol{\theta})), \qquad (a, b) \in \mathcal{P}.$$

hence, the partial derivatives of $F$ are

$$\frac{\partial F}{\partial w_{ab}}(\boldsymbol{w}, \boldsymbol{q}; \boldsymbol{\theta}) = \sum_{(i,j) \in B(\boldsymbol{\theta})} q_{ij} d_{ab}(\boldsymbol{\theta}, \boldsymbol{\theta}_{ij}^*(\boldsymbol{w}; \boldsymbol{\theta})),$$
$$\frac{\partial F}{\partial q_{ij}}(\boldsymbol{w}, \boldsymbol{q}; \boldsymbol{\theta}) = \gamma_{ij}(\boldsymbol{w}; \boldsymbol{\theta}). \tag{1}$$

The KL projection $\boldsymbol{\theta}_{ij}^*(\boldsymbol{w}; \boldsymbol{\theta})$ minimizes $D_{\boldsymbol{w}}(\boldsymbol{\theta} \| \boldsymbol{\theta}')$ over $\Theta_{ij}$. Lemma 4.2 (c) says that, unless this minimizer is forced onto $\partial\Theta$ by the constraint $\|\boldsymbol{\theta}'\|_\infty \leq R$, the halfspace constraint $(\boldsymbol{\theta}_{ij}^*)_{v_{ij}} = (\boldsymbol{\theta}_{ij}^*)_{u_{ij}}$ is active; in other words, the most confusing alternative lies on the hyperplane $\theta'_{v_{ij}} = \theta'_{u_{ij}}$.

**Remark 4.3** (Disconnected sampling designs). If, contrary to our assumption, the parameter class were unbounded and $G(\boldsymbol{w})$ were disconnected, then $\Gamma(\boldsymbol{w}; \boldsymbol{\theta}) = 0$. Hence, if

$\boldsymbol{w}^*$ is an optimal design, $G(\boldsymbol{w}^*)$ is connected. To see this, pick a boundary pair $(u, v) \in B(\boldsymbol{\theta})$ in different connected components of $G(\boldsymbol{w})$. Shift the utilities of all items in the component of $v$ upward and the component of $u$ downward (to preserve $\sum_i \theta'_i = 0$). This flips $\theta'_v \geq \theta'_u$ while leaving all $\eta_{ab}(\boldsymbol{\theta}')$ unchanged for any $(a, b)$ with positive weight, so $D_{\boldsymbol{w}}(\boldsymbol{\theta} \| \boldsymbol{\theta}') = 0$ and thus $\gamma_{uv}(\boldsymbol{w}; \boldsymbol{\theta}) = 0$. On the bounded class $\Theta$, this construction may be infeasible due to $\|\boldsymbol{\theta}'\|_\infty \leq R$, so the conclusion may not hold. However, in Section 5 we show that we can still apply Lemma 4.2.

## 5. Algorithm

We now construct an adaptive fixed-confidence procedure when $\boldsymbol{\theta}$ is not known. The sampling rule is driven by the solution to the oracle saddle-point problem. At round $t$, we compute the current MLE $\hat{\boldsymbol{\theta}}_t$ and its induced boundary set $\hat{B}_t := B(\hat{\boldsymbol{\theta}}_t)$, and evaluate the oracle objective at this estimate, that is we work with $F(\cdot, \cdot; \hat{\boldsymbol{\theta}}_t)$ on $\Delta_{\mathcal{P}} \times \Delta_{\hat{B}_t}$. Re-computing a saddle point at every $t$ is expensive and early on potentially unnecessary when the MLE has high variance. Instead, we learn the saddle point online, maintaining iterates $(\boldsymbol{w}_t, \boldsymbol{q}_t)$ and performing one primal–dual update step per round.

Given $(\boldsymbol{w}_t, \boldsymbol{q}_t, \hat{\boldsymbol{\theta}}_{t-1})$, a round proceeds as follows: (i) select the next pair to sample $A_t = (i_t, j_t)$ based on $\{\boldsymbol{w}_s\}_{s=1}^t$; (ii) observe outcome $Y_t$ and update $\hat{\boldsymbol{\theta}}_t$, $\hat{S}_t = S^*(\hat{\boldsymbol{\theta}}_t)$, and $\hat{B}_t = B(\hat{\boldsymbol{\theta}}_t)$; (iii) check a stopping test to see if we are confident enough to terminate. (iv) If we did not stop, perform one entropic FTRL primal–dual update using the estimated gradients of $F(\cdot, \cdot; \hat{\boldsymbol{\theta}}_t)$ at the current iterates $(\boldsymbol{w}_t, \boldsymbol{q}_t)$, producing the next iterates $(\boldsymbol{w}_{t+1}, \boldsymbol{q}_{t+1})$. A high-level summary of the procedure is given below; the full algorithm pseudocode with notation is provided at the end of this section in Algorithm 1.

---

**High-level Outline of Algorithm 1**

---

1: **for** $t = 1, 2, \ldots$ **do**
2:     Select a pair by tracking $\{\boldsymbol{w}_s\}_{s=1}^t$.    (Section 5.4)
3:     Observe outcome $Y_t$; update $\hat{\boldsymbol{\theta}}_t$.    (Section 5.1)
4:     Check the stopping condition; if it is satisfied output $\hat{S}_t$ and stop.    (Section 5.5)
5:     Estimate the gradient of the current oracle game $F(\boldsymbol{w}_t, \boldsymbol{q}_t; \hat{\boldsymbol{\theta}}_t)$.    (Section 5.3)
6:     Update $\boldsymbol{w}_t$ and $\boldsymbol{q}_t$ to $\boldsymbol{w}_{t+1}$ and $\boldsymbol{q}_{t+1}$ via entropic FTRL.    (Section 5.2)
7: **end for**

---

## 5.1. Maximum Likelihood Estimate

Given observations $(A_1, Y_1), \ldots, (A_t, Y_t)$, define the log-likelihood (up to additive constants)

$$\ell_t(\boldsymbol{\theta}) := \sum_{s=1}^{t} \big[ \eta_{A_s}(\boldsymbol{\theta}) \, T(Y_s) - A(\eta_{A_s}(\boldsymbol{\theta})) \big].$$

Let $\hat{\boldsymbol{\theta}}_t \in \mathrm{argmax}_{\boldsymbol{\theta} \in \Theta} \, \ell_t(\boldsymbol{\theta})$ denote the MLE.

## 5.2. Entropic FTRL Updates

We update the primal and dual iterates from $(\boldsymbol{w}_t, \boldsymbol{q}_t)$ to $(\boldsymbol{w}_{t+1}, \boldsymbol{q}_{t+1})$ using entropic FTRL with gradient estimates of $F(\cdot, \cdot; \hat{\boldsymbol{\theta}}_t)$. The primal player maximizes over $\boldsymbol{w} \in \Delta_{\mathcal{P}}$; the dual player minimizes over $\boldsymbol{q} \in \Delta_{\hat{B}_t}$. At round $t$, we construct unbiased gradient estimates $\hat{\boldsymbol{g}}_t^{(w)}$ and $\hat{\boldsymbol{g}}_t^{(q)}$ for the partial derivatives in equation (1) evaluated at $(\boldsymbol{w}_t, \boldsymbol{q}_t; \hat{\boldsymbol{\theta}}_t)$. Given these estimates, define cumulative scores

$$\boldsymbol{\Psi}_t^{(w)} := \boldsymbol{\Psi}_{t-1}^{(w)} + \hat{\boldsymbol{g}}_t^{(w)}, \qquad \boldsymbol{\Psi}_t^{(q)} := \boldsymbol{\Psi}_{t-1}^{(q)} + \hat{\boldsymbol{g}}_t^{(q)}$$

with $\boldsymbol{\Psi}_0^{(w)} = \boldsymbol{\Psi}_0^{(q)} = \boldsymbol{0}$. The entropic FTRL updates are defined by

$$\boldsymbol{w}_{t+1} = \underset{\boldsymbol{w} \in \Delta_{\mathcal{P}}}{\mathrm{argmax}} \left\{ \langle \boldsymbol{w}, \boldsymbol{\Psi}_t^{(w)} \rangle - \frac{1}{\mu_{t+1}} \mathrm{KL}(\boldsymbol{w} \| \boldsymbol{w}_1) \right\},$$

$$\boldsymbol{q}_{t+1} = \underset{\boldsymbol{q} \in \Delta_{\hat{B}_t}}{\mathrm{argmin}} \left\{ \langle \boldsymbol{q}, \boldsymbol{\Psi}_t^{(q)} \rangle + \frac{1}{\mu_{t+1}} \mathrm{KL}(\boldsymbol{q} \| \boldsymbol{q}_1) \right\}.$$

For each $(a, b) \in \mathcal{P}$ it follows that the primal updates are

$$w_{t+1,ab} = \frac{w_{1,ab} \exp\big( \mu_{t+1} \Psi_{t,ab}^{(w)} \big)}{\sum_{(c,d) \in \mathcal{P}} w_{1,cd} \exp\big( \mu_{t+1} \Psi_{t,cd}^{(w)} \big)} \tag{2}$$

For the dual variable, the feasible set $\Delta_{\hat{B}_t}$ depends on $\hat{\boldsymbol{\theta}}_t$ through $\hat{B}_t$, so it is convenient to implement the update by maintaining an iterate over all pairs and then restricting to the current boundary set. Define the unnormalized weights

$$r_{t+1,ij} := q_{1,ij} \exp\big( -\mu_{t+1} \Psi_{t,ij}^{(q)} \big), \qquad (i, j) \in \mathcal{P}.$$

Then the solution of the dual FTRL problem above is obtained by restricting and renormalizing on the current boundary set:

$$q_{t+1,ij} := \frac{r_{t+1,ij}}{\sum_{(u,v) \in \hat{B}_t} r_{t+1,uv}} \cdot \mathbf{1}\{(i, j) \in \hat{B}_t\}.$$

Initializing $\boldsymbol{w}_1$ to be uniform over $\mathcal{P}$ and updating via FTRL ensures $w_{t,ab} > 0$ for every pair and every $t$, so $G(\boldsymbol{w}_t)$ is complete (hence connected) for all $t$. By Lemma 4.2, each KL projection $\boldsymbol{\theta}_{ij}^*(\boldsymbol{w}_t; \hat{\boldsymbol{\theta}}_t)$ exists and is unique, so the derivatives in (1) are well-defined at $(\boldsymbol{w}_t, \boldsymbol{q}_t; \hat{\boldsymbol{\theta}}_t)$.

The learning rate $\mu_t$ controls how strongly each new gradient increment changes the weights. To obtain standard no-regret guarantees we take $\mu_t = t^{-\alpha}$, where $\alpha \in (0, 1)$, so the updates become progressively less aggressive as $t$ grows.

## 5.3. Stochastic Gradients

Implementing the updates above requires evaluating the partial derivatives in (1). This requires finding $\boldsymbol{\theta}_{ij}^*(\boldsymbol{w}_t; \hat{\boldsymbol{\theta}}_t) \in \mathrm{argmin}_{\boldsymbol{\theta}' \in \Theta_{ij}(t)} D_{\boldsymbol{w}_t}(\hat{\boldsymbol{\theta}}_t \| \boldsymbol{\theta}')$ for each boundary pair $(i, j) \in \hat{B}_t$. Each projection can be solved in $O(n^2)$ time, so solving all $k(n - k)$ projections can make the algorithm unusable in many cases. Instead, we use an unbiased gradient estimate obtained by sampling a single boundary pair. Sample $I_t \sim \mathrm{Unif}(\hat{B}_t)$ and compute

$$\boldsymbol{\theta}_t^* \in \underset{\boldsymbol{\theta}' \in \Theta_{I_t}(t)}{\mathrm{argmin}} \, D_{\boldsymbol{w}_t}(\hat{\boldsymbol{\theta}}_t \| \boldsymbol{\theta}'), \quad \gamma_t := D_{\boldsymbol{w}_t}(\hat{\boldsymbol{\theta}}_t \| \boldsymbol{\theta}_t^*).$$

Define the gradient estimates

$$\begin{aligned} \hat{g}_{t,ab}^{(w)} &:= m \, q_{t,I_t} \, d_{ab}(\hat{\boldsymbol{\theta}}_t, \boldsymbol{\theta}_t^*), \quad (a, b) \in \mathcal{P}, \\ \hat{g}_{t,ij}^{(q)} &:= m \, \gamma_t \, \mathbf{1}\{(i, j) = I_t\}, \quad (i, j) \in \hat{B}_t, \end{aligned} \tag{3}$$

where $m = k(n - k)$. Conditioned on the past iterates, $\mathbb{E}[\hat{\boldsymbol{g}}_t^{(w)}] = \nabla_{\boldsymbol{w}} F(\boldsymbol{w}_t, \boldsymbol{q}_t; \hat{\boldsymbol{\theta}}_t)$ and $\mathbb{E}[\hat{\boldsymbol{g}}_t^{(q)}] = \nabla_{\boldsymbol{q}} F(\boldsymbol{w}_t, \boldsymbol{q}_t; \hat{\boldsymbol{\theta}}_t)$.

**Remark 5.1.** With full gradients, we would not need to maintain iterates $\boldsymbol{q}_t$: we could update $\boldsymbol{w}_t$ directly by a supergradient-ascent step on $\Gamma(\boldsymbol{w}; \hat{\boldsymbol{\theta}}_t) = \min_{(i,j) \in \hat{B}_t} \gamma_{ij}(\boldsymbol{w}; \hat{\boldsymbol{\theta}}_t)$. In particular, if $(i^*, j^*) \in \mathrm{argmin}_{(i,j) \in \hat{B}_t} \gamma_{ij}(\boldsymbol{w}_t; \hat{\boldsymbol{\theta}}_t)$, then by Danskin's theorem, $\nabla_{\boldsymbol{w}} \gamma_{i^*j^*}(\boldsymbol{w}_t; \hat{\boldsymbol{\theta}}_t)$ is a supergradient of $\Gamma(\cdot; \hat{\boldsymbol{\theta}}_t)$ at $\boldsymbol{w}_t$. Equivalently, we can always choose a best response $\boldsymbol{q}^*(\boldsymbol{w}_t)$ that is a point mass on an active minimizer. We maintain $\boldsymbol{q}_t$ only because we use stochastic gradients: we do not know which boundary pair(s) attain the smallest $\gamma_{ij}(\boldsymbol{w}_t; \hat{\boldsymbol{\theta}}_t)$, so $\boldsymbol{q}_t$ provides an online estimate of which boundary pairs are acting as bottlenecks, which then determines the weight $q_{t,I_t}$ in (3).

## 5.4. Sampling via C-Tracking

To select the pair $A_t$, we use the C-tracking procedure from Garivier & Kaufmann (2016). Fix a mixing schedule $\rho_t$ and form the mixed target $\tilde{\boldsymbol{w}}_t := (1 - \rho_t) \boldsymbol{w}_t + \rho_t \boldsymbol{u}$, where $\boldsymbol{u}$ is uniform weights over all pairs. Define $P_{ij}(t) := \sum_{s=1}^{t} \tilde{w}_{s,ij}$ and select

$$A_t \in \underset{(i,j) \in \mathcal{P}}{\mathrm{argmax}} \big( P_{ij}(t) - N_{ij}(t - 1) \big),$$

where $N_{ij}(t) := \sum_{s=1}^{t} \mathbf{1}\{A_s = (i, j)\}$ are the empirical counts.

C-tracking ensures the empirical proportions $\hat{w}_{t,ij}^{\mathrm{emp}} := N_{ij}(t)/t$ track the running average $(1/t) \sum_{s=1}^{t} \tilde{w}_{s,ij}$, which converges to an optimal design. The uniform mixture $\rho_t \boldsymbol{u}$ forces exploration of all pairs and taking $\rho_t = t^{-\gamma}$, where $\gamma \in (0, 1)$, ensures that this exploration does not bias the allocations asymptotically.

## 5.5. Stopping Rule

The threshold (4) and stopping time (5) are derived using a standard likelihood-ratio martingale argument (Kaufmann & Koolen, 2021) combined with a self-normalized concentration bound used commonly in linear bandits (Abbasi-Yadkori et al., 2011; Lattimore & Szepesvári, 2020). For each estimated boundary pair $(i, j) \in \hat{B}_t$ (w.l.o.g. $\hat{\theta}_{t,i} \geq \hat{\theta}_{t,(k)} > \hat{\theta}_{t,j}$), the statistic $Z_{ij}(t)$ compares the likelihood at the current MLE $\hat{\theta}_t$ to the largest likelihood under alternative parameters satisfying $\theta_i' \leq \theta_j'$. Larger values indicate that all parameter vectors that swap this boundary pair explain the data significantly worse than the MLE. The stopping condition given in Equation (5) is satisfied when the minimum of these boundary-pair statistics exceeds the threshold in (4).

For each boundary pair $(i, j) \in \hat{B}_t$, compute the $Z_{ij}(t) := \ell_t(\hat{\theta}_t) - \sup_{\theta' \in \Theta_{ij}(t)} \ell_t(\theta')$. For any $\lambda > 0$ define the threshold

$$\beta(t, \delta) := \log \frac{1}{\delta} + \frac{\lambda}{2} \|\hat{\theta}_t\|_2^2 + \frac{1}{2} \log \det\left(I_n + \frac{\overline{\sigma}^2}{\lambda} \mathcal{L}_t\right), \quad (4)$$

where $\mathcal{L}_t := \sum_{s=1}^{t} x_{A_s} x_{A_s}^\top$ and $\overline{\sigma}^2$ is from Assumption 1. We define the stopping time

$$\tau_\delta := \inf\left\{ t \geq 1 : \min_{(i,j) \in \hat{B}_t} Z_{ij}(t) \geq \beta(t, \delta) \right\}. \quad (5)$$

Upon stopping, output $\hat{S}_{\tau_\delta} = S^*(\hat{\theta}_{\tau_\delta})$.

Computing the stopping condition can be costly. It requires computing $Z_{ij}(t)$ for every boundary pair, which scales with $O(k(n-k)n^2)$. In practice, for large problems, the stopping check can be performed on a sparse schedule to reduce computational cost, with only a small impact on the expected stopping time, and no impact on the asymptotic optimality guarantee.

# 6. Theoretical Guarantees of Algorithm 1

In this section, we establish theoretical guarantees for Algorithm 1. We show: (i) the procedure is $\delta$-correct (Proposition 6.4), and (ii) its expected stopping time asymptotically matches the information-theoretic lower bound of Theorem 3.4 (Theorem 6.5). The main technical step is Proposition 6.3 showing that for sufficiently large $t$ with high probability the empirical information rate $\Gamma(\hat{w}_t^{\text{emp}}; \theta)$ is close to the oracle value $\Gamma^*(\theta)$. We start by showing the MLE is consistent.

**Lemma 6.1.** *Define $N_{\min}(t) := \min_{(i,j) \in \mathcal{P}} N_{ij}(t)$. There is a finite time $t_{\text{warm}}$ such that for every $t \geq t_{\text{warm}}$ and every $\delta \in (0, 1)$, with probability at least $1 - \delta$,*

$$\|\hat{\theta}_t - \theta\|_2 \leq \frac{2\sigma}{\underline{a}} \sqrt{\frac{n \log\left(\frac{2t|\mathcal{P}|}{\delta}\right)}{N_{\min}(t)}}.$$

---

**Algorithm 1** Oracle-Tracking Top-$k$ Identification

**Require:** Risk $\delta$, exponents $\alpha, \gamma \in (0, 1)$
1: Initialize $w_1 = q_1 = r_1 = u$; $\Psi_0^{(w)} = \Psi_0^{(q)} = 0$; $N_{ij}(0) = P_{ij}(0) = 0$
2: **for** $t = 1, 2, \ldots$ **do**
3:      Set $\mu_{t+1} = (t+1)^{-\alpha}$, $\rho_t = t^{-\gamma}$, $\tilde{w}_t = (1-\rho_t)w_t + \rho_t u$
4:      Select $A_t \in \mathrm{argmax}_{(i,j) \in \mathcal{P}} \{ P_{ij}(t-1) + \tilde{w}_{t,ij} - N_{ij}(t-1) \}$
5:      Observe $Y_t$; update MLE $\hat{\theta}_t$; set $\hat{B}_t = B(\hat{\theta}_t)$
6:      If $\min_{(i,j) \in \hat{B}_t} Z_{ij}(t) \geq \beta(t, \delta)$: output $S^*(\hat{\theta}_t)$ and stop
7:      $q_{t,ij} \leftarrow \frac{r_{t,ij}}{\sum_{(u,v) \in \hat{B}_t} r_{t,uv}} \mathbf{1}\{(i,j) \in \hat{B}_t\}$
8:      Sample $I_t \sim \mathrm{Unif}(\hat{B}_t)$; compute $\theta_t^* = \mathrm{argmin}_{\theta' \in \Theta_{I_t}} D_{w_t}(\hat{\theta}_t \| \theta')$
9:      $\hat{g}_{t,ab}^{(w)} \leftarrow m q_{t,I_t} d_{ab}(\hat{\theta}_t, \theta_t^*);$    $\hat{g}_{t,ij}^{(q)} \leftarrow m \gamma_{I_t}(w_t; \hat{\theta}_t) \mathbf{1}\{(i,j) = I_t\}$
10:      $\Psi_t^{(w)} \leftarrow \Psi_{t-1}^{(w)} + \hat{g}_t^{(w)};$ $\Psi_t^{(q)} \leftarrow \Psi_{t-1}^{(q)} + \hat{g}_t^{(q)}$
11:      Update $(w_{t+1}, r_{t+1})$ by FTRL on $F(\cdot, \cdot; \hat{\theta}_t)$
12: **end for**

---

**Corollary 6.2.** $\hat{\theta}_t \to \theta$ *almost surely. Consequently, there exists almost surely a finite time $t_{\text{stab}}$ such that for all $t \geq t_{\text{stab}}$, $\hat{S}_t = S^*(\theta)$, and $\hat{B}_t = B(\theta)$.*

For the analysis in Proposition 6.3 we work on a truncated window. Fix $t$ and define the burn-in index $b_t := \lceil t^{1/4} \rceil$. A union bound applied to Lemma 6.1 over $s = b_t, \ldots, t$ yields an event of probability at least $1 - t^{-p}$ on which $\hat{B}_s = B(\theta)$ for all $s \in \{b_t, \ldots, t\}$. This is the regime in which the oracle-tracking regret analysis applies.

**Proposition 6.3** (High-probability oracle-rate lower bound). *Fix any $p > 1$. There exist constants $C_p < \infty$ and $t_p < \infty$ such that for all $t \geq t_p$,*

$$\mathbb{P}_\theta\Bigg( \Gamma(\hat{w}_t^{\text{emp}}; \theta) \leq \Gamma^*(\theta) - \underbrace{C_p\big(t^{-(1-\alpha)} + t^{-\alpha} + \sqrt{\tfrac{\log t}{t}}\big)}_{\text{oracle-tracking regret}}$$

$$- \underbrace{C_p\, t^{-(1-\gamma)/2} \sqrt{\log t}}_{\text{estimation}} - \underbrace{D_{\max}|\mathcal{P}|\big(\tfrac{|\mathcal{P}|-1}{t} + \tfrac{t^{-\gamma}}{1-\gamma}\big)}_{\text{mixing}}$$

$$- \underbrace{2D_{\max}\, t^{-3/4}}_{\text{burn-in}} \Bigg) \leq t^{-p}.$$

Proposition 6.3 links the information rate our algorithm achieves empirically to the optimal rate as a function of $t$. There are four terms contributing to finite time non-optimality which all vanish asymptotically. The first error

term is from the oracle tracking regret from using FTRL. The second and third terms represent deviations from optimal allocation arising from estimation error and mixing bias. Here, there is a trade-off controlled by $\gamma$. A larger $\gamma$ means faster decay of the mixing weight, thereby decreasing mixing bias; however, estimation bias increases. Ignoring polylog factors, $\gamma = 1/3$ maximizes the minimum decay rate of these two terms. Finally, the last term corresponds to the burn-in phase. In this phase, the estimated boundary may not be the correct boundary, but we can still bound the overall loss accumulated in this phase.

Next, we show that the procedure is $\delta$-correct.

**Proposition 6.4.** *Fix $\delta \in (0, 1)$. Run Algorithm 1 with stopping time $\tau_\delta$ defined in* (5). *Then the procedure is $\delta$-correct, i.e.*

(i) $\mathbb{P}_{\boldsymbol{\theta}}(\hat{S}_{\tau_\delta} \neq S^*(\boldsymbol{\theta})) \leq \delta$.

(ii) $\mathbb{P}_{\boldsymbol{\theta}}(\tau_\delta < \infty) = 1$.

For part (ii), we show that the threshold $\beta(t, \delta)$ grows at most logarithmically in $t$, whereas $\min_{(i,j) \in \hat{B}_t} Z_{ij}(t)$ eventually grows linearly in $t$. Consequently, almost surely there exists a finite time at which the stopping condition is met.

Finally, we show that the expected stopping time matches the information-theoretic lower bound.

**Theorem 6.5** (Asymptotic optimality in expectation)**.** *Fix $\boldsymbol{\theta} \in \Theta^{\mathrm{gap}}$ and let $\tau_\delta$ be the stopping time of Algorithm 1. Then*

$$\lim_{\delta \to 0} \frac{\mathbb{E}_{\boldsymbol{\theta}}[\tau_\delta]}{\log(1/\delta)} = \frac{1}{\Gamma^*(\boldsymbol{\theta})}.$$

On a high level this follows from Proposition 6.3: for large $t$ the algorithm collects information at a rate arbitrarily close to $\Gamma^*(\boldsymbol{\theta})$, so the stopping rule triggers after at most $\log(1/\delta)/\Gamma^*(\boldsymbol{\theta})$ samples (up to lower-order terms). On the other hand, Theorem 3.4 shows that no $\delta$-correct procedure can asymptotically do better in expectation.

# 7. Numerical Illustrations

We evaluate Algorithm 1 on three synthetic instances. For each, we vary the number of items $n$ (with $k = 5$ fixed) and vary $k$ (with $n = 100$ fixed) fixing $\delta = 0.01$.

Following Proposition 6.3, we set the mixing exponent to $\gamma = 1/3$. For the learning-rate schedule $\mu_t = t^{-\alpha}$, we set $\alpha = 0.2$. With stochastic gradients, the early gradient directions can be noisy[1], and if the decay rate is too fast, these will dominate the cumulative gradient scores $\boldsymbol{\Psi}_t^{(w)}, \boldsymbol{\Psi}_t^{(q)}$ for finite $t$. $\alpha$ can be tuned by running the primal–dual algorithm on the oracle problem (with $\boldsymbol{\theta}$ known) over a col-

lection of $\boldsymbol{\theta}$ instances, and selecting the value that converges both quickly and stably to $\boldsymbol{w}^*(\boldsymbol{\theta})$.

**Baselines.** We compare against the most competitive fixed-confidence baselines for $\delta$-correct top-$k$ identification from pairwise comparisons: (i) *SEEKS* and *SEEKS-v2* (Ren et al., 2020), (ii) *Active Ranking* from Heckel et al. (2019).

We consider three data-generating processes; in all experiments, the comparison feedback is binary, and Algorithm 1 is implemented using the Bradley–Terry likelihood.[2]

**Random utilities.** Draw $\tilde{\theta}_1, \ldots, \tilde{\theta}_n \overset{\text{i.i.d.}}{\sim} \mathrm{Unif}[-5, 5]$ and center by $\boldsymbol{\theta} := \tilde{\boldsymbol{\theta}} - \frac{1}{n}\left(\sum_{i=1}^n \tilde{\theta}_i\right)\mathbf{1}$. For computational purposes, we reject the draw unless $\theta_k - \theta_{k+1} \geq 0.02$. Data are generated under the Bradley–Terry model.

**Equally spaced utilities.** Fix gap $= 0.1$ and set $\theta_i = \frac{n+1-2i}{2} \cdot$ gap for $i \in [n]$. Comparisons are generated under the Bradley–Terry model as above.

**SST preference matrix.** To test the algorithm when the model is misspecified, we generate outcomes from a pairwise probability matrix $P$ that satisfies SST but is not induced by latent utilities. The construction closely follows the "independent bands" generative process of Shah et al. (2017). Fix $\Delta = 0.05$ and construct $P$ diagonal-by-diagonal. Set $p_{ii} = \frac{1}{2}$ and draw $p_{i,i+1} \sim \mathrm{Unif}[p_{ii}, p_{ii} + \Delta]$ for $i = 1, \ldots, n - 1$. Then, for each subsequent diagonal band, define the lower bound $L_{ij} := \max\{p_{i,j-1}, p_{i+1,j}\}$ and draw $p_{ij} \sim \mathrm{Unif}\big[L_{ij}, \min\{L_{ij} + \Delta, 1\}\big]$. This ensures $p_{ij} \geq \max\{p_{i\ell}, p_{\ell j}\}$ for all $i < \ell < j$, satisfying SST. Similar to the random utilities instance, we enforce $p_{k,k+1} \geq 0.51$ for computational reasons.

## 7.1. Results

Figure 1 presents the mean stopping time across 100 simulations on a log-log scale. All algorithms returned the correct top-$k$ set across each instance.

**Random utilities** (left column). Algorithm 1 outperforms the baselines as both $n$ and $k$ vary. As $n$ increases, all algorithms' stopping times increase because the instances become progressively harder. The range of $\boldsymbol{\theta}$ is fixed, so on average, there are boundary pairs with smaller utility gaps.

**Equally spaced** (center column). In this experiment, we include an oracle benchmark. The oracle computes $\boldsymbol{w}^*(\boldsymbol{\theta})$ offline, allocates online according to those proportions, and uses the stopping rule in section 5.5. We see that the stopping time of Algorithm 1 closely tracks the oracle, demonstrating that even at $\delta = .01$ our performance is close to the oracle performance. However, as $n$ grows, the stopping time

---

[1]See Appendix G.3 for further discussion on this and tuning $\alpha$.

[2]We are not aware of fixed-confidence top-$k$ identification baselines for cardinal pairwise feedback that would allow a direct comparison.

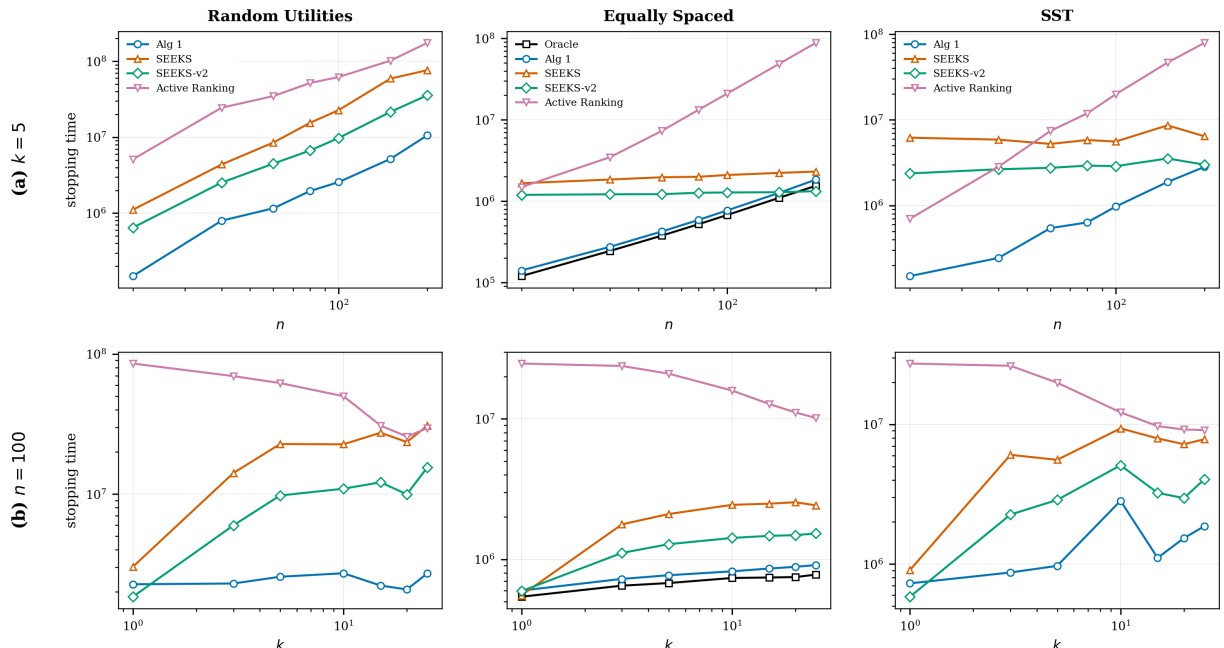

*Figure 1.* Mean stopping times over 100 simulations with $\delta = 0.01$. Columns correspond to the three instances; rows correspond to (a) varying $n$ with $k = 5$ fixed, and (b) varying $k$ with $n = 100$ fixed.

increases, but does not increase for SEEKS and SEEKS-v2. The primary driver is that our stopping threshold grows with $\|\hat{\boldsymbol{\theta}}_t\|_2^2$, which grows with $n$ in this equally spaced instance. Theorem 6.5 guarantees that for sufficiently small $\delta$ our expected stopping time will be smaller, and this is demonstrated in Figure 2. However as $\|\boldsymbol{\theta}\|_2^2$ grows the stopping threshold is only tight when $\delta$ is small, so $\log(\frac{1}{\delta})$ is a dominant term in $\beta(t, \delta)$.

**SST** (right column). Even under misspecification, Algorithm 1 remains competitive across $n$. We observe a similar pattern to the equally spaced instance. As $n$ grows, $\|\hat{\boldsymbol{\theta}}_t\|_2^2$ grows so the stopping threshold is loose at $\delta = .01$.

## 8. Conclusions

For top-$k$ identification from pairwise comparisons, we construct an online algorithm that is optimal as $\delta \to 0$. Reducing the alternative set to parameters that invert a boundary pair yields a structured saddle-point problem that naturally supports maintaining both primal and dual iterates. This lowers the per-round computational burden relative to approaches that rely on computing a best-response oracle each iteration. As a result, the method scales to moderately large $n$. In simulations, our procedure is competitive with existing baselines, and typically performs best when utilities are confined to a fixed range (so many boundary gaps are small when $n$ is large) and as $k$ grows.

The main weakness of the algorithm is the stopping thresh-

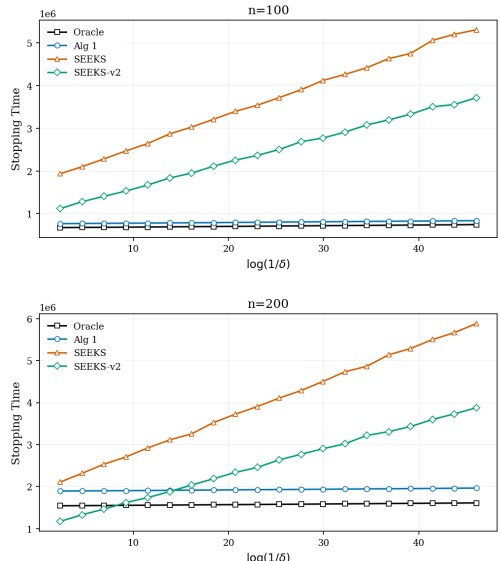

*Figure 2.* Mean stopping time over 100 simulations for the Equally Spaced instance with $k = 5$.

old at moderate $\delta$, which in some instances can even cause the oracle to be outperformed by existing baselines. Future work is required to investigate if a tighter stopping threshold can be constructed for moderate $\delta$ while maintaining optimality as $\delta \to 0$.

## Impact Statement

This paper presents work whose goal is to advance the field of machine learning. There are many potential societal consequences of our work, none of which we feel must be specifically highlighted here.

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

# Notation

This table lists notation from the main text that is used in the appendix without being redefined there. Additional notation introduced inside a proof is local to that proof.

| Notation | Definition |
|---|---|
| *Problem and comparison model* | |
| $n, k, [n]$ | number of items, identification set size, and $\{1, \dots, n\}$ |
| $\mathcal{P}$ | $\{(i,j) : 1 \le i < j \le n\}$ |
| $\boldsymbol{\theta}$ | true utility vector |
| $\Theta, \Theta_0, \Theta^{\mathrm{gap}}$ | parameter space, centered subspace, and positive top-$k$ gap subset |
| $R$ | bound of $\|\boldsymbol{\theta}\|_\infty \le R$ |
| $A_t, Y_t, \mathcal{F}_t$ | sampled pair, outcome, and observation filtration at time $t$ |
| $\boldsymbol{x}_{ij}, \eta_{ij}(\boldsymbol{\theta})$ | comparison direction and natural parameter, $\eta_{ij} = \theta_i - \theta_j$ |
| $A(\eta), T(y)$ | log-partition function and sufficient statistic |
| $\overline{\sigma}^2, \underline{a}, \sigma^2$ | global curvature upper bound, local curvature lower bound, and sub-Gaussian variance bound |
| $\ell_t(\boldsymbol{\theta})$ | log-likelihood after $t$ observations |
| | |
| *Top-k alternatives* | |
| $S^*(\boldsymbol{\theta})$ | true top-$k$ set |
| $B(\boldsymbol{\theta})$ | boundary pairs |
| $u_{ij}, v_{ij}$ | top-$k$ and non-top-$k$ endpoints of boundary pair $(i,j)$ |
| $\Theta_{ij}$ | set of alternatives that swap the ranking of $(i,j) \in B(\boldsymbol{\theta})$ |
| $\Theta_{ij}(t)$ | estimated version of $\Theta_{ij}$ formed from $\hat{\boldsymbol{\theta}}_t$ |
| | |
| *Oracle game* | |
| $\Delta_\mathcal{P}, \Delta_B$ | probability simplexes over all pairs and estimated boundary pairs |
| $d_{ij}(\boldsymbol{\theta}, \boldsymbol{\theta}')$ | $\mathrm{KL}(P_{\eta_{ij}(\boldsymbol{\theta})} \| P_{\eta_{ij}(\boldsymbol{\theta}')})$ |
| $D_{\boldsymbol{w}}(\boldsymbol{\theta} \| \boldsymbol{\theta}')$ | $\sum_{(a,b) \in \mathcal{P}} w_{ab}\, d_{ab}(\boldsymbol{\theta}, \boldsymbol{\theta}')$ |
| $\gamma_{ij}(\boldsymbol{w}; \boldsymbol{\theta})$ | $\inf_{\boldsymbol{\theta}' \in \Theta_{ij}} D_{\boldsymbol{w}}(\boldsymbol{\theta} \| \boldsymbol{\theta}')$ |
| $\boldsymbol{\theta}^*_{ij}(\boldsymbol{w}; \boldsymbol{\theta})$ | $\operatorname{argmin}_{\boldsymbol{\theta}' \in \Theta_{ij}} D_{\boldsymbol{w}}(\boldsymbol{\theta} \| \boldsymbol{\theta}')$ |
| $\Gamma(\boldsymbol{w}; \boldsymbol{\theta})$ | $\min_{(i,j) \in B(\boldsymbol{\theta})} \gamma_{ij}(\boldsymbol{w}; \boldsymbol{\theta})$ |
| $\Gamma^*(\boldsymbol{\theta})$ | $\sup_{\boldsymbol{w} \in \Delta_\mathcal{P}} \Gamma(\boldsymbol{w}; \boldsymbol{\theta})$ |
| $F(\boldsymbol{w}, \boldsymbol{q}; \boldsymbol{\theta})$ | $\sum_{(i,j) \in B(\boldsymbol{\theta})} q_{ij}\, \gamma_{ij}(\boldsymbol{w}; \boldsymbol{\theta})$ |
| $G(\boldsymbol{w})$ | support graph $([n], \{\{i, j\} : w_{ij} > 0\})$ |

| Notation | Definition |
|---|---|
| *Sequential algorithm* | |
| $\hat{\boldsymbol{\theta}}_t, \hat{S}_t, \hat{B}_t$ | MLE and its induced top-$k$ set and boundary set |
| $N_{ij}(t), N_{\min}(t)$ | empirical count for pair $(i,j)$ and the minimum pair count |
| $\hat{w}^{\mathrm{emp}}_{t,ij}$ | empirical allocation proportion $N_{ij}(t)/t$ |
| $\boldsymbol{w}_t, \boldsymbol{q}_t$ | primal and dual iterates |
| $I_t$ | boundary pair sampled for the stochastic-gradient update |
| $\boldsymbol{\theta}^*_t$ | $\operatorname{argmin}_{\boldsymbol{\theta}' \in \Theta_{I_t}(t)} D_{\boldsymbol{w}_t}(\hat{\boldsymbol{\theta}}_t \| \boldsymbol{\theta}')$ |
| $\gamma_t$ | $D_{\boldsymbol{w}_t}(\hat{\boldsymbol{\theta}}_t \| \boldsymbol{\theta}^*_t) = \gamma_{I_t}(\boldsymbol{w}_t; \hat{\boldsymbol{\theta}}_t)$ |
| $\mu_t, \rho_t$ | FTRL learning rate and C-tracking mixing weight |
| $\boldsymbol{u}$ | uniform distribution over $\mathcal{P}$ |
| $Z_{ij}(t), Z_{\hat{S}_t}(t)$ | boundary-pair GLR statistic and its minimum over $\hat{B}_t$ |

## A. Proofs for Section 4

### A.1. Proof of Lemma 4.1

*Proof.* For any $\boldsymbol{\theta}' \in \Theta_{ij}$, $\boldsymbol{w} \mapsto D_{\boldsymbol{w}}(\boldsymbol{\theta}\|\boldsymbol{\theta}')$ is linear. Taking the infimum over $\boldsymbol{\theta}' \in \Theta_{ij}$ is a pointwise infimum of linear functions, thus is concave.

For continuity: $(\boldsymbol{w}, \boldsymbol{\theta}') \mapsto D_{\boldsymbol{w}}(\boldsymbol{\theta}\|\boldsymbol{\theta}')$ is continuous and $\Theta_{ij}$ is compact, so $\gamma_{ij}(\boldsymbol{w}; \boldsymbol{\theta}) = \min_{\boldsymbol{\theta}' \in \Theta_{ij}} D_{\boldsymbol{w}}(\boldsymbol{\theta}\|\boldsymbol{\theta}')$ is continuous as a minimum of a continuous function over a compact set. $\square$

### A.2. Proof of Lemma 4.2

*Proof. (a)*

$$\nabla^2_{\boldsymbol{\theta}'} D_{\boldsymbol{w}}(\boldsymbol{\theta}\|\boldsymbol{\theta}') = \sum_{(a,b)\in\mathcal{P}} w_{ab}\, A''(\eta_{ab}(\boldsymbol{\theta}'))\, \boldsymbol{x}_{ab}\boldsymbol{x}_{ab}^\top.$$

So for any $\boldsymbol{v} \in \mathbb{R}^n$,

$$\begin{aligned} &\boldsymbol{v}^\top \nabla^2_{\boldsymbol{\theta}'} D_{\boldsymbol{w}}(\boldsymbol{\theta}\|\boldsymbol{\theta}')\boldsymbol{v} \\ &= \sum_{(a,b)\in\mathcal{P}} w_{ab}\, A''(\eta_{ab}(\boldsymbol{\theta}'))\, (v_a - v_b)^2 \geq 0, \end{aligned}$$

with equality iff $v_a = v_b$ on every edge in $G(\boldsymbol{w})$. Since $G(\boldsymbol{w})$ is connected this implies $\boldsymbol{v} = c\boldsymbol{1}$. Restricting to $\boldsymbol{v} \in \Theta_0$ forces $c = 0$, so the Hessian is positive definite on $\Theta_0$.

*(b)* This follows from (a) and the fact that $\Theta_{ij}$ is compact.

*(c)* Let $\boldsymbol{\theta}^* = \boldsymbol{\theta}^*_{ij}(\boldsymbol{w}; \boldsymbol{\theta})$ be the unique minimizer over $\Theta_{ij}$. If $\boldsymbol{\theta}^* \in \partial\Theta$ we are done. Otherwise $\boldsymbol{\theta}^* \in \text{int}(\Theta)$. If $\theta^*_{v_{ij}} > \theta^*_{u_{ij}}$, then there exists $\lambda \in (0,1)$ such that $\bar{\boldsymbol{\theta}} := (1-\lambda)\boldsymbol{\theta}^* + \lambda\boldsymbol{\theta}$ satisfies $\bar{\theta}_{v_{ij}} = \bar{\theta}_{u_{ij}}$. And $\bar{\boldsymbol{\theta}} \in \Theta$. Also $\bar{\boldsymbol{\theta}} \in \Theta_{ij}$ by construction. By convexity of $D_{\boldsymbol{w}}(\boldsymbol{\theta}\|\cdot)$ and $D_{\boldsymbol{w}}(\boldsymbol{\theta}\|\boldsymbol{\theta}) = 0$,

$$D_{\boldsymbol{w}}(\boldsymbol{\theta}\|\bar{\boldsymbol{\theta}}) \leq (1-\lambda)D_{\boldsymbol{w}}(\boldsymbol{\theta}\|\boldsymbol{\theta}^*) + \lambda D_{\boldsymbol{w}}(\boldsymbol{\theta}\|\boldsymbol{\theta}) < D_{\boldsymbol{w}}(\boldsymbol{\theta}\|\boldsymbol{\theta}^*),$$

contradiction. So if $\boldsymbol{\theta}^* \in \text{int}(\Theta)$ the halfspace must bind. $\square$

## B. Proof of Lemma 6.1

Define

$$\Xi_{ij}(t) := \sum_{s=1}^{t} \mathbf{1}_{\{A_s=(i,j)\}}\Big(T(Y_s) - A'(\eta_{ij}(\boldsymbol{\theta}))\Big),$$

*Proof.* Let $\boldsymbol{\Delta} := \hat{\boldsymbol{\theta}}_t - \boldsymbol{\theta} \in \Theta_0$ and for each $(i,j) \in \mathcal{P}$ set $z_{ij} := \boldsymbol{x}_{ij}^\top\boldsymbol{\Delta} = \Delta_i - \Delta_j$. By definition of $\ell_t$ and $\Xi_{ij}(t)$,

$$\begin{aligned} &\ell_t(\boldsymbol{\theta} + \boldsymbol{\Delta}) - \ell_t(\boldsymbol{\theta}) \\ &= \sum_{(i,j)\in\mathcal{P}} \Big(z_{ij}\,\Xi_{ij}(t) - N_{ij}(t)\, d_{ij}(\boldsymbol{\theta}, \boldsymbol{\theta}+\boldsymbol{\Delta})\Big). \end{aligned}$$

Since $\hat{\boldsymbol{\theta}}_t$ maximizes $\ell_t$ over the constraint set and $\boldsymbol{\theta}$ is feasible, $\ell_t(\boldsymbol{\theta} + \boldsymbol{\Delta}) - \ell_t(\boldsymbol{\theta}) \geq 0$.

By assumption $A'' \geq \underline{a}$ on $[-2R, 2R]$, so strong convexity gives $d_{ij}(\boldsymbol{\theta}, \boldsymbol{\theta} + \boldsymbol{\Delta}) = A(\eta_{ij}(\boldsymbol{\theta}) + z_{ij}) - A(\eta_{ij}(\boldsymbol{\theta})) - A'(\eta_{ij}(\boldsymbol{\theta}))\, z_{ij} \geq \frac{\underline{a}}{2} z_{ij}^2$, so

$$0 \leq \sum_{(i,j)\in\mathcal{P}} \Big(z_{ij}\,\Xi_{ij}(t) - \frac{\underline{a}}{2}N_{ij}(t)\,z_{ij}^2\Big). \tag{6}$$

Rearranging gives

$$\frac{\underline{a}}{2} \sum_{(i,j)\in\mathcal{P}} N_{ij}(t)\,z_{ij}^2 \leq \sum_{(i,j)\in\mathcal{P}} z_{ij}\,\Xi_{ij}(t). \tag{7}$$

Apply Cauchy–Schwarz:

$$\begin{aligned} &\sum_{(i,j)} z_{ij}\Xi_{ij}(t) \\ &= \sum_{(i,j)} \big(\sqrt{N_{ij}(t)}\,z_{ij}\big)\cdot\Big(\frac{\Xi_{ij}(t)}{\sqrt{N_{ij}(t)}}\Big) \\ &\leq \sqrt{\sum_{(i,j)} N_{ij}(t)\,z_{ij}^2}\,\sqrt{\sum_{(i,j)} \frac{\Xi_{ij}(t)^2}{N_{ij}(t)}}. \end{aligned}$$

Let

$$A := \sum_{(i,j)\in\mathcal{P}} N_{ij}(t)\,z_{ij}^2, \qquad B := \sum_{(i,j)\in\mathcal{P}} \frac{\Xi_{ij}(t)^2}{N_{ij}(t)}.$$

Then (7) and the above imply

$$\frac{\underline{a}}{2}A \leq \sqrt{A}\sqrt{B} \quad\Longrightarrow\quad A \leq \frac{4}{\underline{a}^2}B,$$

i.e.

$$\sum_{(i,j)\in\mathcal{P}} N_{ij}(t)\,(\boldsymbol{x}_{ij}^\top\boldsymbol{\Delta})^2 \leq \frac{4}{\underline{a}^2} \sum_{(i,j)\in\mathcal{P}} \frac{\Xi_{ij}(t)^2}{N_{ij}(t)}. \tag{8}$$

Define the weighted Laplacian

$$L_t := \sum_{(i,j)\in\mathcal{P}} N_{ij}(t)\,\boldsymbol{x}_{ij}\boldsymbol{x}_{ij}^\top,$$

so $\boldsymbol{\Delta}^\top L_t \boldsymbol{\Delta} = \sum_{(i,j)} N_{ij}(t)(\boldsymbol{x}_{ij}^\top\boldsymbol{\Delta})^2$. Since $N_{ij}(t) \geq N_{\min}(t)$ for all pairs,

$$L_t \succeq N_{\min}(t) \sum_{1 \leq i < j \leq n} \boldsymbol{x}_{ij}\boldsymbol{x}_{ij}^\top = N_{\min}(t)\,(nI_n - \boldsymbol{1}\boldsymbol{1}^\top).$$

$\boldsymbol{\Delta} \in \Theta_0$ implies $\boldsymbol{1}^\top\boldsymbol{\Delta} = 0$, so

$$\boldsymbol{\Delta}^\top L_t \boldsymbol{\Delta} \geq n\, N_{\min}(t)\, \|\boldsymbol{\Delta}\|_2^2.$$

Combining with (8) gives

$$\|\boldsymbol{\Delta}\|_2^2 \leq \frac{4}{\underline{a}^2\, n\, N_{\min}(t)} \sum_{(i,j)\in\mathcal{P}} \frac{\Xi_{ij}(t)^2}{N_{ij}(t)}. \tag{9}$$

It remains to bound $\sum_{(i,j)} \Xi_{ij}(t)^2/N_{ij}(t)$ with high probability. Fix a pair $(i,j) \in \mathcal{P}$ and define martingale differences

$$\xi_s^{ij} := \mathbf{1}_{\{A_s=(i,j)\}}\Big(T(Y_s) - A'(\eta_{ij}(\boldsymbol{\theta}))\Big), \qquad s \geq 1,$$

so that $\Xi_{ij}(t) = \sum_{s=1}^t \xi_s^{ij}$ and $N_{ij}(t) = \sum_{s=1}^t \mathbf{1}_{\{A_s=(i,j)\}}$. By Assumption 2, for all $\lambda \in \mathbb{R}$,

$$\mathbb{E}_{\boldsymbol{\theta}}\big[\exp\big(\lambda \xi_s^{ij}\big) \mid \mathcal{F}_{s-1}\big] \leq \exp\Big(\frac{\sigma^2 \lambda^2}{2}\mathbf{1}_{\{A_s=(i,j)\}}\Big).$$

Therefore for each $\lambda$ the process

$$M_s(\lambda) := \exp\Big(\lambda \Xi_{ij}(s) - \frac{\sigma^2 \lambda^2}{2}N_{ij}(s)\Big)$$

is a nonnegative supermartingale, so $\mathbb{E}_{\boldsymbol{\theta}}[M_t(\lambda)] \leq 1$. Fix $m \in \{1, \ldots, t\}$ and $u > 0$, and set $\lambda := \sqrt{2u}/(\sigma\sqrt{m})$. On the event $\{N_{ij}(t) = m\}$ we have

$$\Big\{\Xi_{ij}(t) \geq \sigma\sqrt{2mu}\Big\} \subseteq \Big\{M_t(\lambda) \geq e^u\Big\},$$

since $\lambda\sigma\sqrt{2mu} = 2u$ and $(\sigma^2\lambda^2/2)m = u$. Thus by Markov's inequality,

$$\mathbb{P}_{\boldsymbol{\theta}}\Big(\Xi_{ij}(t) \geq \sigma\sqrt{2mu},\, N_{ij}(t) = m\Big) \leq e^{-u}.$$

Applying the same argument to $-\Xi_{ij}(t)$ gives

$$\mathbb{P}_{\boldsymbol{\theta}}\Big(|\Xi_{ij}(t)| \geq \sigma\sqrt{2mu},\, N_{ij}(t) = m\Big) \leq 2e^{-u}.$$

Sum over $m = 1, \ldots, t$ to get

$$\mathbb{P}_{\boldsymbol{\theta}}\Big(\frac{\Xi_{ij}(t)^2}{N_{ij}(t)} \geq 2\sigma^2 u,\, N_{ij}(t) \geq 1\Big) \leq 2t\, e^{-u}.$$

Choose $u = \log\big(\frac{2t|\mathcal{P}|}{\delta}\big)$ and apply a union bound over $(i,j) \in \mathcal{P}$: with probability at least $1 - \delta$,

$$\max_{(i,j)\in\mathcal{P}} \frac{\Xi_{ij}(t)^2}{N_{ij}(t)} \leq 2\sigma^2 \log\Big(\frac{2t|\mathcal{P}|}{\delta}\Big),$$

hence

$$\sum_{(i,j)\in\mathcal{P}} \frac{\Xi_{ij}(t)^2}{N_{ij}(t)} \leq 2\sigma^2 |\mathcal{P}| \log\Big(\frac{2t|\mathcal{P}|}{\delta}\Big).$$

Plugging into (9) gives

$$\|\hat{\boldsymbol{\theta}}_t - \boldsymbol{\theta}\|_2^2 \leq \frac{8\sigma^2|\mathcal{P}|}{\underline{a}^2\, n\, N_{\min}(t)} \log\Big(\frac{2t|\mathcal{P}|}{\delta}\Big).$$

$|\mathcal{P}| = \frac{n(n-1)}{2} \leq \frac{n^2}{2}$ so

$$\|\hat{\boldsymbol{\theta}}_t - \boldsymbol{\theta}\|_2 \leq \frac{2\sigma}{\underline{a}} \sqrt{\frac{n\log\big(\frac{2t|\mathcal{P}|}{\delta}\big)}{N_{\min}(t)}},$$

$\square$

## B.1. Proof of Corollary 6.2

*Proof.* Apply Lemma 6.1 with $\delta_t = t^{-2}$. Since $\sum_{t\geq 1}\delta_t < \infty$, by Borel–Cantelli almost surely there exists $t_1 < \infty$ such that for all $t \geq t_1$,

$$\|\hat{\boldsymbol{\theta}}_t - \boldsymbol{\theta}\|_2 \leq \frac{2\sigma}{\underline{a}} \sqrt{\frac{n\log\big(\frac{2t|\mathcal{P}|}{\delta_t}\big)}{N_{\min}(t)}}.$$

By Lemma F.1(e), $N_{\min}(t)/\log t \to \infty$ almost surely, while $\log\big(\frac{2t|\mathcal{P}|}{\delta_t}\big) = \log(2t|\mathcal{P}|t^2) = O(\log t)$. Hence the right-hand side tends to 0 almost surely, so $\|\hat{\boldsymbol{\theta}}_t - \boldsymbol{\theta}\|_2 \to 0$ almost surely.

Since $\boldsymbol{\theta} \in \Theta^{\mathrm{gap}}$, define $\Delta_k := \theta_{(k)} - \theta_{(k+1)} > 0$. Because $\|\hat{\boldsymbol{\theta}}_t - \boldsymbol{\theta}\|_\infty \leq \|\hat{\boldsymbol{\theta}}_t - \boldsymbol{\theta}\|_2$, almost surely there exists $t_{\mathrm{stab}} < \infty$ such that $\|\hat{\boldsymbol{\theta}}_t - \boldsymbol{\theta}\|_\infty \leq \Delta_k/4$ for all $t \geq t_{\mathrm{stab}}$. For such $t$, if $u \in S^*(\boldsymbol{\theta})$ and $v \notin S^*(\boldsymbol{\theta})$ then $\theta_u - \theta_v \geq \Delta_k$, and thus

$$\hat{\theta}_{t,u} - \hat{\theta}_{t,v} \geq (\theta_u - \theta_v) - 2\|\hat{\boldsymbol{\theta}}_t - \boldsymbol{\theta}\|_\infty \geq \Delta_k - \frac{\Delta_k}{2} > 0.$$

Therefore $\hat{S}_t = S^*(\boldsymbol{\theta})$ and $\hat{B}_t = B(\boldsymbol{\theta})$ for all $t \geq t_{\mathrm{stab}}$ $\square$

## C. Proof of Proposition 6.3

**Outline.** The proof decomposes into (i) online learning error, (ii) statistical error from estimating $\boldsymbol{\theta}$, and (iii) C-tracking/mixing bias. First, Lemma C.1 establishes that the stochastic gradients are unbiased estimators of the gradients and are uniformly bounded, which is the bounded-noise condition needed for exponential-weights/FTRL analysis. Next, Lemma C.2 gives high probability regret bounds for both the primal and dual players. Lemma C.3 bounds the duality gap by averaged regret. Finally, we combine these bounds. We introduce a burn-in index $b_t := \lceil t^{1/4} \rceil$ so that, with high probability, the set of boundary pairs is correct for every time $s \in \{b_t, \ldots, t\}$.

Some supporting lemmas are applications of standard online learning regret bounds and online saddle point games; we include full proofs for completeness.

Fix $\boldsymbol{\theta} \in \Theta^{\mathrm{gap}}$ and define $B := B(\boldsymbol{\theta})$ and $m := |B| = k(n-k)$. Let $D_{\max}$ and $L$ be as in Lemma F.3, and define $G := mD_{\max}$. For each horizon $T \geq 1$ set $b_T := \lceil T^{1/4} \rceil$.

**Lemma C.1.** *Take $t \geq 1$ and $I_t \sim \mathrm{Unif}(B)$ is sampled after $(\hat{\boldsymbol{\theta}}_t, \boldsymbol{w}_t, \boldsymbol{q}_t)$ are formed. Recall the stochastic gradients from (3):*

$$\hat{g}_{t,ab}^{(w)} := m\, q_{t,I_t}\, d_{ab}(\hat{\boldsymbol{\theta}}_t, \boldsymbol{\theta}_t^*),\ (a,b) \in \mathcal{P},$$
$$\hat{g}_{t,ij}^{(q)} := m\, \gamma_t\, \mathbf{1}\{(i,j) = I_t\},\ (i,j) \in B.$$

*Then,*

$$\mathbb{E}\left[\hat{g}_{t,ab}^{(w)} \mid \hat{\boldsymbol{\theta}}_t, \boldsymbol{w}_t, \boldsymbol{q}_t\right] = \frac{\partial F}{\partial w_{ab}}(\boldsymbol{w}_t, \boldsymbol{q}_t; \hat{\boldsymbol{\theta}}_t),$$

$$\mathbb{E}\left[\hat{g}_{t,ij}^{(q)} \mid \hat{\boldsymbol{\theta}}_t, \boldsymbol{w}_t, \boldsymbol{q}_t\right] = \frac{\partial F}{\partial q_{ij}}(\boldsymbol{w}_t, \boldsymbol{q}_t; \hat{\boldsymbol{\theta}}_t),$$

*additionally, $\|\hat{\boldsymbol{g}}_t^{(w)}\|_\infty \le mD_{\max}$ and $\|\hat{\boldsymbol{g}}_t^{(q)}\|_\infty \le mD_{\max}$ almost surely. If $G(\boldsymbol{w}_t)$ is connected (so the KL projections are unique), the right-hand sides above are gradients; otherwise, they are supergradients.*

*Proof.* Choose $(a, b) \in \mathcal{P}$. Each boundary pair has probability $1/m$ of being selected. It follows that

$$\mathbb{E}\left[\hat{g}_{t,ab}^{(w)} \mid \hat{\boldsymbol{\theta}}_t, \boldsymbol{w}_t, \boldsymbol{q}_t\right] = \sum_{(i,j)\in B} q_{t,ij}\, d_{ab}(\hat{\boldsymbol{\theta}}_t, \boldsymbol{\theta}_{ij}^*(\boldsymbol{w}_t; \hat{\boldsymbol{\theta}}_t)),$$

Similarly, for each $(i, j) \in B$,

$$\mathbb{E}\left[\hat{g}_{t,ij}^{(q)} \mid \hat{\boldsymbol{\theta}}_t, \boldsymbol{w}_t, \boldsymbol{q}_t\right]$$
$$= \sum_{(p,\ell)\in B} \frac{1}{m}\, m\, \gamma_{p\ell}(\boldsymbol{w}_t; \hat{\boldsymbol{\theta}}_t)\, \mathbf{1}\{(p,\ell) = (i,j)\}$$
$$= \gamma_{ij}(\boldsymbol{w}_t; \hat{\boldsymbol{\theta}}_t) = \frac{\partial F}{\partial q_{ij}}(\boldsymbol{w}_t, \boldsymbol{q}_t; \hat{\boldsymbol{\theta}}_t).$$

Finally, $0 \le q_{t,I_t} \le 1$, $0 \le d_{ab}(\cdot, \cdot) \le D_{\max}$, and $0 \le \gamma_t \le D_{\max}$, so $|\hat{g}_{t,ab}^{(w)}| \le mD_{\max}$ and $|\hat{g}_{t,ij}^{(q)}| \le mD_{\max}$. $\square$

**Lemma C.2.** *Fix $T \ge 1$, Define*

$$\mathrm{Reg}_w(T) := \sup_{\boldsymbol{w}\in\Delta_{\mathcal{P}}} \sum_{t=b_T}^{T} \left( F_{\hat{\boldsymbol{\theta}}_t}(\boldsymbol{w}, \boldsymbol{q}_t) - F_{\hat{\boldsymbol{\theta}}_t}(\boldsymbol{w}_t, \boldsymbol{q}_t) \right),$$

$$\mathrm{Reg}_q(T) := \sup_{\boldsymbol{q}\in\Delta_B} \sum_{t=b_T}^{T} \left( F_{\hat{\boldsymbol{\theta}}_t}(\boldsymbol{w}_t, \boldsymbol{q}_t) - F_{\hat{\boldsymbol{\theta}}_t}(\boldsymbol{w}_t, \boldsymbol{q}) \right).$$

*Then for every $\delta \in (0, 1)$, with probability at least $1 - \delta$,*

$$\mathrm{Reg}_w(T) \le \frac{D_{\boldsymbol{w}}(1)}{\mu_T} + \frac{G^2}{2}\sum_{t=1}^{T}\mu_t + 2G(b_T - 1)$$
$$+ 4G\sqrt{2(T - b_T + 1)\log\left(\frac{8|\mathcal{P}|}{\delta}\right)}, \quad (10)$$

$$\mathrm{Reg}_q(T) \le \frac{D_{\boldsymbol{q}}(1)}{\mu_T} + \frac{G^2}{2}\sum_{t=1}^{T}\mu_t + 2G(b_T - 1)$$
$$+ 4G\sqrt{2(T - b_T + 1)\log\left(\frac{8m}{\delta}\right)}. \quad (11)$$

*where*

$$D_{\boldsymbol{w}}(1) := \sup_{\boldsymbol{w}\in\Delta_{\mathcal{P}}} \mathrm{KL}(\boldsymbol{w}\|\boldsymbol{w}_1) = \max_{(a,b)\in\mathcal{P}} \log\frac{1}{w_{1,ab}},$$

$$D_{\boldsymbol{q}}(1) := \sup_{\boldsymbol{q}\in\Delta_B} \mathrm{KL}(\boldsymbol{q}\|\boldsymbol{q}_1) = \max_{(i,j)\in B} \log\frac{1}{q_{1,ij}}.$$

*Proof.* We prove (10). For each $t \in \{b_T, \ldots, T\}$ define $\boldsymbol{g}_t^{(w)} := \nabla_{\boldsymbol{w}} F_{\hat{\boldsymbol{\theta}}_t}(\boldsymbol{w}_t, \boldsymbol{q}_t)$. So for all $\boldsymbol{w} \in \Delta_{\mathcal{P}}$,

$$F_{\hat{\boldsymbol{\theta}}_t}(\boldsymbol{w}, \boldsymbol{q}_t) - F_{\hat{\boldsymbol{\theta}}_t}(\boldsymbol{w}_t, \boldsymbol{q}_t) \le \langle \boldsymbol{g}_t^{(w)}, \boldsymbol{w} - \boldsymbol{w}_t \rangle.$$

Summing and taking the supremum over $\boldsymbol{w}$ gives

$$\mathrm{Reg}_w(T) \le \sup_{\boldsymbol{w}\in\Delta_{\mathcal{P}}} \sum_{t=b_T}^{T} \langle \boldsymbol{g}_t^{(w)}, \boldsymbol{w} - \boldsymbol{w}_t \rangle.$$

Now decompose $\boldsymbol{g}_t^{(w)} = \hat{\boldsymbol{g}}_t^{(w)} + \boldsymbol{\Delta}_t^{(g)}$, where $\boldsymbol{\Delta}_t^{(g)} := \boldsymbol{g}_t^{(w)} - \hat{\boldsymbol{g}}_t^{(w)}$. Then

$$\sup_{\boldsymbol{w}} \sum_{t=b_T}^{T} \langle \boldsymbol{g}_t^{(w)}, \boldsymbol{w} - \boldsymbol{w}_t \rangle$$
$$\le \underbrace{\sup_{\boldsymbol{w}} \sum_{t=b_T}^{T} \langle \hat{\boldsymbol{g}}_t^{(w)}, \boldsymbol{w} - \boldsymbol{w}_t \rangle}_{(\mathrm{I})} + \underbrace{\sup_{\boldsymbol{w}} \sum_{t=b_T}^{T} \langle \boldsymbol{\Delta}_t^{(g)}, \boldsymbol{w} - \boldsymbol{w}_t \rangle}_{(\mathrm{II})}.$$

We can bound term (I) by applying Lemma F.4 to $\hat{\boldsymbol{g}}_t = \hat{\boldsymbol{g}}_t^{(w)}$ to get

$$(\mathrm{I}) \le \frac{D_{\boldsymbol{w}}(1)}{\mu_T} + \frac{G^2}{2}\sum_{t=1}^{T}\mu_t + 2G(b_T - 1).$$

To bound term (II) we split it into two parts. Let $\mathcal{H}_t$ denote the algorithm history at round $t$ after observing $Y_t$ and forming $(\hat{\boldsymbol{\theta}}_t, \boldsymbol{w}_t, \boldsymbol{q}_t)$, but before sampling $I_t$; and let $\mathcal{G}_t$ denote the history up to time $t$ including the draw $I_t$. By Lemma C.1, for each coordinate $(a, b)$,

$$\mathbb{E}[\hat{g}_{t,ab}^{(w)} \mid \mathcal{H}_t] = g_{t,ab}^{(w)},$$

so $\mathbb{E}[\Delta_{t,ab}^{(g)} \mid \mathcal{H}_t] = 0$. Since $\mathcal{G}_{t-1} \subseteq \mathcal{H}_t$, we also have

$$\mathbb{E}[\Delta_{t,ab}^{(g)} \mid \mathcal{G}_{t-1}] = 0,$$

and therefore $(\Delta_{t,ab}^{(g)})_{t\ge b_T}$ is a martingale difference sequence with respect to $(\mathcal{G}_t)$. Also,

$$|\Delta_{t,ab}^{(g)}| \le |g_{t,ab}^{(w)}| + |\hat{g}_{t,ab}^{(w)}| \le D_{\max} + G \le 2G.$$

Thus each increment is bounded in $[-2G, 2G]$. For any $\boldsymbol{w} \in \Delta_{\mathcal{P}}$,

$$\sum_{t=b_T}^{T} \langle \boldsymbol{\Delta}_t^{(g)}, \boldsymbol{w} - \boldsymbol{w}_t \rangle$$
$$= \left\langle \sum_{t=b_T}^{T} \boldsymbol{\Delta}_t^{(g)}, \boldsymbol{w} \right\rangle - \sum_{t=b_T}^{T} \langle \boldsymbol{\Delta}_t^{(g)}, \boldsymbol{w}_t \rangle$$
$$\le \underbrace{\max_{(a,b)\in\mathcal{P}} \sum_{t=b_T}^{T} \Delta_{t,ab}^{(g)}}_{(\mathrm{IIa})} + \underbrace{\left| \sum_{t=b_T}^{T} \langle \boldsymbol{\Delta}_t^{(g)}, \boldsymbol{w}_t \rangle \right|}_{(\mathrm{IIb})}.$$

*(IIa) Bounding* $\max_{(a,b)} \sum \Delta_{t,ab}^{(g)}$. Fix a coordinate $(a,b)$ and define the martingale

$$M_s^{ab} := \sum_{t=b_T}^{s} \Delta_{t,ab}^{(g)}, \qquad s = b_T, \ldots, T.$$

It has bounded increments $|\Delta_{t,ab}^{(g)}| \leq 2G$ and mean-zero conditional increments. By Azuma–Hoeffding, for any $x > 0$,

$$\mathbb{P}(M_T^{ab} \geq x) \leq \exp\left(-\frac{x^2}{8G^2(T - b_T + 1)}\right).$$

Choose

$$x := 2G\sqrt{2(T - b_T + 1)\log\left(\frac{4|\mathcal{P}|}{\delta}\right)}.$$

Then $\mathbb{P}(M_T^{ab} \geq x) \leq \delta/(4|\mathcal{P}|)$. A union bound over $(a,b) \in \mathcal{P}$ gives

$$\mathbb{P}\bigg(\max_{(a,b)\in\mathcal{P}} \sum_{t=b_T}^{T} \Delta_{t,ab}^{(g)}$$
$$\geq 2G\sqrt{2(T - b_T + 1)\log\left(\frac{4|\mathcal{P}|}{\delta}\right)}\bigg) \leq \frac{\delta}{4}.$$

*(IIb) Bounding* $\left|\sum\langle \boldsymbol{\Delta}_t^{(g)}, \boldsymbol{w}_t\rangle\right|$. Define the martingale

$$M_s := \sum_{t=b_T}^{s} \langle \boldsymbol{\Delta}_t^{(g)}, \boldsymbol{w}_t\rangle, \qquad s = b_T, \ldots, T,$$

We have,

$$|\langle \boldsymbol{\Delta}_t^{(g)}, \boldsymbol{w}_t\rangle| \leq \|\boldsymbol{\Delta}_t^{(g)}\|_\infty \|\boldsymbol{w}_t\|_1 \leq 2G.$$

By Azuma–Hoeffding, for any $y > 0$,

$$\mathbb{P}(|M_T| \geq y) \leq 2\exp\left(-\frac{y^2}{8G^2(T - b_T + 1)}\right).$$

Choose

$$y := 2G\sqrt{2(T - b_T + 1)\log\left(\frac{8}{\delta}\right)}.$$

Then $\mathbb{P}(|M_T| \geq y) \leq \delta/4$. Combining (IIa) and (IIb) and using that $\log(8/\delta) \leq \log(8|\mathcal{P}|/\delta)$ we get that with probability at least $1 - \delta/2$,

$$\sup_{\boldsymbol{w}\in\Delta_\mathcal{P}} \sum_{t=b_T}^{T} \langle \boldsymbol{\Delta}_t^{(g)}, \boldsymbol{w} - \boldsymbol{w}_t\rangle \leq 4G\sqrt{2(T - b_T + 1)\log\left(\frac{8|\mathcal{P}|}{\delta}\right)}.$$

We combine the bounds of terms I, IIa, and IIb to get that with probability at least $1 - \delta/2$

$$\mathrm{Reg}_w(T) \leq \frac{D_{\boldsymbol{w}}(1)}{\mu_T} + \frac{G^2}{2}\sum_{t=1}^{T} \mu_t + 2G(b_T - 1)$$
$$+ 4G\sqrt{2(T - b_T + 1)\log\left(\frac{8|\mathcal{P}|}{\delta}\right)}.$$

This is (10). (11) follows by the similar argument applied to $\boldsymbol{q}$. The $\boldsymbol{q}$-player is minimizing, and the update $\boldsymbol{r}_{t+1} = \exp(-\mu_{t+1}\boldsymbol{\Psi}_t^{(q)})$ corresponds to exponential weights on losses $\hat{\boldsymbol{g}}_t^{(q)}$ (equivalently, exponential weights on gains $-\hat{\boldsymbol{g}}_t^{(q)}$). Applying Lemma F.4 on $\Delta_B$ (with $D_{\boldsymbol{q}}(1)$ in place of $D_{\boldsymbol{w}}(1)$) and the same martingale argument gives (11) with probability at least $1 - \delta/2$. By a union bound, both inequalities (10)–(11) hold simultaneously with probability at least $1 - \delta$. $\qquad\square$

**Lemma C.3.** *Let $T \geq 1$ and define $T' := T - b_T + 1$. For any sequence $(\boldsymbol{w}_t, \boldsymbol{q}_t)_{t=b_T}^{T}$, define the averages*

$$\bar{\boldsymbol{w}}_{b_T:T} := \frac{1}{T'}\sum_{t=b_T}^{T} \boldsymbol{w}_t, \qquad \bar{\boldsymbol{q}}_{b_T:T} := \frac{1}{T'}\sum_{t=b_T}^{T} \boldsymbol{q}_t.$$

*Define the duality gap for $F_{\boldsymbol{\theta}}$:*

$$\mathrm{gap}_T^{\boldsymbol{\theta}} := \sup_{\boldsymbol{w}\in\Delta_\mathcal{P}} F_{\boldsymbol{\theta}}(\boldsymbol{w}, \bar{\boldsymbol{q}}_{b_T:T}) - \inf_{\boldsymbol{q}\in\Delta_B} F_{\boldsymbol{\theta}}(\bar{\boldsymbol{w}}_{b_T:T}, \boldsymbol{q}).$$

*Then*

$$\mathrm{gap}_T^{\boldsymbol{\theta}} \leq \frac{\mathrm{Reg}_w^{\boldsymbol{\theta}}(T) + \mathrm{Reg}_q^{\boldsymbol{\theta}}(T)}{T'},$$

*where*

$$\mathrm{Reg}_w^{\boldsymbol{\theta}}(T) := \sup_{\boldsymbol{w}\in\Delta_\mathcal{P}} \sum_{t=b_T}^{T} \big(F_{\boldsymbol{\theta}}(\boldsymbol{w}, \boldsymbol{q}_t) - F_{\boldsymbol{\theta}}(\boldsymbol{w}_t, \boldsymbol{q}_t)\big),$$
$$\mathrm{Reg}_q^{\boldsymbol{\theta}}(T) := \sup_{\boldsymbol{q}\in\Delta_B} \sum_{t=b_T}^{T} \big(F_{\boldsymbol{\theta}}(\boldsymbol{w}_t, \boldsymbol{q}_t) - F_{\boldsymbol{\theta}}(\boldsymbol{w}_t, \boldsymbol{q})\big).$$

*Proof.* By linearity in $\boldsymbol{q}$,

$$F_{\boldsymbol{\theta}}(\boldsymbol{w}, \bar{\boldsymbol{q}}_{b_T:T}) = \frac{1}{T'}\sum_{t=b_T}^{T} F_{\boldsymbol{\theta}}(\boldsymbol{w}, \boldsymbol{q}_t)$$

so

$$\sup_{\boldsymbol{w}} F_{\boldsymbol{\theta}}(\boldsymbol{w}, \bar{\boldsymbol{q}}_{b_T:T}) = \sup_{\boldsymbol{w}} \frac{1}{T'}\sum_{t=b_T}^{T} F_{\boldsymbol{\theta}}(\boldsymbol{w}, \boldsymbol{q}_t).$$

Additionally, for any fixed $\boldsymbol{q}$, concavity of $\boldsymbol{w} \mapsto F_{\boldsymbol{\theta}}(\boldsymbol{w}, \boldsymbol{q})$ and Jensen's inequality give

$$F_{\boldsymbol{\theta}}(\bar{\boldsymbol{w}}_{b_T:T}, \boldsymbol{q}) \geq \frac{1}{T'}\sum_{t=b_T}^{T} F_{\boldsymbol{\theta}}(\boldsymbol{w}_t, \boldsymbol{q}),$$

so

$$\inf_{\boldsymbol{q}} F_{\boldsymbol{\theta}}(\bar{\boldsymbol{w}}_{b_T:T}, \boldsymbol{q}) \geq \inf_{\boldsymbol{q}} \frac{1}{T'}\sum_{t=b_T}^{T} F_{\boldsymbol{\theta}}(\boldsymbol{w}_t, \boldsymbol{q}).$$

Thus,

$$
\begin{aligned}
\mathrm{gap}_T^{\boldsymbol{\theta}} &\leq \sup_{\boldsymbol{w}} \frac{1}{T'} \sum_{t=b_T}^{T} F_{\boldsymbol{\theta}}(\boldsymbol{w}, \boldsymbol{q}_t) - \inf_{\boldsymbol{q}} \frac{1}{T'} \sum_{t=b_T}^{T} F_{\boldsymbol{\theta}}(\boldsymbol{w}_t, \boldsymbol{q}) \\
&\leq \frac{1}{T'} \left( \sup_{\boldsymbol{w}} \sum_{t=b_T}^{T} \big( F_{\boldsymbol{\theta}}(\boldsymbol{w}, \boldsymbol{q}_t) - F_{\boldsymbol{\theta}}(\boldsymbol{w}_t, \boldsymbol{q}_t) \big) \right) \\
&\quad + \frac{1}{T'} \left( \sup_{\boldsymbol{q}} \sum_{t=b_T}^{T} \big( F_{\boldsymbol{\theta}}(\boldsymbol{w}_t, \boldsymbol{q}_t) - F_{\boldsymbol{\theta}}(\boldsymbol{w}_t, \boldsymbol{q}) \big) \right) \\
&= \frac{\mathrm{Reg}_w^{\boldsymbol{\theta}}(T) + \mathrm{Reg}_q^{\boldsymbol{\theta}}(T)}{T'}.
\end{aligned}
$$

$\square$

We now prove Proposition 6.3.

### C.1. Proof of Proposition 6.3

*Proof.* Fix $p > 1$ and define

$$
b_t := \left\lceil t^{1/4} \right\rceil, \quad T' := t - b_t + 1, \quad \Delta_k := \theta_{(k)} - \theta_{(k+1)} > 0.
$$

For each $s \in \{b_t, \dots, t\}$ define the event

$$
\mathcal{M}_{t,s} := \left\{ \|\hat{\boldsymbol{\theta}}_s - \boldsymbol{\theta}\|_2 \leq \frac{2\sigma}{\underline{a}} \sqrt{\frac{n \log\big(\frac{2s|\mathcal{P}|}{\delta_{t,s}}\big)}{N_{\min}(s)}} \right\}.
$$

where $\delta_{t,s} = t^{-(p+2)}$. Define the joint event

$$
\mathcal{M}_t := \bigcap_{s=b_t}^{t} \mathcal{M}_{t,s}.
$$

By Lemma 6.1, for each fixed $s$, $\mathbb{P}_{\boldsymbol{\theta}}(\mathcal{M}_{t,s}^c) \leq \delta_{t,s} = t^{-(p+2)}$. Therefore, by a union bound over the $T' = t - b_t + 1$ indices $s = b_t, \dots, t$,

$$
\mathbb{P}_{\boldsymbol{\theta}}(\mathcal{M}_t^c) \leq \sum_{s=b_t}^{t} \mathbb{P}_{\boldsymbol{\theta}}(\mathcal{M}_{t,s}^c) \leq (t - b_t + 1) t^{-(p+2)} \leq t^{-(p+1)}.
$$
(12)

Note that for all $s \leq t$,

$$
\log\Big(\frac{2s|\mathcal{P}|}{\delta_{t,s}}\Big) = \log\big(2s|\mathcal{P}| t^{p+2}\big) \leq \log\big(2|\mathcal{P}| t^{p+3}\big).
$$

Also, by Lemma F.1(e), for each $s$,

$$
N_{\min}(s) \geq \frac{1}{|\mathcal{P}|} \cdot \frac{(s+1)^{1-\gamma} - 1}{1 - \gamma} - (|\mathcal{P}| - 1).
$$

Define

$$
s_\gamma := \left\lceil \max\Big\{ 2^{1/(1-\gamma)}, \ \big(4|\mathcal{P}|(1-\gamma)(|\mathcal{P}|-1)\big)^{1/(1-\gamma)} \Big\} \right\rceil
$$

and $c_\gamma := \frac{1}{4|\mathcal{P}|(1-\gamma)}$. For all $s \geq s_\gamma$ we have $N_{\min}(s) \geq c_\gamma s^{1-\gamma}$ (by Fact F.2). Now define the time

$$
t_p := \min \Bigg\{ t \geq \max\{2, s_\gamma^4\} : \\
\frac{2\sigma}{\underline{a}} \sqrt{\frac{n \log\big(2|\mathcal{P}| t^{p+3}\big)}{c_\gamma b_t^{1-\gamma}}} \leq \frac{\Delta_k}{4} \Bigg\}.
$$

Fix any $t \geq t_p$. Then $b_t \geq s_\gamma$, so for every $s \in \{b_t, \dots, t\}$ we have $N_{\min}(s) \geq c_\gamma s^{1-\gamma} \geq c_\gamma b_t^{1-\gamma}$. On $\mathcal{M}_t$, for each such $s$,

$$
\|\hat{\boldsymbol{\theta}}_s - \boldsymbol{\theta}\|_2 \leq \frac{2\sigma}{\underline{a}} \sqrt{\frac{n \log\big(2|\mathcal{P}| t^{p+3}\big)}{c_\gamma b_t^{1-\gamma}}} \leq \frac{\Delta_k}{4},
$$

Thus $\|\hat{\boldsymbol{\theta}}_s - \boldsymbol{\theta}\|_\infty \leq \Delta_k/4$ for all $s \in \{b_t, \dots, t\}$, and by the same argument in Corollary 6.2

$$
\hat{S}_s = S^*(\boldsymbol{\theta}) \quad \text{and} \quad \hat{B}_s = B(\boldsymbol{\theta}) \qquad \forall s = b_t, \dots, t.
$$

Now we bound the failure probability of the regret bounds derived in Lemma C.2. Let $\delta_t := t^{-(p+2)}$ and $\mathcal{R}_t := \{\text{both bounds (10)–(11) hold at horizon } T = t\}$. By Lemma C.2 with confidence $\delta_t$,

$$
\mathbb{P}_{\boldsymbol{\theta}}(\mathcal{R}_t^c) \leq \delta_t = t^{-(p+2)}.
$$
(13)

Next, we bound $\mathrm{Reg}_w(t)$ and $\mathrm{Reg}_q(t)$ from Lemma C.2 by powers of $t$. We have $b_t/t \leq t^{-3/4}$. Also $\mu_s = s^{-\alpha}$ so Fact F.2 gives $\sum_{s=1}^{t} \mu_s \leq 1 + \frac{t^{1-\alpha}-1}{1-\alpha}$. Also $T' = t - b_t + 1 \geq t/2$ for all $t \geq 2$. Therefore, on $\mathcal{R}_t$,

$$
\begin{aligned}
\frac{\mathrm{Reg}_w(t) + \mathrm{Reg}_q(t)}{T'} &\leq C_{p,1} \Big( t^{-(1-\alpha)} + t^{-\alpha} \\
&\quad + \sqrt{\tfrac{\log t}{t}} + t^{-3/4} \Big),
\end{aligned}
$$
(14)

for a finite constant $C_{p,1}$ depending only on fixed parameters.

These regret bounds are with $\hat{\boldsymbol{\theta}}$ we now use them to bound the 'true' regret. Define the true-game regrets:

$$
\mathrm{Reg}_w^{\boldsymbol{\theta}}(t) := \sup_{\boldsymbol{w} \in \Delta_{\mathcal{P}}} \sum_{s=b_t}^{t} \big( F_{\boldsymbol{\theta}}(\boldsymbol{w}, \boldsymbol{q}_s) - F_{\boldsymbol{\theta}}(\boldsymbol{w}_s, \boldsymbol{q}_s) \big),
$$

$$
\mathrm{Reg}_q^{\boldsymbol{\theta}}(t) := \sup_{\boldsymbol{q} \in \Delta_B} \sum_{s=b_t}^{t} \big( F_{\boldsymbol{\theta}}(\boldsymbol{w}_s, \boldsymbol{q}_s) - F_{\boldsymbol{\theta}}(\boldsymbol{w}_s, \boldsymbol{q}) \big).
$$

For each $s$ define $\Delta F_s := \sup_{\boldsymbol{w} \in \Delta_{\mathcal{P}}, \boldsymbol{q} \in \Delta_B} |F_{\hat{\boldsymbol{\theta}}_s}(\boldsymbol{w}, \boldsymbol{q}) - F_{\boldsymbol{\theta}}(\boldsymbol{w}, \boldsymbol{q})|$. Then for any $\boldsymbol{w}$ and each $s$,

$$
F_{\boldsymbol{\theta}}(\boldsymbol{w}, \boldsymbol{q}_s) - F_{\boldsymbol{\theta}}(\boldsymbol{w}_s, \boldsymbol{q}_s) \leq F_{\hat{\boldsymbol{\theta}}_s}(\boldsymbol{w}, \boldsymbol{q}_s) - F_{\hat{\boldsymbol{\theta}}_s}(\boldsymbol{w}_s, \boldsymbol{q}_s) + 2\Delta F_s.
$$

Summing from $s = b_t$ to $t$ and taking $\sup_{\boldsymbol{w}}$ gives

$$\operatorname{Reg}_w^{\boldsymbol{\theta}}(t) \leq \operatorname{Reg}_w(t) + 2 \sum_{s=b_t}^{t} \Delta F_s,$$

$$\operatorname{Reg}_q^{\boldsymbol{\theta}}(t) \leq \operatorname{Reg}_q(t) + 2 \sum_{s=b_t}^{t} \Delta F_s,$$

hence

$$\operatorname{Reg}_w^{\boldsymbol{\theta}}(t) + \operatorname{Reg}_q^{\boldsymbol{\theta}}(t) \leq \operatorname{Reg}_w(t) + \operatorname{Reg}_q(t) + 4 \sum_{s=b_t}^{t} \Delta F_s.$$
(15)

By Lemma F.3, $\Delta F_s \leq L\|\hat{\boldsymbol{\theta}}_s - \boldsymbol{\theta}\|_2$. Therefore,

$$\frac{1}{T'} \sum_{s=b_t}^{t} \Delta F_s \leq L \cdot \frac{1}{T'} \sum_{s=b_t}^{t} \|\hat{\boldsymbol{\theta}}_s - \boldsymbol{\theta}\|_2.$$

On $\mathcal{M}_t$, for each $s \in \{b_t, \dots, t\}$ we already have

$$\|\hat{\boldsymbol{\theta}}_s - \boldsymbol{\theta}\|_2 \leq \frac{2\sigma}{\underline{a}} \sqrt{\frac{n \log(2|\mathcal{P}| t^{p+3})}{c_\gamma \, s^{1-\gamma}}} = C \sqrt{\log t} \, s^{-(1-\gamma)/2}$$

for a finite constant $C$. Since $T' \geq t/2$ for $t \geq 2$, and by Fact F.2

$$\frac{1}{T'} \sum_{s=b_t}^{t} s^{-(1-\gamma)/2} \leq \frac{2}{t}\left(1 + \frac{t^{(1+\gamma)/2}}{(1+\gamma)/2}\right)$$

$$\leq \frac{8}{1+\gamma} t^{-(1-\gamma)/2} \qquad (t \geq 2).$$

Thus on $\mathcal{M}_t$,

$$\frac{1}{T'} \sum_{s=b_t}^{t} \|\hat{\boldsymbol{\theta}}_s - \boldsymbol{\theta}\|_2 \leq C_{p,2} \, t^{-(1-\gamma)/2} \sqrt{\log t}. \qquad (16)$$

Combining (15), (14), and (16) and choosing finite constant $C_{p,3}$ gives that on $\mathcal{M}_t \cap \mathcal{R}_t$,

$$\frac{\operatorname{Reg}_w^{\boldsymbol{\theta}}(t) + \operatorname{Reg}_q^{\boldsymbol{\theta}}(t)}{T'} \leq C_{p,3}\left(t^{-(1-\alpha)} + t^{-\alpha} + \sqrt{\frac{\log t}{t}}\right.$$
$$\left. + t^{-3/4} + t^{-(1-\gamma)/2}\sqrt{\log t}\right)$$
(17)

By Lemma C.3,

$$\operatorname{gap}_t^{\boldsymbol{\theta}} \leq \frac{\operatorname{Reg}_w^{\boldsymbol{\theta}}(t) + \operatorname{Reg}_q^{\boldsymbol{\theta}}(t)}{T'}.$$

Also, $\sup_{\boldsymbol{w}} F_{\boldsymbol{\theta}}(\boldsymbol{w}, \bar{\boldsymbol{q}}_{b_t:t}) \geq \Gamma^*(\boldsymbol{\theta})$, hence

$$\Gamma(\bar{\boldsymbol{w}}_{b_t:t}; \boldsymbol{\theta}) = \inf_{\boldsymbol{q}} F_{\boldsymbol{\theta}}(\bar{\boldsymbol{w}}_{b_t:t}, \boldsymbol{q}) \geq \Gamma^*(\boldsymbol{\theta}) - \operatorname{gap}_t^{\boldsymbol{\theta}}.$$

Therefore, on $\mathcal{M}_t \cap \mathcal{R}_t$, combining with (17) gives

$$\Gamma(\bar{\boldsymbol{w}}_{b_t:t}; \boldsymbol{\theta}) \geq \Gamma^*(\boldsymbol{\theta})$$
$$- C_{p,3}\left(t^{-(1-\alpha)} + t^{-\alpha} + \sqrt{\frac{\log t}{t}}\right.$$
$$\left. + t^{-3/4} + t^{-(1-\gamma)/2}\sqrt{\log t}\right)$$

Next, we transfer this guarantee to $\boldsymbol{w}_t^{\text{emp}}$. First,

$$\|\bar{\boldsymbol{w}}_t - \bar{\boldsymbol{w}}_{b_t:t}\|_1 \leq \frac{2(b_t - 1)}{t} \leq 2t^{-3/4}.$$

Second, $|\Gamma(\boldsymbol{w}; \boldsymbol{\theta}) - \Gamma(\boldsymbol{w}'; \boldsymbol{\theta})| \leq D_{\max}\|\boldsymbol{w} - \boldsymbol{w}'\|_1$. Combining these gives

$$\Gamma(\bar{\boldsymbol{w}}_t; \boldsymbol{\theta}) \geq \Gamma(\bar{\boldsymbol{w}}_{b_t:t}; \boldsymbol{\theta}) - 2D_{\max} t^{-3/4}.$$

Finally, by Lemma F.1(c),

$$\|\hat{\boldsymbol{w}}_t^{\text{emp}} - \bar{\boldsymbol{w}}_t\|_\infty \leq \frac{|\mathcal{P}| - 1}{t} + \frac{1}{t}\sum_{s=1}^{t} \rho_s \leq \frac{|\mathcal{P}| - 1}{t} + \frac{t^{-\gamma}}{1-\gamma}$$

Therefore,

$$\|\hat{\boldsymbol{w}}_t^{\text{emp}} - \bar{\boldsymbol{w}}_t\|_1 \leq |\mathcal{P}|\left(\frac{|\mathcal{P}| - 1}{t} + \frac{t^{-\gamma}}{1-\gamma}\right)$$

By Lipschitzness of $\Gamma$ and the previous bounds, on $\mathcal{M}_t \cap \mathcal{R}_t$ we obtain

$$\Gamma(\hat{\boldsymbol{w}}_t^{\text{emp}}; \boldsymbol{\theta}) \geq \Gamma^*(\boldsymbol{\theta})$$
$$- C_{p,3}\left(t^{-(1-\alpha)} + t^{-\alpha} + \sqrt{\frac{\log t}{t}} + t^{-3/4}\right)$$
$$- C_{p,3}\, t^{-(1-\gamma)/2}\sqrt{\log t}$$
$$- D_{\max}|\mathcal{P}|\left(\frac{|\mathcal{P}| - 1}{t} + \frac{1}{1-\gamma} t^{-\gamma}\right)$$
$$- 2D_{\max} t^{-3/4}.$$

This is the inequality in Proposition 6.3.

For $t \geq t_p$, we have shown $\mathcal{M}_t \cap \mathcal{R}_t$ implies this inequality. Therefore, the failure event at time $t$ is contained in $\mathcal{M}_t^c \cup \mathcal{R}_t^c$. Hence, by (12) and (13),

$$\mathbb{P}_{\boldsymbol{\theta}}(\text{failure at time } t) \leq \mathbb{P}_{\boldsymbol{\theta}}(\mathcal{M}_t^c) + \mathbb{P}_{\boldsymbol{\theta}}(\mathcal{R}_t^c)$$
$$\leq t^{-(p+1)} + t^{-(p+2)}$$
$$\leq t^{-p} \qquad (\forall t \geq 2).$$

This proves Proposition 6.3 for all $t \geq t_p$. $\qquad \square$

## D. Proof of Proposition 6.4

**Outline.** This section proves the two parts of Proposition 6.4: (i) the $\delta$-correct risk bound and (ii) almost sure finite stopping. For (i), we first introduce a mixture

likelihood-ratio martingale valid under adaptive sampling (Lemma D.1), then lower bound it to obtain the stopping threshold (Lemma D.2), and finally apply Ville's inequality to get the risk bound (Proposition D.3). For (ii), we need to show that the GLR statistic eventually will be larger than the stopping threshold. To do this, bound the difference of the normalized GLR statistic to the information rate once the boundary constraints are correct (Lemma D.5). Then (Lemma D.6 and Corollary D.7) show this difference goes to 0 almost surely. Finally, (Lemma D.8 and Proposition D.9) show that the GLR statistic grows linearly and the stopping threshold grows sublinearly, to guarantee a finite stopping time. The combination of Propositions D.3 and D.9 gives Proposition 6.4.

### D.1. Part I: Risk Bound

**Lemma D.1.** *Fix $\boldsymbol{\theta} \in \Theta$ and let $\pi$ be any probability distribution on $\Theta_0$. Define*

$$M_t := \int_{\Theta_0} \exp\big(\ell_t(\boldsymbol{u}) - \ell_t(\boldsymbol{\theta})\big)\, \pi(d\boldsymbol{u}), \qquad M_0 := 1.$$

*Then $(M_t)_{t \geq 0}$ is a nonnegative martingale with respect to $\mathcal{F}_t = \sigma((A_s, Y_s) : s \leq t)$ for any adaptive sampling rule.*

*Proof.* See Kaufmann & Koolen (2021), Sec. 2.3. $\qquad\square$

The argument in Lemma D.2 closely follows existing work on stopping thresholds for linear bandits, for example, see Abbasi-Yadkori et al. (2011).

**Lemma D.2.** *Fix $\boldsymbol{\theta} \in \Theta$ and let $d = \dim(\Theta_0) = n - 1$. Let $\lambda > 0$. Let $\boldsymbol{\zeta} \sim \mathcal{N}(0, \lambda^{-1}I_n)$ in $\mathbb{R}^n$ and let*

$$\Pi := I_n - \frac{1}{n}\mathbf{1}\mathbf{1}^\top$$

*denote the orthogonal projector onto $\Theta_0$. Define $\boldsymbol{\vartheta} := \Pi\boldsymbol{\zeta} \in \Theta_0$, and let $\pi_\lambda$ denote its distribution. Define the mixture process*

$$M_t := \mathbb{E}\big[\exp\big(\ell_t(\boldsymbol{\vartheta}) - \ell_t(\boldsymbol{\theta})\big)\big], \qquad M_0 := 1,$$

*and let $\hat{\boldsymbol{\theta}}_t \in \mathrm{argmax}_{\boldsymbol{\vartheta} \in \Theta} \ell_t(\boldsymbol{\vartheta})$ be a constrained MLE. Define the Laplacian at time $t$, $\mathcal{L}_t := \sum_{s=1}^t \boldsymbol{x}_{A_s}\boldsymbol{x}_{A_s}^\top$*

*Then for all $t \geq 1$,*

$$\begin{aligned} M_t &\geq \exp\big(\ell_t(\hat{\boldsymbol{\theta}}_t) - \ell_t(\boldsymbol{\theta})\big) \\ &\quad \cdot \exp\Big(-\frac{\lambda}{2}\|\hat{\boldsymbol{\theta}}_t\|_2^2\Big) \cdot \det\Big(I_n + \frac{\overline{\sigma}^2}{\lambda}\mathcal{L}_t\Big)^{-1/2}. \end{aligned} \quad (18)$$

*Equivalently,*

$$\ell_t(\hat{\boldsymbol{\theta}}_t) - \ell_t(\boldsymbol{\theta}) \leq \log M_t + \frac{\lambda}{2}\|\hat{\boldsymbol{\theta}}_t\|_2^2 + \frac{1}{2}\log\det\Big(I_n + \frac{\overline{\sigma}^2}{\lambda}\mathcal{L}_t\Big).$$

*Proof.* Take $U \in \mathbb{R}^{n \times d}$ that has orthonormal columns spanning $\Theta_0$, so $U^\top U = I_d$ and $U^\top \mathbf{1} = 0$. Then $\Pi = UU^\top$, and every $\boldsymbol{\vartheta} \in \Theta_0$ can be written uniquely as $\boldsymbol{\vartheta} = U\boldsymbol{z}$ with $\boldsymbol{z} \in \mathbb{R}^d$ and $\|\boldsymbol{\vartheta}\|_2 = \|\boldsymbol{z}\|_2$. Define

$$\tilde{\ell}_t(\boldsymbol{z}) := \ell_t(U\boldsymbol{z}), \qquad \boldsymbol{z}_{\boldsymbol{\theta}} := U^\top\boldsymbol{\theta}, \qquad \hat{\boldsymbol{z}}_t := U^\top\hat{\boldsymbol{\theta}}_t.$$

With $\boldsymbol{\zeta}$ as in the statement, set $\boldsymbol{Z} := U^\top\boldsymbol{\zeta}$. Then $\boldsymbol{Z} \sim \mathcal{N}(0, \lambda^{-1}I_d)$ and $\boldsymbol{\vartheta} = \Pi\boldsymbol{\zeta} = UU^\top\boldsymbol{\zeta} = U\boldsymbol{Z}$. Therefore

$$\begin{aligned} M_t &= \mathbb{E}\Big[\exp\big(\tilde{\ell}_t(\boldsymbol{Z}) - \tilde{\ell}_t(\boldsymbol{z}_{\boldsymbol{\theta}})\big)\Big] \\ &= \int_{\mathbb{R}^d} \exp\big(\tilde{\ell}_t(\boldsymbol{z}) - \tilde{\ell}_t(\boldsymbol{z}_{\boldsymbol{\theta}})\big)\, \varphi_\lambda(\boldsymbol{z})\, d\boldsymbol{z} \\ &= \Big(\frac{\lambda}{2\pi}\Big)^{d/2} e^{-\tilde{\ell}_t(\boldsymbol{z}_{\boldsymbol{\theta}})} \int_{\mathbb{R}^d} \exp\Big(\tilde{\ell}_t(\boldsymbol{z}) - \frac{\lambda}{2}\|\boldsymbol{z}\|_2^2\Big)\, d\boldsymbol{z}. \end{aligned}$$
$$\quad (19)$$

Next, we bound the Hessian. For any $\boldsymbol{\vartheta} \in \Theta_0$, by Assumption 1

$$\begin{aligned} -\nabla^2\ell_t(\boldsymbol{\vartheta}) &= \sum_{s=1}^t A''\big(\eta_{A_s}(\boldsymbol{\vartheta})\big)\, \boldsymbol{x}_{A_s}\boldsymbol{x}_{A_s}^\top \\ &\preceq \overline{\sigma}^2 \sum_{s=1}^t \boldsymbol{x}_{A_s}\boldsymbol{x}_{A_s}^\top = \overline{\sigma}^2\mathcal{L}_t \end{aligned}$$

Projecting onto $\Theta_0$ gives, for all $\boldsymbol{z} \in \mathbb{R}^d$,

$$-\nabla^2\tilde{\ell}_t(\boldsymbol{z}) = -U^\top\nabla^2\ell_t(U\boldsymbol{z})\, U \preceq \overline{\sigma}^2 U^\top\mathcal{L}_t U.$$

Next, we lower bound $f_t(\boldsymbol{z}) := \tilde{\ell}_t(\boldsymbol{z}) - \frac{\lambda}{2}\|\boldsymbol{z}\|_2^2$. For all $\boldsymbol{z} \in \mathbb{R}^d$,

$$-\nabla^2 f_t(\boldsymbol{z}) = -\nabla^2\tilde{\ell}_t(\boldsymbol{z}) + \lambda I_d \preceq \overline{\sigma}^2 U^\top\mathcal{L}_t U + \lambda I_d =: H_t,$$

where $H_t \succ 0$ since $\lambda > 0$. Let $\boldsymbol{z}_t^\lambda \in \mathrm{argmax}_{\boldsymbol{z} \in \mathbb{R}^d} f_t(\boldsymbol{z})$, which exists and is unique by $\lambda$-strong concavity. Since $\nabla f_t(\boldsymbol{z}_t^\lambda) = 0$ and $-\nabla^2 f_t \preceq H_t$ everywhere, Taylor's theorem gives, for all $\boldsymbol{z} \in \mathbb{R}^d$,

$$f_t(\boldsymbol{z}) \geq f_t(\boldsymbol{z}_t^\lambda) - \frac{1}{2}(\boldsymbol{z} - \boldsymbol{z}_t^\lambda)^\top H_t(\boldsymbol{z} - \boldsymbol{z}_t^\lambda).$$

Thus,

$$\begin{aligned} \int_{\mathbb{R}^d} e^{f_t(\boldsymbol{z})} d\boldsymbol{z} &\geq e^{f_t(\boldsymbol{z}_t^\lambda)} \int_{\mathbb{R}^d} \exp\big(-\tfrac{1}{2}(\boldsymbol{z} - \boldsymbol{z}_t^\lambda)^\top H_t(\boldsymbol{z} - \boldsymbol{z}_t^\lambda)\big) d\boldsymbol{z} \\ &= e^{f_t(\boldsymbol{z}_t^\lambda)} \frac{(2\pi)^{d/2}}{\det(H_t)^{1/2}}. \end{aligned}$$

Plugging into (19) yields

$$M_t \geq \exp\big(f_t(\boldsymbol{z}_t^\lambda) - \tilde{\ell}_t(\boldsymbol{z}_{\boldsymbol{\theta}})\big) \cdot \frac{\lambda^{d/2}}{\det(H_t)^{1/2}}. \quad (20)$$

$\boldsymbol{z}_t^\lambda$ maximizes $f_t$ over $\mathbb{R}^d$, we have $f_t(\boldsymbol{z}_t^\lambda) \geq f_t(\hat{\boldsymbol{z}}_t)$. Noting $\tilde{\ell}_t(\hat{\boldsymbol{z}}_t) = \ell_t(\hat{\boldsymbol{\theta}}_t)$ and $\|\hat{\boldsymbol{z}}_t\|_2 = \|\hat{\boldsymbol{\theta}}_t\|_2$, we get

$$f_t(\hat{\boldsymbol{z}}_t) - \tilde{\ell}_t(\boldsymbol{z}_{\boldsymbol{\theta}}) = \ell_t(\hat{\boldsymbol{\theta}}_t) - \ell_t(\boldsymbol{\theta}) - \frac{\lambda}{2}\|\hat{\boldsymbol{\theta}}_t\|_2^2.$$

Also,

$$H_t = \lambda I_d + \overline{\sigma}^2\, U^\top \mathcal{L}_t U = \lambda\Big(I_d + \frac{\overline{\sigma}^2}{\lambda} U^\top \mathcal{L}_t U\Big),$$

so $\det(H_t) = \lambda^d \det(I_d + \frac{\overline{\sigma}^2}{\lambda} U^\top \mathcal{L}_t U)$ and therefore

$$\frac{\lambda^{d/2}}{\det(H_t)^{1/2}} = \det\Big(I_d + \frac{\overline{\sigma}^2}{\lambda} U^\top \mathcal{L}_t U\Big)^{-1/2}$$
$$= \det\Big(I_n + \frac{\overline{\sigma}^2}{\lambda}\mathcal{L}_t\Big)^{-1/2}$$

where the last equality follows from $\mathcal{L}_t \mathbf{1} = 0$. Combining these with (20) gives (18). □

**Proposition D.3.** *Fix any $\boldsymbol{\theta} \in \Theta^{\mathrm{gap}}$ and $\lambda > 0$. Define the threshold*

$$\beta(t,\delta) := \log\frac{1}{\delta} + \frac{\lambda}{2}\|\hat{\boldsymbol{\theta}}_t\|_2^2 + \frac{1}{2}\log\det\Big(I_n + \frac{\overline{\sigma}^2}{\lambda}\mathcal{L}_t\Big), \tag{21}$$

*and the stopping time*

$$\tau_\delta := \inf\{t \geq 1 : Z_{\hat{S}_t}(t) \geq \beta(t,\delta)\}, \qquad \text{output } \hat{S}_{\tau_\delta}.$$

*Then*

$$\mathbb{P}_{\boldsymbol{\theta}}(\hat{S}_{\tau_\delta} \neq S^*(\boldsymbol{\theta})) \leq \delta.$$

*Proof.* On the event $\{\tau_\delta = t,\ \hat{S}_t \neq S^*(\boldsymbol{\theta})\}$ there exist $u \in \hat{S}_t \setminus S^*(\boldsymbol{\theta})$ and $v \in S^*(\boldsymbol{\theta}) \setminus \hat{S}_t$. Then $\theta_v \geq \theta_u$, so $\boldsymbol{\theta} \in \Theta_{uv}(t)$. And $(u,v) \in \hat{B}_t$, so

$$Z_{uv}(t) = \ell_t(\hat{\boldsymbol{\theta}}_t) - \sup_{\boldsymbol{\theta}' \in \Theta_{uv}(t)} \ell_t(\boldsymbol{\theta}') \leq \ell_t(\hat{\boldsymbol{\theta}}_t) - \ell_t(\boldsymbol{\theta}).$$

Since $\tau_\delta = t$ implies $Z_{\hat{S}_t}(t) = \min_{(i,j)\in\hat{B}_t} Z_{ij}(t) \geq \beta(t,\delta)$, we have

$$\ell_t(\hat{\boldsymbol{\theta}}_t) - \ell_t(\boldsymbol{\theta}) \geq Z_{uv}(t) \geq \beta(t,\delta).$$

Now let $\pi_\lambda$ be the Gaussian prior on $\Theta_0$ in Lemma D.2 and define

$$M_t := \int_{\Theta_0} \exp\big(\ell_t(\boldsymbol{u}) - \ell_t(\boldsymbol{\theta})\big)\, \pi_\lambda(d\boldsymbol{u}), \qquad M_0 := 1.$$

By Lemma D.1 $(M_t)$ is a nonnegative martingale. Apply Lemma D.2 at $t = \tau_\delta$,

$$M_{\tau_\delta} \geq \exp\big(\ell_{\tau_\delta}(\hat{\boldsymbol{\theta}}_{\tau_\delta}) - \ell_{\tau_\delta}(\boldsymbol{\theta})\big)$$
$$\cdot \exp\Big(-\frac{\lambda}{2}\|\hat{\boldsymbol{\theta}}_{\tau_\delta}\|_2^2\Big) \cdot \det\Big(I_n + \frac{\overline{\sigma}^2}{\lambda}\mathcal{L}_{\tau_\delta}\Big)^{-1/2}.$$

On the event $\{\ell_{\tau_\delta}(\hat{\boldsymbol{\theta}}_{\tau_\delta}) - \ell_{\tau_\delta}(\boldsymbol{\theta}) \geq \beta(\tau_\delta,\delta)\}$,

$$M_{\tau_\delta} \geq \exp\big(\beta(\tau_\delta,\delta)\big) \cdot \exp\Big(-\frac{\lambda}{2}\|\hat{\boldsymbol{\theta}}_{\tau_\delta}\|_2^2\Big)$$
$$\cdot \det\Big(I_n + \frac{\overline{\sigma}^2}{\lambda}\mathcal{L}_{\tau_\delta}\Big)^{-1/2}$$
$$= \exp\Big(\log\tfrac{1}{\delta} + \tfrac{\lambda}{2}\|\hat{\boldsymbol{\theta}}_{\tau_\delta}\|_2^2 + \tfrac{1}{2}\log\det\big(I_n + \tfrac{\overline{\sigma}^2}{\lambda}\mathcal{L}_{\tau_\delta}\big)\Big)$$
$$\cdot \exp\Big(-\tfrac{\lambda}{2}\|\hat{\boldsymbol{\theta}}_{\tau_\delta}\|_2^2\Big) \cdot \det\Big(I_n + \tfrac{\overline{\sigma}^2}{\lambda}\mathcal{L}_{\tau_\delta}\Big)^{-1/2}$$
$$= \frac{1}{\delta}.$$

Hence

$$\{\hat{S}_{\tau_\delta} \neq S^*(\boldsymbol{\theta})\} \subseteq \Big\{\sup_{t\geq 0} M_t \geq 1/\delta\Big\}.$$

Since $(M_t)$ is a nonnegative martingale with $M_0 = 1$, Ville's inequality gives

$$\mathbb{P}_{\boldsymbol{\theta}}\Big(\sup_{t\geq 0} M_t \geq 1/\delta\Big) \leq \delta.$$

□

## D.2. Part II: Finite Stopping

First, we show that the data-dependent threshold from Lemma D.2 is bounded by some constant.

**Lemma D.4.** *Fix $\lambda > 0$ and $\delta \in (0,1)$, and let $\beta(t,\delta)$ be as in Proposition D.3. Let $d = \dim(\Theta_0) = n-1$ and define the constant*

$$C_\beta := \frac{2\overline{\sigma}^2}{\lambda}.$$

*Then for all $t \geq 1$,*

$$\beta(t,\delta) \leq \bar{\beta}(t,\delta) := \log\frac{1}{\delta} + \frac{\lambda}{2}nR^2 + \frac{d}{2}\log\big(1 + C_\beta t\big).$$

*Proof.* First, $\|\hat{\boldsymbol{\theta}}_t\|_\infty \leq R$ implies $\|\hat{\boldsymbol{\theta}}_t\|_2^2 \leq nR^2$. Next, $\mathcal{L}_t = \sum_{s=1}^t \boldsymbol{x}_{A_s}\boldsymbol{x}_{A_s}^\top \succeq 0$ and $\mathcal{L}_t \mathbf{1} = 0$, so $\mathcal{L}_t$ has at most $d$ nonzero eigenvalues. Also $\|\boldsymbol{x}_{ij}\|_2^2 = 2$ implies $\|\boldsymbol{x}_{A_s}\boldsymbol{x}_{A_s}^\top\|_{\mathrm{op}} = 2$, hence $\|\mathcal{L}_t\|_{\mathrm{op}} \leq 2t$. Therefore,

$$\det\Big(I_n + \frac{\overline{\sigma}^2}{\lambda}\mathcal{L}_t\Big) \leq \Big(1 + \frac{\overline{\sigma}^2}{\lambda}\|\mathcal{L}_t\|_{\mathrm{op}}\Big)^d \leq (1 + C_\beta t)^d,$$

□

The threshold $\beta(t,\delta)$ grows at most logarithmically with $t$. Thus, to show finite stopping, it is sufficient to show that $\frac{1}{t}Z_{\hat{S}_t}(t)$ converges to some positive number.

For $\boldsymbol{\theta}' \in \Theta$, define

$$
\begin{aligned}
\bar{L}_t(\boldsymbol{\theta}') &:= -\frac{1}{t}\,\ell_t(\boldsymbol{\theta}') \\
&= \frac{1}{t}\sum_{s=1}^{t}\Big(A(\eta_{A_s}(\boldsymbol{\theta}')) - \eta_{A_s}(\boldsymbol{\theta}')\,T(Y_s)\Big),
\end{aligned}
$$

$$
L_t(\boldsymbol{\theta}') := \frac{1}{t}\sum_{s=1}^{t}\mathbb{E}_{\boldsymbol{\theta}}\Big[A(\eta_{A_s}(\boldsymbol{\theta}')) - \eta_{A_s}(\boldsymbol{\theta}')\,T(Y_s)\,\Big|\,\mathcal{F}_{s-1}\Big],
$$

and the uniform deviation

$$
\Delta_t := \sup_{\boldsymbol{\theta}'\in\Theta}\big|\bar{L}_t(\boldsymbol{\theta}') - L_t(\boldsymbol{\theta}')\big|.
$$

**Lemma D.5.** *Fix $t \geq 1$ and $\boldsymbol{\theta} \in \Theta^{\mathrm{gap}}$.*

*(a) For every $\boldsymbol{\theta}' \in \Theta$,*

$$
L_t(\boldsymbol{\theta}') \;=\; L_t(\boldsymbol{\theta}) \;+\; D_{\hat{\boldsymbol{w}}_t^{\mathrm{emp}}}(\boldsymbol{\theta}\|\boldsymbol{\theta}'). \qquad (22)
$$

*(b) Suppose that at time $t$ the estimated boundary constraints are correct, i.e. $\hat{B}_t = B(\boldsymbol{\theta})$ and $\Theta_{ij}(t) = \Theta_{ij}$ for all $(i,j) \in B(\boldsymbol{\theta})$. Then for every $(i,j) \in B(\boldsymbol{\theta})$,*

$$
\Big|\frac{1}{t}Z_{ij}(t) \;-\; \gamma_{ij}(\hat{\boldsymbol{w}}_t^{\mathrm{emp}};\boldsymbol{\theta})\Big| \;\leq\; 2\Delta_t, \qquad (23)
$$

*Proof.* (a)

$$
\mathbb{E}_{\boldsymbol{\theta}}\big[T(Y_s)\mid\mathcal{F}_{s-1}\big] \;=\; A'(\eta_{A_s}(\boldsymbol{\theta})).
$$

Therefore,

$$
\begin{aligned}
L_t(\boldsymbol{\theta}') - L_t(\boldsymbol{\theta}) &= \frac{1}{t}\sum_{s=1}^{t}\Big(A(\eta_{A_s}(\boldsymbol{\theta}')) - A(\eta_{A_s}(\boldsymbol{\theta})) \\
&\quad - \big(\eta_{A_s}(\boldsymbol{\theta}') - \eta_{A_s}(\boldsymbol{\theta})\big)A'(\eta_{A_s}(\boldsymbol{\theta}))\Big) \\
&= \frac{1}{t}\sum_{s=1}^{t}d_{A_s}(\boldsymbol{\theta},\boldsymbol{\theta}').
\end{aligned}
$$

Since $\hat{w}_{t,ij}^{\mathrm{emp}} = N_{ij}(t)/t$,

$$
D_{\hat{\boldsymbol{w}}_t^{\mathrm{emp}}}(\boldsymbol{\theta}\|\boldsymbol{\theta}') = \sum_{(i,j)}\hat{w}_{t,ij}^{\mathrm{emp}}\,d_{ij}(\boldsymbol{\theta},\boldsymbol{\theta}') = \frac{1}{t}\sum_{s=1}^{t}d_{A_s}(\boldsymbol{\theta},\boldsymbol{\theta}'),
$$

which proves (22).

**Part (b).** By assumption, $\hat{B}_t = B(\boldsymbol{\theta})$ and for each $(i,j) \in B(\boldsymbol{\theta})$, the constrained maximization in $Z_{ij}(t)$ is over $\Theta_{ij}$. Let

$$
\hat{\boldsymbol{\theta}}_{ij,t} \in \operatorname*{argmax}_{\boldsymbol{\theta}'\in\Theta_{ij}}\ell_t(\boldsymbol{\theta}') \Leftrightarrow \hat{\boldsymbol{\theta}}_{ij,t} \in \operatorname*{argmin}_{\boldsymbol{\theta}'\in\Theta_{ij}}\bar{L}_t(\boldsymbol{\theta}'),
$$

and recall $\hat{\boldsymbol{\theta}}_t \in \operatorname*{argmax}_{\Theta}\ell_t(\boldsymbol{\theta}') \Leftrightarrow \hat{\boldsymbol{\theta}}_t \in \operatorname*{argmin}_{\Theta}\bar{L}_t(\boldsymbol{\theta}')$. Then

$$
\frac{1}{t}Z_{ij}(t) = \bar{L}_t(\hat{\boldsymbol{\theta}}_{ij,t}) - \bar{L}_t(\hat{\boldsymbol{\theta}}_t) = \inf_{\boldsymbol{\theta}'\in\Theta_{ij}}\bar{L}_t(\boldsymbol{\theta}') - \inf_{\boldsymbol{\theta}'\in\Theta}\bar{L}_t(\boldsymbol{\theta}').
$$

On the other hand, by (22) and nonnegativity of KL divergence, $\inf_{\boldsymbol{\theta}'\in\Theta}L_t(\boldsymbol{\theta}') = L_t(\boldsymbol{\theta})$, so

$$
\begin{aligned}
\inf_{\boldsymbol{\theta}'\in\Theta_{ij}}L_t(\boldsymbol{\theta}') &= L_t(\boldsymbol{\theta}) + \inf_{\boldsymbol{\theta}'\in\Theta_{ij}}D_{\hat{\boldsymbol{w}}_t^{\mathrm{emp}}}(\boldsymbol{\theta}\|\boldsymbol{\theta}') \\
&= L_t(\boldsymbol{\theta}) + \gamma_{ij}(\hat{\boldsymbol{w}}_t^{\mathrm{emp}};\boldsymbol{\theta}).
\end{aligned}
$$

Thus

$$
\gamma_{ij}(\hat{\boldsymbol{w}}_t^{\mathrm{emp}};\boldsymbol{\theta}) = \inf_{\Theta_{ij}}L_t - \inf_{\Theta}L_t.
$$

Using $|\inf f - \inf g| \leq \sup|f-g|$,

$$
\begin{aligned}
\Big|\frac{1}{t}Z_{ij}(t) - \gamma_{ij}(\hat{\boldsymbol{w}}_t^{\mathrm{emp}};\boldsymbol{\theta})\Big| &= \Big|\big(\inf_{\Theta_{ij}}\bar{L}_t - \inf_{\Theta}\bar{L}_t\big) \\
&\quad - \big(\inf_{\Theta_{ij}}L_t - \inf_{\Theta}L_t\big)\Big| \\
&\leq \Big|\inf_{\Theta_{ij}}\bar{L}_t - \inf_{\Theta_{ij}}L_t\Big| \\
&\quad + \Big|\inf_{\Theta}\bar{L}_t - \inf_{\Theta}L_t\Big| \leq 2\Delta_t,
\end{aligned}
$$

which proves (23). $\qquad\square$

Next, we will show that $\Delta_t \to 0$ almost surely. Define

$$
\boldsymbol{V}_t := \sum_{s=1}^{t}\boldsymbol{x}_{A_s}\Big(T(Y_s) - A'(\eta_{A_s}(\boldsymbol{\theta}))\Big) \in \mathbb{R}^n.
$$

So for every $\boldsymbol{\theta}' \in \Theta$,

$$
\bar{L}_t(\boldsymbol{\theta}') - L_t(\boldsymbol{\theta}') \;=\; -\frac{1}{t}\,\boldsymbol{\theta}'^{\top}\boldsymbol{V}_t. \qquad (24)
$$

**Lemma D.6.** *For every $t \geq 1$ and every $\delta \in (0,1)$,*

$$
\mathbb{P}_{\boldsymbol{\theta}}\left(\Delta_t \;\geq\; R\,\sigma\,n\,\sqrt{\frac{2\log(2n/\delta)}{t}}\right) \;\leq\; \delta. \qquad (25)
$$

*Proof.* By (24),

$$
\begin{aligned}
\Delta_t &= \sup_{\boldsymbol{\theta}'\in\Theta}\frac{|\boldsymbol{\theta}'^{\top}\boldsymbol{V}_t|}{t} \\
&\leq \sup_{\|\boldsymbol{\theta}'\|_\infty\leq R}\frac{|\boldsymbol{\theta}'^{\top}\boldsymbol{V}_t|}{t} = \frac{R}{t}\|\boldsymbol{V}_t\|_1 \leq \frac{Rn}{t}\max_{1\leq i\leq n}|(\boldsymbol{V}_t)_i|.
\end{aligned}
$$

Fix $i \in [n]$ and write $(\boldsymbol{V}_t)_i = \sum_{s=1}^{t}\xi_{s,i}$ where

$$
\xi_{s,i} := (\boldsymbol{x}_{A_s})_i\Big(T(Y_s) - A'(\eta_{A_s}(\boldsymbol{\theta}))\Big).
$$

Conditionally on $\mathcal{F}_{s-1}$, $(\boldsymbol{x}_{A_s})_i \in \{-1, 0, 1\}$ is fixed, and $T(Y_s) - A'(\eta_{A_s}(\boldsymbol{\theta}))$ is centered and $\sigma^2$-sub-Gaussian by Assumption 2. So for all $\lambda \in \mathbb{R}$,

$$\mathbb{E}_{\boldsymbol{\theta}}[\exp(\lambda \xi_{s,i}) \mid \mathcal{F}_{s-1}] \leq \exp\left(\frac{\sigma^2 \lambda^2}{2}\right).$$

Define

$$M_{s,i}(\lambda) := \exp\left(\lambda(\boldsymbol{V}_s)_i - \frac{\sigma^2 \lambda^2}{2} s\right).$$

Then $(M_{s,i}(\lambda))_{s \geq 0}$ is a nonnegative supermartingale, so $\mathbb{E}_{\boldsymbol{\theta}}[M_{t,i}(\lambda)] \leq 1$. By Markov's inequality, for any $a > 0$,

$$\mathbb{P}_{\boldsymbol{\theta}}((\boldsymbol{V}_t)_i \geq a) \leq \exp\left(-\lambda a + \frac{\sigma^2 \lambda^2}{2} t\right).$$

Take $\lambda = a/(\sigma^2 t)$ to get

$$\mathbb{P}_{\boldsymbol{\theta}}((\boldsymbol{V}_t)_i \geq a) \leq \exp\left(-\frac{a^2}{2\sigma^2 t}\right),$$

$$\mathbb{P}_{\boldsymbol{\theta}}(|(\boldsymbol{V}_t)_i| \geq a) \leq 2\exp\left(-\frac{a^2}{2\sigma^2 t}\right).$$

With $a = \sigma\sqrt{2t \log(2n/\delta)}$,

$$\mathbb{P}_{\boldsymbol{\theta}}\left(|(\boldsymbol{V}_t)_i| \geq \sigma\sqrt{2t \log(2n/\delta)}\right) \leq \frac{\delta}{n}.$$

A union bound over $i = 1, \ldots, n$ gives, with probability at least $1 - \delta$,

$$\max_{1 \leq i \leq n} |(\boldsymbol{V}_t)_i| \leq \sigma\sqrt{2t \log(2n/\delta)}.$$

Substituting into $\Delta_t \leq \frac{Rn}{t} \max_i |(\boldsymbol{V}_t)_i|$ yields (25). $\qquad \square$

**Corollary D.7.** *Under Assumption 2, we have $\Delta_t \to 0$ almost surely.*

*Proof.* Apply Lemma D.6 with $\delta_t = t^{-2}$ and use Borel–Cantelli. $\qquad \square$

D.2.1. FINITE STOPPING

**Lemma D.8.** *Fix $\boldsymbol{\theta} \in \Theta^{\mathrm{gap}}$. Under the conditions of Proposition 6.3,*

$$\Gamma(\hat{\boldsymbol{w}}_t^{\mathrm{emp}}; \boldsymbol{\theta}) \longrightarrow \Gamma^*(\boldsymbol{\theta}) \qquad \textit{almost surely.}$$

*Proof.* For each $t \geq 1$, we have $\Gamma(\hat{\boldsymbol{w}}_t^{\mathrm{emp}}; \boldsymbol{\theta}) \leq \Gamma^*(\boldsymbol{\theta})$ by definition. Let $\varepsilon_t$ denote the error term subtracted from $\Gamma^*(\boldsymbol{\theta})$ in Proposition 6.3, so $\varepsilon_t \to 0$ as $t \to \infty$. Choose any $p > 1$ and take $t$ large enough that Proposition 6.3 applies. Then

$$\mathbb{P}_{\boldsymbol{\theta}}\left(\Gamma(\hat{\boldsymbol{w}}_t^{\mathrm{emp}}; \boldsymbol{\theta}) \leq \Gamma^*(\boldsymbol{\theta}) - \varepsilon_t\right) \leq t^{-p}.$$

Since $\sum_{t \geq 1} t^{-p} < \infty$, Borel–Cantelli implies that almost surely,

$$\Gamma(\hat{\boldsymbol{w}}_t^{\mathrm{emp}}; \boldsymbol{\theta}) \geq \Gamma^*(\boldsymbol{\theta}) - \varepsilon_t \quad \text{for all sufficiently large } t.$$

Taking $\liminf$ and using $\varepsilon_t \to 0$ yields $\liminf_{t \to \infty} \Gamma(\hat{\boldsymbol{w}}_t^{\mathrm{emp}}; \boldsymbol{\theta}) \geq \Gamma^*(\boldsymbol{\theta})$. Combining with $\Gamma(\hat{\boldsymbol{w}}_t^{\mathrm{emp}}; \boldsymbol{\theta}) \leq \Gamma^*(\boldsymbol{\theta})$ gives almost sure convergence. $\qquad \square$

**Proposition D.9.** *Fix $\boldsymbol{\theta} \in \Theta^{\mathrm{gap}}$ and $\delta \in (0, 1)$. Let $\tau_\delta$ be the stopping time defined in* (5). *Then*

$$\mathbb{P}_{\boldsymbol{\theta}}(\tau_\delta < \infty) = 1.$$

*Proof.* By Corollary 6.2, almost surely there exists a finite time $t_{\mathrm{stab}}$ such that $\hat{S}_t = S^*(\boldsymbol{\theta})$ and $\hat{B}_t = B(\boldsymbol{\theta})$ for all $t \geq t_{\mathrm{stab}}$; in particular, for all $t \geq t_{\mathrm{stab}}$ the correctness condition of Lemma D.5*(b)* holds.

On this almost sure event, for all $t \geq t_{\mathrm{stab}}$, Lemma D.5*(b)* gives

$$\left|\frac{1}{t} Z_{\hat{S}_t}(t) - \Gamma(\hat{\boldsymbol{w}}_t^{\mathrm{emp}}; \boldsymbol{\theta})\right| \leq 2\Delta_t.$$

By Corollary D.7, $\Delta_t \to 0$ almost surely, and by Lemma D.8, $\Gamma(\hat{\boldsymbol{w}}_t^{\mathrm{emp}}; \boldsymbol{\theta}) \to \Gamma^*(\boldsymbol{\theta})$ almost surely. Thus,

$$\frac{1}{t} Z_{\hat{S}_t}(t) \longrightarrow \Gamma^*(\boldsymbol{\theta}) \qquad \text{almost surely.} \qquad (26)$$

We now compare $Z_{\hat{S}_t}(t)$ to the threshold. By Lemma D.4, for each fixed $\delta \in (0, 1)$ we have $\beta(t, \delta) \leq \bar{\beta}(t, \delta)$ for all $t$ and $\bar{\beta}(t, \delta)/t \to 0$ as $t \to \infty$.

Finally, we show that $\Gamma^*(\boldsymbol{\theta}) > 0$. For each boundary pair $(u, v) \in B(\boldsymbol{\theta})$ we have $\eta_{uv}(\boldsymbol{\theta}) = \theta_u - \theta_v > 0$, and any $\boldsymbol{\theta}' \in \Theta_{uv}$ has $\eta_{uv}(\boldsymbol{\theta}') \leq 0$. By continuity of $d_{uv}(\boldsymbol{\theta}, \boldsymbol{\theta}')$ and compactness of $\Theta_{uv}$, $\inf_{\boldsymbol{\theta}' \in \Theta_{uv}} d_{uv}(\boldsymbol{\theta}, \boldsymbol{\theta}') > 0$. Thus, $\Gamma^*(\boldsymbol{\theta}) > 0$.

So almost surely, for all sufficiently large $t$, $Z_{\hat{S}_t}(t) \geq \beta(t, \delta)$. Thus, $\tau_\delta < \infty$ almost surely. $\qquad \square$

# E. Proof of Theorem 6.5

*Proof.* Fix $\varepsilon \in (0, 1)$ and define

$$t_\delta := \left\lceil \frac{1 + \varepsilon}{\Gamma^*(\boldsymbol{\theta})} \log \frac{1}{\delta} \right\rceil, \qquad \varepsilon' := \frac{\varepsilon}{2(1 + \varepsilon)} \Gamma^*(\boldsymbol{\theta}) > 0.$$

By Lemma D.4, $\beta(t, \delta) \leq \bar{\beta}(t, \delta)$ for all $t$ and $t \mapsto \bar{\beta}(t, \delta)/t$ is decreasing. So for all $t \geq t_\delta$,

$$\frac{\beta(t, \delta)}{t} \leq \frac{\bar{\beta}(t, \delta)}{t} \leq \frac{\bar{\beta}(t_\delta, \delta)}{t_\delta},$$

where

$$\frac{\bar{\beta}(t_\delta, \delta)}{t_\delta} = \frac{\log(1/\delta)}{t_\delta} + \frac{1}{t_\delta}\left(\frac{\lambda}{2}nR^2\right) + \frac{(n-1)\log(1 + C_\beta t_\delta)}{2t_\delta}.$$

By definition of $t_\delta$,

$$\frac{\log(1/\delta)}{t_\delta} \leq \frac{\Gamma^*(\boldsymbol{\theta})}{1+\varepsilon}.$$

Since $t_\delta \to \infty$ as $\delta \to 0$, there exists $\delta_0 \in (0,1)$ such that for all $\delta \in (0, \delta_0)$,

$$\frac{1}{t_\delta}\left(\frac{\lambda}{2}nR^2\right) + \frac{n-1}{2} \cdot \frac{\log(1 + C_\beta t_\delta)}{t_\delta} \leq \varepsilon'.$$

Therefore, for all $\delta \in (0, \delta_0)$ and all $t \geq t_\delta$,

$$\frac{\beta(t, \delta)}{t} \leq \frac{\Gamma^*(\boldsymbol{\theta})}{1+\varepsilon} + \varepsilon' = \Gamma^*(\boldsymbol{\theta}) - \varepsilon'. \qquad (27)$$

Next, fix $p > 1$ and set $\delta_t := t^{-(p+2)}$. Take $\delta \in (0, \delta_0)$ and $t \geq t_\delta$. Since $\tau_\delta > t$ implies the stopping condition fails at time $t$,

$$\mathbb{P}_{\boldsymbol{\theta}}(\tau_\delta > t) \leq \mathbb{P}_{\boldsymbol{\theta}}\left(Z_{\hat{S}_t}(t) < \beta(t, \delta)\right)$$
$$= \mathbb{P}_{\boldsymbol{\theta}}\left(\frac{1}{t}Z_{\hat{S}_t}(t) \leq \frac{\beta(t, \delta)}{t}\right).$$

By (27), for all $t \geq t_\delta$ we have $\beta(t, \delta)/t \leq \Gamma^*(\boldsymbol{\theta}) - \varepsilon'$, hence

$$\mathbb{P}_{\boldsymbol{\theta}}(\tau_\delta > t) \leq \mathbb{P}_{\boldsymbol{\theta}}\left(\frac{1}{t}Z_{\hat{S}_t}(t) \leq \Gamma^*(\boldsymbol{\theta}) - \varepsilon'\right). \qquad (28)$$

Let $\Delta_k := \theta_{(k)} - \theta_{(k+1)} > 0$. Choose times $t_{\mathrm{bdry}}, t_\Delta, t_\Gamma < \infty$ such that for all $t$ large enough the following hold:

*(i) Boundary correctness.* Apply Lemma 6.1 at time $t$ with confidence $\delta_t$. Using Lemma F.1*(e)*, the deviation bound tends to 0 as $t \to \infty$, so there exists $t_{\mathrm{bdry}}$ such that for all $t \geq t_{\mathrm{bdry}}$,

$$\mathbb{P}_{\boldsymbol{\theta}}\left(\hat{S}_t \neq S^*(\boldsymbol{\theta})\right) \leq \delta_t.$$

*(ii) One-time control of $\Delta_t$.* By Lemma D.6 with confidence $\delta_t$, and since $R\sigma n\sqrt{2\log(2n/\delta_t)/t} = o(1)$, there exists $t_\Delta$ such that for all $t \geq t_\Delta$,

$$\mathbb{P}_{\boldsymbol{\theta}}\left(\Delta_t \geq \frac{\varepsilon'}{4}\right) \leq \delta_t.$$

*(iii) One-time lower bound for $\Gamma(\hat{\boldsymbol{w}}_t^{\mathrm{emp}}; \boldsymbol{\theta})$.* Apply Proposition 6.3 with exponent $p + 2$. Choose $t_\Gamma$ such that the error $\mathrm{err}_t \leq \varepsilon'/2$ for all $t \geq t_\Gamma$. Then for all $t \geq t_\Gamma$,

$$\mathbb{P}_{\boldsymbol{\theta}}\left(\Gamma(\hat{\boldsymbol{w}}_t^{\mathrm{emp}}; \boldsymbol{\theta}) \leq \Gamma^*(\boldsymbol{\theta}) - \frac{\varepsilon'}{2}\right) \leq \delta_t.$$

Set $t_* := \max\{t_{\mathrm{bdry}}, t_\Delta, t_\Gamma\}$ and fix $t \geq t_*$. On the event

$$E_t := \left\{\hat{S}_t = S^*(\boldsymbol{\theta})\right\}$$
$$\cap \left\{\Delta_t \leq \varepsilon'/4\right\} \cap \left\{\Gamma(\hat{\boldsymbol{w}}_t^{\mathrm{emp}}; \boldsymbol{\theta}) \geq \Gamma^*(\boldsymbol{\theta}) - \varepsilon'/2\right\},$$

Lemma D.5*(b)* applies at time $t$, so

$$\frac{1}{t}Z_{\hat{S}_t}(t) \geq \Gamma(\hat{\boldsymbol{w}}_t^{\mathrm{emp}}; \boldsymbol{\theta}) - 2\Delta_t$$
$$\geq \left(\Gamma^*(\boldsymbol{\theta}) - \frac{\varepsilon'}{2}\right) - 2 \cdot \frac{\varepsilon'}{4}$$
$$= \Gamma^*(\boldsymbol{\theta}) - \varepsilon'.$$

Therefore, for all $t \geq t_*$,

$$\mathbb{P}_{\boldsymbol{\theta}}\left(\frac{1}{t}Z_{\hat{S}_t}(t) \leq \Gamma^*(\boldsymbol{\theta}) - \varepsilon'\right)$$
$$\leq \mathbb{P}_{\boldsymbol{\theta}}(E_t^c)$$
$$\leq \mathbb{P}_{\boldsymbol{\theta}}\left(\hat{S}_t \neq S^*(\boldsymbol{\theta})\right) + \mathbb{P}_{\boldsymbol{\theta}}\left(\Delta_t \geq \frac{\varepsilon'}{4}\right)$$
$$+ \mathbb{P}_{\boldsymbol{\theta}}\left(\Gamma(\hat{\boldsymbol{w}}_t^{\mathrm{emp}}; \boldsymbol{\theta}) \leq \Gamma^*(\boldsymbol{\theta}) - \frac{\varepsilon'}{2}\right)$$
$$\leq 3\delta_t = 3t^{-(p+2)}.$$

Combining with (28), we obtain that for all $\delta \in (0, \delta_0)$ and all $t \geq \max\{t_\delta, t_*\}$,

$$\mathbb{P}_{\boldsymbol{\theta}}(\tau_\delta > t) \leq 3t^{-(p+2)}.$$

Define $T_\delta := \max\{t_\delta, t_*\}$.

$$\mathbb{E}_{\boldsymbol{\theta}}[\tau_\delta] = \sum_{t=0}^{\infty}\mathbb{P}_{\boldsymbol{\theta}}(\tau_\delta > t) \leq T_\delta + \sum_{t=T_\delta}^{\infty}\mathbb{P}_{\boldsymbol{\theta}}(\tau_\delta > t)$$
$$\leq T_\delta + 3\sum_{t=T_\delta}^{\infty}t^{-(p+2)}$$
$$\leq T_\delta + \frac{3}{p+1}(T_\delta - 1)^{-(p+1)}.$$

Divide by $\log(1/\delta)$ and take $\limsup_{\delta \to 0}$. Since $t_\delta \to \infty$ as $\delta \to 0$ and $t_*$ is fixed, $T_\delta = t_\delta$ for all sufficiently small $\delta$, and the second term goes to 0 after division by $\log(1/\delta)$. Therefore,

$$\limsup_{\delta \to 0}\frac{\mathbb{E}_{\boldsymbol{\theta}}[\tau_\delta]}{\log(1/\delta)} \leq \limsup_{\delta \to 0}\frac{t_\delta}{\log(1/\delta)} = \frac{1+\varepsilon}{\Gamma^*(\boldsymbol{\theta})}.$$

Finally, let $\varepsilon \downarrow 0$.

$\square$

# F. Auxiliary Lemmas

## F.1. C-tracking

**C-tracking lemma (Garivier–Kaufmann).** Let $\boldsymbol{p}(1), \ldots, \boldsymbol{p}(T) \in \Delta_{\mathcal{P}}$. Define $\Lambda(k) := \sum_{s=1}^{k} \boldsymbol{p}(s)$ and $N(0) = 0$. If for each $k \in \{0, \ldots, T-1\}$,

$$I_{k+1} \in \operatorname*{argmax}_{(i,j) \in \mathcal{P}} \big( \Lambda_{ij}(k+1) - N_{ij}(k) \big),$$
$$N(k+1) := N(k) + \delta_{I_{k+1}},$$

then

$$\max_{(i,j) \in \mathcal{P}} \big| N_{ij}(T) - \Lambda_{ij}(T) \big| \leq |\mathcal{P}| - 1.$$

**Lemma F.1.** *Run C-tracking with targets $\boldsymbol{p}(t) = \tilde{\boldsymbol{w}}_t$. For $t \geq 1$, define*

$$\bar{\tilde{\boldsymbol{w}}}_t := \frac{1}{t} \sum_{s=1}^{t} \tilde{\boldsymbol{w}}_s, \qquad \bar{\boldsymbol{w}}_t := \frac{1}{t} \sum_{s=1}^{t} \boldsymbol{w}_s.$$

*Then:*

*(a)* $\displaystyle \max_{(i,j) \in \mathcal{P}} \big| N_{ij}(t) - P_{ij}(t) \big| \leq |\mathcal{P}| - 1.$

*(b)* $\displaystyle \big\| \hat{\boldsymbol{w}}_t^{\mathrm{emp}} - \bar{\tilde{\boldsymbol{w}}}_t \big\|_{\infty} \leq \frac{|\mathcal{P}| - 1}{t}.$

*(c)* $\displaystyle \big\| \hat{\boldsymbol{w}}_t^{\mathrm{emp}} - \bar{\boldsymbol{w}}_t \big\|_{\infty} \leq \frac{|\mathcal{P}| - 1}{t} + \frac{1}{t} \sum_{s=1}^{t} \rho_s.$

*(d) for every $(i,j) \in \mathcal{P}$, $N_{ij}(t) \geq \frac{1}{|\mathcal{P}|} \sum_{s=1}^{t} \rho_s - (|\mathcal{P}| - 1)$*

*(e) If $\rho_t = t^{-\gamma}$ for some $\gamma \in (0,1)$, then for all $t \geq 1$,*

$$N_{\min}(t) \geq \frac{1}{|\mathcal{P}|} \cdot \frac{(t+1)^{1-\gamma} - 1}{1 - \gamma} - (|\mathcal{P}| - 1).$$

*In particular, $N_{\min}(t) / \log t \to \infty$.*

*Proof.* (a) Apply the Garivier–Kaufmann C-tracking lemma stated above with $T = t$ and $\boldsymbol{p}(s) = \tilde{\boldsymbol{w}}_s$. Then $P_{ij}(t) = \Lambda_{ij}(t)$ and $N_{ij}(t) = N_{ij}(t)$, so $\max_{(i,j)} |N_{ij}(t) - P_{ij}(t)| \leq |\mathcal{P}| - 1$.

(b) Divide (a) by $t$ and use $\bar{\tilde{w}}_{t,ij} = P_{ij}(t) / t$.

(c)

$$\bar{\tilde{\boldsymbol{w}}}_t - \bar{\boldsymbol{w}}_t = \frac{1}{t} \sum_{s=1}^{t} \rho_s (\boldsymbol{u} - \boldsymbol{w}_s),$$

and since $\boldsymbol{u}, \boldsymbol{w}_s \in \Delta_{\mathcal{P}}$ we have $\|\boldsymbol{u} - \boldsymbol{w}_s\|_{\infty} \leq 1$, hence $\|\bar{\tilde{\boldsymbol{w}}}_t - \bar{\boldsymbol{w}}_t\|_{\infty} \leq \frac{1}{t} \sum_{s=1}^{t} \rho_s$. Combine with (b) and the triangle inequality.

(d) From (a), $N_{ij}(t) \geq P_{ij}(t) - (|\mathcal{P}| - 1)$. Also $\tilde{w}_{s,ij} = (1 - \rho_s) w_{s,ij} + \rho_s u_{ij} \geq \rho_s u_{ij} = \rho_s / |\mathcal{P}|$, hence $P_{ij}(t) = \sum_{s=1}^{t} \tilde{w}_{s,ij} \geq \frac{1}{|\mathcal{P}|} \sum_{s=1}^{t} \rho_s$.

(e) Follows from (d) and Fact F.2. $\qquad\square$

**Fact F.2.** *Fix $r \in (0,1)$. For every integer $t \geq 1$,*

$$\frac{(t+1)^{1-r} - 1}{1 - r} \leq \sum_{s=1}^{t} s^{-r} \leq 1 + \frac{t^{1-r} - 1}{1 - r}.$$

*And, for any integers $1 \leq a \leq b$,*

$$\sum_{s=a}^{b} s^{-r} \leq 1 + \frac{b^{1-r} - (a-1)^{1-r}}{1 - r} \leq 1 + \frac{b^{1-r}}{1 - r}.$$

## F.2. Supporting Lemmas for the Proof of Proposition 6.3

**Lemma F.3.** *Define*

$$D_{\max} := \max_{(i,j) \in \mathcal{P}} \sup_{\boldsymbol{\vartheta}, \boldsymbol{\vartheta}' \in \Theta} d_{ij}(\boldsymbol{\vartheta}, \boldsymbol{\vartheta}'), \quad L := 4R \bar{\sigma}^2 \sqrt{2}.$$

*Then for any nonempty $B \subseteq \mathcal{P}$, any $\boldsymbol{w} \in \Delta_{\mathcal{P}}$, any $\boldsymbol{q} \in \Delta_B$, and any $\boldsymbol{\vartheta} \in \Theta$,*

$$0 \leq \gamma_{ij}(\boldsymbol{w}; \boldsymbol{\vartheta}) \leq D_{\max}, \qquad 0 \leq F_{\boldsymbol{\vartheta}}(\boldsymbol{w}, \boldsymbol{q}) \leq D_{\max}.$$

*Additionally, for all $\boldsymbol{\vartheta}, \boldsymbol{\vartheta}' \in \Theta$,*

$$\sup_{\boldsymbol{w} \in \Delta_{\mathcal{P}}, \, \boldsymbol{q} \in \Delta_B} \big| F_{\boldsymbol{\vartheta}}(\boldsymbol{w}, \boldsymbol{q}) - F_{\boldsymbol{\vartheta}'}(\boldsymbol{w}, \boldsymbol{q}) \big| \leq L \|\boldsymbol{\vartheta} - \boldsymbol{\vartheta}'\|_2.$$

*Proof.* Fix $\boldsymbol{\vartheta}, \boldsymbol{\vartheta}' \in \Theta$, $(a,b) \in \mathcal{P}$, and $\boldsymbol{\theta}'' \in \Theta$, and write $\eta = \eta_{ab}(\boldsymbol{\vartheta})$, $\eta' = \eta_{ab}(\boldsymbol{\vartheta}')$, and $\nu = \eta_{ab}(\boldsymbol{\theta}'')$. Define $g(u) := d(u, \nu)$, where $d(u, \nu) = A(\nu) - A(u) - A'(u)(\nu - u)$. Then $g'(u) = A''(u)(u - \nu)$. Since $|u|, |\nu| \leq 2R$ for $u = \eta_{ab}(\cdot)$ on $\Theta$, we have $|u - \nu| \leq 4R$ and thus $|g'(u)| \leq 4R \bar{\sigma}^2$ for all $|u| \leq 2R$. By the mean value theorem,

$$\begin{aligned} |d_{ab}(\boldsymbol{\vartheta}, \boldsymbol{\theta}'') - d_{ab}(\boldsymbol{\vartheta}', \boldsymbol{\theta}'')| &= |d(\eta, \nu) - d(\eta', \nu)| \\ &\leq 4R \bar{\sigma}^2 \, |\eta - \eta'| \\ &\leq 4R \bar{\sigma}^2 \, \|\boldsymbol{x}_{ab}\|_2 \, \|\boldsymbol{\vartheta} - \boldsymbol{\vartheta}'\|_2 \\ &= 4R \bar{\sigma}^2 \sqrt{2} \, \|\boldsymbol{\vartheta} - \boldsymbol{\vartheta}'\|_2. \end{aligned}$$

Averaging over $\boldsymbol{w}$ and using $|\inf f - \inf g| \leq \sup |f - g|$ yields, for each $(i,j) \in B$,

$$\sup_{\boldsymbol{w} \in \Delta_{\mathcal{P}}} |\gamma_{ij}(\boldsymbol{w}; \boldsymbol{\vartheta}) - \gamma_{ij}(\boldsymbol{w}; \boldsymbol{\vartheta}')| \leq 4R \bar{\sigma}^2 \sqrt{2} \, \|\boldsymbol{\vartheta} - \boldsymbol{\vartheta}'\|_2.$$

Since $F_{\boldsymbol{\vartheta}}(\boldsymbol{w}, \boldsymbol{q}) = \sum_{(i,j) \in B} q_{ij} \gamma_{ij}(\boldsymbol{w}; \boldsymbol{\vartheta})$ with $\boldsymbol{q} \in \Delta_B$, we obtain

$$\sup_{\boldsymbol{w}, \boldsymbol{q}} \big| F_{\boldsymbol{\vartheta}}(\boldsymbol{w}, \boldsymbol{q}) - F_{\boldsymbol{\vartheta}'}(\boldsymbol{w}, \boldsymbol{q}) \big| \leq 4R \bar{\sigma}^2 \sqrt{2} \, \|\boldsymbol{\vartheta} - \boldsymbol{\vartheta}'\|_2.$$

Finally, $0 \leq d_{ij}(\boldsymbol{\vartheta}, \boldsymbol{\vartheta}') \leq D_{\max}$ for all $(i,j), \boldsymbol{\vartheta}, \boldsymbol{\vartheta}'$ by definition, hence $0 \leq D_{\boldsymbol{w}}(\boldsymbol{\vartheta} \| \boldsymbol{\vartheta}') \leq D_{\max}$ and therefore $0 \leq \gamma_{ij}(\boldsymbol{w}; \boldsymbol{\vartheta}) \leq D_{\max}$ and $0 \leq F_{\boldsymbol{\vartheta}}(\boldsymbol{w}, \boldsymbol{q}) \leq D_{\max}$. $\qquad\square$

Lemma F.4 is a known result with slight modification for our setting; see, e.g., Shalev-Shwartz (2012); Hazan (2016); Arora et al. (2012). We provide the full proof here.

**Lemma F.4** (entropic-FTRL bound). *Fix $T \geq 1$ and define $b_T := \lceil T^{1/4} \rceil$. Let $(\hat{\boldsymbol{g}}_t)_{t=1}^T$ be any sequence in $\mathbb{R}^{\mathcal{P}}$ with $\|\hat{\boldsymbol{g}}_t\|_\infty \leq G$ for all $t$. Let $(\mu_t)_{t \geq 1}$ be nonincreasing. Define cumulative scores $\boldsymbol{\Psi}_0 = \boldsymbol{0}$ and $\boldsymbol{\Psi}_t = \boldsymbol{\Psi}_{t-1} + \hat{\boldsymbol{g}}_t$. Define the exponential-weights / FTRL updates*

$$\boldsymbol{w}_t = \frac{\boldsymbol{w}_1 \odot \exp(\mu_t \boldsymbol{\Psi}_{t-1})}{\langle \boldsymbol{w}_1, \exp(\mu_t \boldsymbol{\Psi}_{t-1}) \rangle}, \qquad t \geq 1,$$

*where $\boldsymbol{w}_1 \in \Delta_{\mathcal{P}}$ has full support. Then*

$$\sup_{\boldsymbol{w} \in \Delta_{\mathcal{P}}} \sum_{t=b_T}^T \langle \hat{\boldsymbol{g}}_t, \boldsymbol{w} - \boldsymbol{w}_t \rangle \leq \frac{D_{\boldsymbol{w}}(1)}{\mu_T} + \frac{G^2}{2} \sum_{t=1}^T \mu_t + 2G(b_T - 1),$$

*Proof.* For $t \geq 0$ and $\eta > 0$ define

$$Z_t(\eta) := \sum_{(a,b) \in \mathcal{P}} w_{1,ab} \exp(\eta \Psi_{t,ab}), \quad \Phi_t := \frac{1}{\mu_t} \log Z_t(\mu_t).$$

We first show that for each $t \in \{1, \ldots, T\}$,

$$\Phi_t - \Phi_{t-1} \leq \langle \boldsymbol{w}_t, \hat{\boldsymbol{g}}_t \rangle + \frac{\mu_t G^2}{2}.$$

Since $(\mu_t)$ is nonincreasing, $\mu_t \leq \mu_{t-1}$ and $x \mapsto x^{\mu_t/\mu_{t-1}}$ is concave on $\mathbb{R}_+$, so Jensen gives

$$Z_{t-1}(\mu_t) = \sum_{(a,b)} w_{1,ab} \Big( \exp(\mu_{t-1} \Psi_{t-1,ab}) \Big)^{\mu_t/\mu_{t-1}}$$

$$\leq Z_{t-1}(\mu_{t-1})^{\mu_t/\mu_{t-1}}.$$

Thus

$$\frac{1}{\mu_t} \log Z_{t-1}(\mu_t) \leq \frac{1}{\mu_{t-1}} \log Z_{t-1}(\mu_{t-1}) = \Phi_{t-1}.$$

Also, since $\boldsymbol{\Psi}_t = \boldsymbol{\Psi}_{t-1} + \hat{\boldsymbol{g}}_t$,

$$\frac{Z_t(\mu_t)}{Z_{t-1}(\mu_t)} = \mathbb{E}_{X \sim \boldsymbol{w}_t} \Big[ \exp(\mu_t \hat{g}_{t,X}) \Big],$$

so

$$\Phi_t - \frac{1}{\mu_t} \log Z_{t-1}(\mu_t) = \frac{1}{\mu_t} \log \mathbb{E}_{X \sim \boldsymbol{w}_t} \Big[ \exp(\mu_t \hat{g}_{t,X}) \Big].$$

Let $Y := \hat{g}_{t,X} \in [-G, G]$ a.s. Define $\psi(\lambda) := \log \mathbb{E}[\exp(\lambda Y)]$. Then $\psi'(0) = \mathbb{E}[Y]$ and $\psi''(\lambda) = \text{Var}_\lambda(Y) \leq G^2$ for all $\lambda$, hence

$$\frac{1}{\mu_t} \log \mathbb{E}[\exp(\mu_t Y)] \leq \mathbb{E}[Y] + \frac{\mu_t G^2}{2} = \langle \boldsymbol{w}_t, \hat{\boldsymbol{g}}_t \rangle + \frac{\mu_t G^2}{2}.$$

Combining gives the bound. Summing from $t = 1$ to $T$ gives

$$\Phi_T \leq \sum_{t=1}^T \langle \boldsymbol{w}_t, \hat{\boldsymbol{g}}_t \rangle + \frac{G^2}{2} \sum_{t=1}^T \mu_t.$$

Now take any $\boldsymbol{w} \in \Delta_{\mathcal{P}}$. Nonnegativity of KL divergence gives

$$\langle \boldsymbol{w}, \boldsymbol{\Psi}_T \rangle \leq \Phi_T + \frac{1}{\mu_T} \text{KL}(\boldsymbol{w} \| \boldsymbol{w}_1),$$

so

$$\sum_{t=1}^T \langle \hat{\boldsymbol{g}}_t, \boldsymbol{w} - \boldsymbol{w}_t \rangle = \langle \boldsymbol{w}, \boldsymbol{\Psi}_T \rangle - \sum_{t=1}^T \langle \boldsymbol{w}_t, \hat{\boldsymbol{g}}_t \rangle$$

$$\leq \frac{1}{\mu_T} \text{KL}(\boldsymbol{w} \| \boldsymbol{w}_1) + \frac{G^2}{2} \sum_{t=1}^T \mu_t.$$

Taking $\sup_{\boldsymbol{w}}$ yields

$$\sup_{\boldsymbol{w} \in \Delta_{\mathcal{P}}} \sum_{t=1}^T \langle \hat{\boldsymbol{g}}_t, \boldsymbol{w} - \boldsymbol{w}_t \rangle \leq \frac{D_{\boldsymbol{w}}(1)}{\mu_T} + \frac{G^2}{2} \sum_{t=1}^T \mu_t.$$

Finally, for each $t$, $|\langle \hat{\boldsymbol{g}}_t, \boldsymbol{w} - \boldsymbol{w}_t \rangle| \leq \|\hat{\boldsymbol{g}}_t\|_\infty \|\boldsymbol{w} - \boldsymbol{w}_t\|_1 \leq 2G$, so removing the first $(b_T - 1)$ terms gives

$$\sup_{\boldsymbol{w} \in \Delta_{\mathcal{P}}} \sum_{t=b_T}^T \langle \hat{\boldsymbol{g}}_t, \boldsymbol{w} - \boldsymbol{w}_t \rangle \leq \frac{D_{\boldsymbol{w}}(1)}{\mu_T} + \frac{G^2}{2} \sum_{t=1}^T \mu_t + 2G(b_T - 1).$$

$\square$

# G. Additional Discussion

This section contains three supplemental discussions. Subsection G.1 derives how quickly the stopping time converges to the lower bound as $\delta \to 0$. Subsection G.2 explains where the boundedness of the parameter space is used in the analysis. Subsection G.3 discusses the choice of the learning-rate exponent $\alpha$.

### G.1. Rate of Convergence

Here, we derive an upper bound on the convergence error of $\frac{\mathbb{E}_{\boldsymbol{\theta}}[\tau_\delta]}{\log(1/\delta)}$ to $\frac{1}{\Gamma^*(\boldsymbol{\theta})}$ as $\delta \to 0$.

**Proposition G.1** (Refinement of Theorem 6.5). *Let*

$$\rho := \min \left\{ \alpha, \, 1 - \alpha, \, \gamma, \, \frac{1-\gamma}{2} \right\}.$$

*Fix $\boldsymbol{\theta} \in \Theta^{\text{gap}}$ and let $\tau_\delta$ be the stopping time of Algorithm 1. Then there exist constants $C_{\boldsymbol{\theta}} < \infty$ and $\delta_0(\boldsymbol{\theta}) \in (0,1)$ such that, for all $\delta < \delta_0(\boldsymbol{\theta})$,*

$$\mathbb{E}_{\boldsymbol{\theta}}[\tau_\delta] \leq \frac{\log(1/\delta)}{\Gamma^*(\boldsymbol{\theta})} + C_{\boldsymbol{\theta}} (\log(1/\delta))^{1-\rho} \sqrt{\log \log(1/\delta)}.$$

*Hence,*

$$\frac{\mathbb{E}_{\boldsymbol{\theta}}[\tau_\delta]}{\log(1/\delta)} \leq \frac{1}{\Gamma^*(\boldsymbol{\theta})} + \widetilde{O}\big((\log(1/\delta))^{-\rho}\big).$$

*Proof.* We follow the proof of Theorem 6.5 in Appendix E. The only change is that the fixed slack $\varepsilon'$ used there is replaced by the explicit rate obtained from Proposition 6.3.

Apply Proposition 6.3 with $p = 4$, Lemma D.6 with confidence level $t^{-4}$, and Lemma 6.1 with confidence level $t^{-4}$. As in the proof of Theorem 6.5, Lemma 6.1 together with Lemma F.1*(e)* gives correctness of the estimated boundary constraints for all sufficiently large $t$ with probability at least $1 - t^{-4}$. Additionally, all error terms in Proposition 6.3 and Lemma D.6 are bounded, for all sufficiently large $t$, by a constant multiple of $t^{-\rho}\sqrt{\log t}$. Therefore, by the same union-bound argument over the three events used in Appendix E, and by the same application of Lemma D.5*(b)*, there exist constants $C_1 < \infty$ and $t^*(\boldsymbol{\theta}) < \infty$ such that, for all $t \geq t^*(\boldsymbol{\theta})$,

$$\mathbb{P}_{\boldsymbol{\theta}}\left(\frac{1}{t}Z_{\hat{S}_t}(t) \leq \Gamma^*(\boldsymbol{\theta}) - C_1 t^{-\rho}\sqrt{\log t}\right) \leq 3t^{-4}. \quad (29)$$

We now define the deterministic cutoff $t_\delta$, the candidate upper bound for the stopping time. We show that after this time, the probability that the algorithm has not stopped is at most $3t^{-4}$. First, choose $M > 0$ large enough so that

$$\frac{(\Gamma^*(\boldsymbol{\theta}))^2 M}{4} > 2C_1(\Gamma^*(\boldsymbol{\theta}))^\rho.$$

Define

$$t_\delta = \left\lceil \frac{\log(1/\delta)}{\Gamma^*(\boldsymbol{\theta})} + M(\log(1/\delta))^{1-\rho}\sqrt{\log\log(1/\delta)} \right\rceil.$$

By Lemma D.4, $\beta(t, \delta) \leq \bar{\beta}(t, \delta)$, where

$$\bar{\beta}(t, \delta) = \log(1/\delta) + \frac{\lambda}{2}nR^2 + \frac{n-1}{2}\log(1 + C_\beta t),$$

and, as used in Appendix E, $t \mapsto \bar{\beta}(t, \delta)/t$ is decreasing. Since $t_\delta \asymp \log(1/\delta)$,

$$\bar{\beta}(t_\delta, \delta) = \log(1/\delta) + O(\log\log(1/\delta)).$$

Also,

$$\log\log(1/\delta) = o\left((\log(1/\delta))^{1-\rho}\sqrt{\log\log(1/\delta)}\right).$$

Thus, for all sufficiently small $\delta$,

$$\Gamma^*(\boldsymbol{\theta})t_\delta - \bar{\beta}(t_\delta, \delta) \geq \frac{\Gamma^*(\boldsymbol{\theta})M}{2}(\log(1/\delta))^{1-\rho}\sqrt{\log\log(1/\delta)}.$$

Moreover, for all sufficiently small $\delta$,

$$t_\delta \leq \frac{2\log(1/\delta)}{\Gamma^*(\boldsymbol{\theta})}.$$

Thus dividing by $t_\delta$ gives

$$\Gamma^*(\boldsymbol{\theta}) - \frac{\bar{\beta}(t_\delta, \delta)}{t_\delta} \geq \frac{(\Gamma^*(\boldsymbol{\theta}))^2 M}{4}(\log(1/\delta))^{-\rho}\sqrt{\log\log(1/\delta)}.$$

Set

$$c := \frac{(\Gamma^*(\boldsymbol{\theta}))^2 M}{4}.$$

Since $\beta(t, \delta) \leq \bar{\beta}(t, \delta)$ and $\bar{\beta}(t, \delta)/t$ is decreasing, for every $t \geq t_\delta$,

$$\frac{\beta(t, \delta)}{t} \leq \Gamma^*(\boldsymbol{\theta}) - c(\log(1/\delta))^{-\rho}\sqrt{\log\log(1/\delta)}. \quad (30)$$

Since $t \mapsto t^{-\rho}\sqrt{\log t}$ is eventually decreasing and $t_\delta \to \infty$ as $\delta \to 0$, we choose $\delta_0(\boldsymbol{\theta})$ small enough so that this monotonicity holds on $[t_\delta, \infty)$ and $t_\delta \geq t^*(\boldsymbol{\theta})$. Hence, for all $\delta < \delta_0(\boldsymbol{\theta})$ and all $t \geq t_\delta$,

$$C_1 t^{-\rho}\sqrt{\log t} \leq C_1 t_\delta^{-\rho}\sqrt{\log t_\delta}$$
$$\leq 2C_1(\Gamma^*(\boldsymbol{\theta}))^\rho (\log(1/\delta))^{-\rho}\sqrt{\log\log(1/\delta)}.$$

Define $c' := 2C_1(\Gamma^*(\boldsymbol{\theta}))^\rho$. By the choice of $M$, $c > c'$. Therefore, combining the high-probability GLR lower bound (29) with the threshold bound (30), and using the same argument as in Appendix E, we get

$$\mathbb{P}_{\boldsymbol{\theta}}(\tau_\delta > t) \leq 3t^{-4}, \qquad t \geq t_\delta.$$

Finally, by the same tail-sum argument as in the proof of Theorem 6.5,

$$\mathbb{E}_{\boldsymbol{\theta}}[\tau_\delta] \leq t_\delta + 3\sum_{t=t_\delta}^{\infty} t^{-4} = t_\delta + O(t_\delta^{-3}).$$

Substituting the definition of $t_\delta$ and absorbing the ceiling, the tail term, and $M$ into $C_{\boldsymbol{\theta}} < \infty$ gives

$$\mathbb{E}_{\boldsymbol{\theta}}[\tau_\delta] \leq \frac{\log(1/\delta)}{\Gamma^*(\boldsymbol{\theta})} + C_{\boldsymbol{\theta}}(\log(1/\delta))^{1-\rho}\sqrt{\log\log(1/\delta)}.$$

Thus,

$$\frac{\mathbb{E}_{\boldsymbol{\theta}}[\tau_\delta]}{\log(1/\delta)} \leq \frac{1}{\Gamma^*(\boldsymbol{\theta})} + C_{\boldsymbol{\theta}}(\log(1/\delta))^{-\rho}\sqrt{\log\log(1/\delta)}.$$

Equivalently,

$$\frac{\mathbb{E}_{\boldsymbol{\theta}}[\tau_\delta]}{\log(1/\delta)} \leq \frac{1}{\Gamma^*(\boldsymbol{\theta})} + \widetilde{O}\left((\log(1/\delta))^{-\rho}\right).$$

$\square$

### G.2. Bounded Parameter Space Assumption

Here, we discuss where the boundedness assumption is used. First, since every pairwise difference $\theta_i - \theta_j$ lies in $[-2R, 2R]$, we get the bound

$$a = \inf_{|\eta| \leq 2R} A''(\eta) > 0,$$

which is used for the MLE concentration argument. Second, bounded $\Theta$ makes $D_{\max}$ finite, which is the Lipschitz constant that controls how much $\Gamma(\boldsymbol{w}; \boldsymbol{\theta})$ can change with changes to the allocation $\boldsymbol{w}$. Third, it makes $L$ finite, which appears in

$$\Delta F_s := \sup_{\boldsymbol{w}, \boldsymbol{q}} \left| F_{\hat{\boldsymbol{\theta}}_s}(\boldsymbol{w}, \boldsymbol{q}) - F_{\boldsymbol{\theta}}(\boldsymbol{w}, \boldsymbol{q}) \right| \leq L \|\hat{\boldsymbol{\theta}}_s - \boldsymbol{\theta}\|_2,$$

used to map estimation error to the error in the achieved information rate. Finally, in the stopping proof, boundedness gives us

$$\Delta_t = \sup_{\boldsymbol{\theta}' \in \Theta} \left| \bar{L}_t(\boldsymbol{\theta}') - L_t(\boldsymbol{\theta}') \right|$$

finite and tending to zero, and

$$\frac{\lambda}{2} \|\hat{\boldsymbol{\theta}}_t\|_2^2 \leq \frac{\lambda}{2} n R^2,$$

which allows us to conclude that the stopping threshold grows logarithmically.

Thus, the boundedness assumption is primarily used as a tool in the proofs (also a common assumption in linear bandits), but we do not view it as particularly restrictive in practice. For example, under the Bradley–Terry model, if $R = 5$, then $P(i \succ j)$ can be as high as $0.99995$. Thus in most settings, it is likely that any practically relevant utilities lie within such a bounded range.

### G.3. Hyperparameter $\alpha$ Tuning

Here we discuss what is meant by "the early gradient directions can be noisy" and the tuning of $\alpha$.

There are two mechanisms contributing to this "noise". The first is the stochastic gradient approximation itself. The full $\boldsymbol{w}$-gradient averages over all boundary pairs in $B(\boldsymbol{\theta})$, whereas the update in Algorithm 1 uses only the sampled pair $I_t$ and the corresponding alternative $\boldsymbol{\theta}_t^*$. Thus, early on, depending on which pair is sampled, the update can put too much weight on comparisons that are especially informative for ruling out that one sampled inversion, rather than on those that matter most for the full gradient. For example, if $n = 4$ and $k = 2$, the most challenging boundary pair is to differentiate ranks 2 and 3, but we may draw ranks $(1, 4)$ several times early on. The resulting updates overweight comparisons useful for ruling out item 4 overtaking item 1, which causes the cumulative scores $\boldsymbol{\Psi}_t^{(w)}$ and $\boldsymbol{\Psi}_t^{(q)}$ to be biased early. Hence, if $\mu_t = t^{-\alpha}$ decays too quickly, later updates will take a long time to undo this bias; if it decays too slowly, then the iterates remain overly sensitive to one-step noise. This is only an issue in early rounds, since later, non-bottleneck boundary pairs receive very little mass under $\boldsymbol{q}_t$, so even if such a pair is sampled, its effect on the $\boldsymbol{w}$-update is small. This is the mechanism that leads to improved performance of the oracle problem, where $\boldsymbol{\theta}$ is known, when $\alpha$ is moderately small (smaller than .5).

When $\boldsymbol{\theta}$ is not known, there is a second mechanism that arises since the gradients are evaluated at $\hat{\boldsymbol{\theta}}_t$ rather than at $\boldsymbol{\theta}$. Even if we were to compute the full gradient, it could still be misaligned early, because the estimated boundary set and the associated $\boldsymbol{\theta}_{ij}^*$ may be off. When tuning $\alpha$, we therefore consider both its effect on the oracle problem and on the full problem with estimation error. Empirically, these choices are quite similar. The slower decay that helps mitigate the issue of using a stochastic gradient also gives the MLE and the estimated boundary set time to stabilize.

Figures 3 and 4 show the mean stopping time over 100 simulations of the online oracle[3] and Algorithm 1, on the Equally Spaced instance with $\delta = 0.01$. Figure 3 fixes $n = 50$, $\mathrm{gap} = 0.25$ and varies $k$; Figure 4 fixes $k = 5$ and varies $n$ and $\mathrm{gap}$. The plots demonstrate that the choice of $\alpha$ is important; values that are too large or too small can result in significantly worse performance. However, as illustrated in both figures, the algorithm generally performs well for $\alpha \in [0.15, 0.3]$, and this holds robustly for other configurations as well.

---

[3]This is different from the oracle of the plots in Section 7. Here the oracle knows $\boldsymbol{\theta}$, but still learns $\boldsymbol{w}^*$ online, so its only advantage is the lack of noise in the MLE.

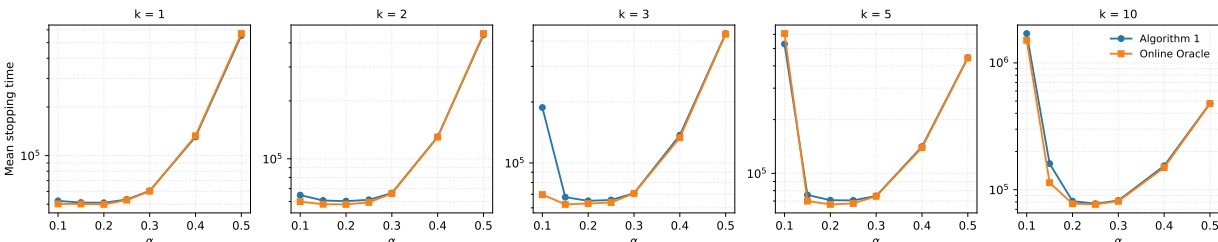

*Figure 3.* Mean stopping time over 100 simulations as a function of the learning-rate exponent $\alpha$ for varying $k$, on the Equally Spaced instance with $n = 50$, gap $= 0.25$, and $\delta = 0.01$. Each panel compares Algorithm 1 (blue) to the Online Oracle (orange) at $k \in \{1, 2, 3, 5, 10\}$.

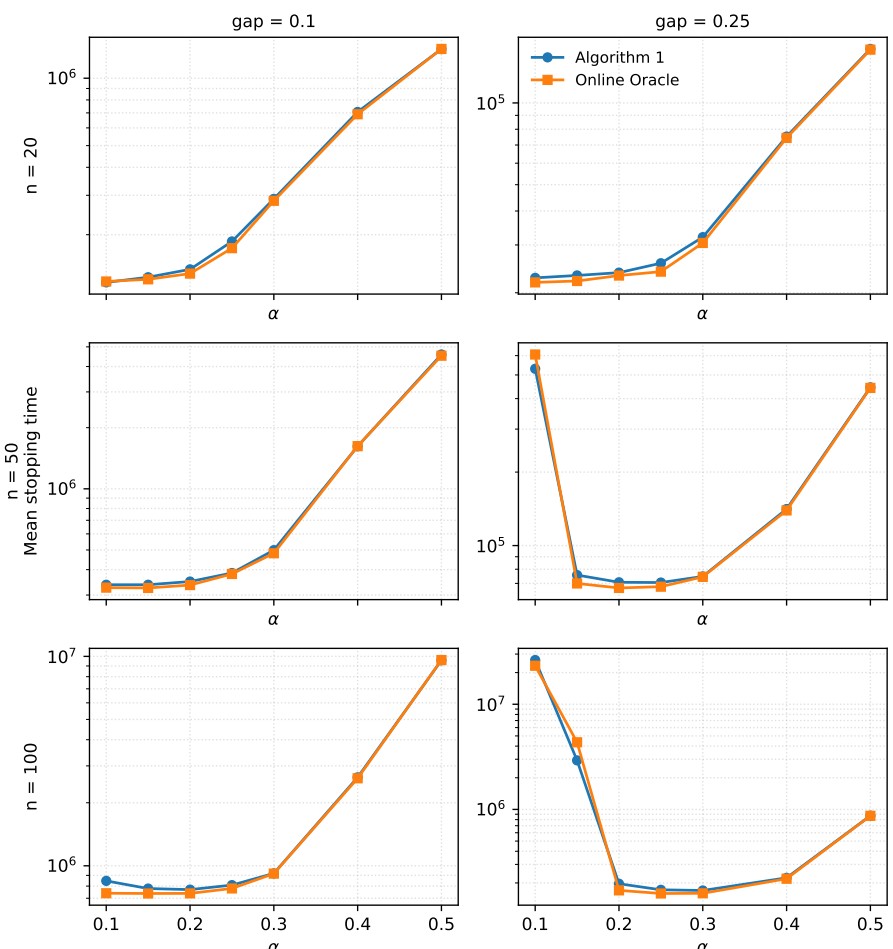

*Figure 4.* Mean stopping time over 100 simulations as a function of the learning-rate exponent $\alpha$, on the Equally Spaced instance with $k = 5$ and $\delta = 0.01$. Rows vary $n \in \{20, 50, 100\}$; columns vary gap $\in \{0.1, 0.25\}$. Algorithm 1 (blue) versus Online Oracle (orange).

