# OpenReview forum: "Optimal Top-$k$ Identification from Pairwise Comparisons"
_ICML.cc/2026/Conference — ICML 2026 regular_

### Official Review · Reviewer_opv8 · 2026-03-08

**Soundness:** 4
**Presentation:** 3
**Significance:** 4
**Originality:** 3
**Overall Recommendation:** 5
**Confidence:** 3

**Summary:**

This paper studies "top-k identification" with pairwise comparisons in a latent-utility model.  More carefully, there are $n$ items $[n]$, and item $i$ has true value $\theta_i$.  We are allowed to query pairs of items, and when we query some pair $(i,j)$ we get back some random value drawn from a distribution that is a one-parameter exponential family with parameter $\theta_i - \theta_j$.  For example, we could get binary information back ("item $i$ is better tha item $j$), where the probability of this is determined by how much better $i$ is than $j$.  Or we might get back some noisy measurement of how much better $i$ is than $j$.  Our goal is to make as few queries as possible so that we can actually identify the $k$ best items with probability at least $1-\delta$.  The main result is an algorithm for this problem and an analysis of its rate (i.e., how many sample are needed), which "asymptotically matches'' a known lower bound.  This "asymptotic optimality" means that as $\delta \rightarrow 0$, the number of samples needed converges to ths known lower bound.  For any fixed $\delta$ there is still a gap.

**Compliance With Llm Reviewing Policy:**

Affirmed.

**Final Justification:**

The rebuttal basically addressed my questions, though not in as clean a way as I might have hoped.  But my questions were quite minor, so the rebuttal essentially reinforced my prior assessment.  This is a nice paper that I think should be accepted.

**Key Questions For Authors:**

- What is the gap between your bound and the lower bound as a function of $\delta$, not just in the limit?
- How do your bounds compare to related work?

**Limitations:**

yes

**Strengths And Weaknesses:**

### Strengths:
- This is a very natural problem
- While the papers makes some assumption on what precise feedback we receive, their assumptions are still fairly general, and include most previous models and natural extensions.
- The result is quite strong -- asymptotic optimality is a nice result!
- While the techniques aren't necessarily groundbreaking (they seemed to me to essentially be clever modifications of standard ideas, and something like them would probably be most people's first attempt), there are a number of technical difficulties that push this above the bar (at least for me).

### Weaknesses:
- I think the authors could have done a better job comparing to previous work. For example, from the literature review section, it seems like Chen et al. (2018) and Ren et al. (2020) and (2018) give extremely closely related results to this paper.  There are differences (PAC vs $\delta$-correct, latent utility vs stochastic transitivity), but clearly there are close connections.  While these papers are mentioned, the bounds they provide are not.  Does this paper give better or worse bounds than those achieved by these related papers?
- Asymptotic optimality is great, but it is not clear what the gap is for fixed $\delta$.  The precise bound achieved (Proposition 6.3) is quite complex.  Theorem 6.5 show that this is asymptotically optimal, but the exact bound from Proposition 6.3 is difficult to parse for fixed $\delta$.  And the known lower bound is not even presented for fixed $\delta$.  So while the limit as $\delta$ goes to $0$ is good, it is difficult to figure out from this paper what the rate actually is.  A priori, it could be the case that $\delta$ has to be impossibly small in order to get anywhere close to the optimal bound.  It would have been good if the authors were able to bound the *rate* at which their bound converges to the optimal bound (as $\delta$ tends to $0$.)

---

> ### Author Rebuttal · Authors · 2026-03-31
>
> Thank you for the careful and constructive feedback. Together with the feedback from the other 3 reviewers, we have spent the last 6 days working to incorporate it and feel that it has substantially strengthened the paper. Since there is some overlap between your feedback and reviewers yJri and RdNJ, we will refer to some of the responses to their reviews to avoid repetition.
>
> **Q1 and Weakness 2:** Thank you for bringing this up. Because the focus was on the asymptotic result, we did not consider deriving it formally. However, based on the arguments in Proposition 6.3, Lemmas D.4–D.6, and Appendix E, we obtain the following result.
> Define
> $\rho=\min\Big\lbrace\alpha,1-\alpha,\gamma,\frac{1-\gamma}{2}\Big\rbrace,$
> for all sufficiently small $\delta$,
> $$\frac{ E_\theta[\tau_\delta]}{\log(1/\delta)} \le \frac{1}{\Gamma^*(\theta)}+ \widetilde O\left((\log(1/\delta))^{-\rho}\right).$$
> The statement is for $\delta<\delta_0(\theta)$, where $\delta_0(\theta)$ is the instance-dependent threshold at which the large time conditions used in Appendix E begin to hold. Taking $\alpha \in (\frac{1}{3},\frac{2}{3}),\gamma = \frac{1}{3}$ gives $\rho = \frac{1}{3}$. The choice we use in simulation (as discussed in the paper) gives $\rho = \frac{1}{5}$. A note about this and its proof (which is lengthy, so we leave it out of this response) has been added to the appendix.
>
> **Q2 and Weakness 1:** Here is an overview of the closely related work and the strengths of their guarantees relative to ours. We omit the additional discussion we will add about Ren et al. (2020), which we provide in the response to reviewer RdNJ (Weakness 1a)  and reviewer yJri (Q2).
>
> Space permitting, the following discussion will be added to the literature review.
>
> > Chen et al. (2018) study top-$k$ identification under the Multinomial Logit model with listwise preferences, with Bradley-Terry as the pairwise special case $(l=2)$. However, it is not formulated as a $\delta$-correct algorithm, in the sense that the user can’t choose $\delta$. The upper bound is proven “with high probability”, and the lower bound holds with probability at least $7/8$. The upper and lower bounds match up to polylog factors, so under the Bradley-Terry/Multinomial Logit model, they prove that their algorithm is near optimal with probability at least $7/8$.
>
> > Ren et al. (2018) study PAC top-$k$ identification under the multinomial logit model. Given $\epsilon$ and $\delta$, the goal is to return an $\epsilon$-top-$k$ set with error probability at most $\delta$. In the pairwise case, they prove the lower bound $\Omega\left(n\epsilon^{-2}\log(k/\delta)\right)$ and an upper bound of the same order, so their PAC result is order-optimal. But this PAC guarantee is not directly comparable to an exact $\delta$-correct guarantee: PAC allows items within an $\epsilon$-deviation around the $k$th boundary to be treated as interchangeable, whereas exact identification must recover the true top-$k$ set itself.
>
> After this we will insert the more detailed discussion of Heckel et al. (2019) and Ren et al. (2020).
>
> Our guarantees are asymptotic, however, in this regime we are not only order optimal but actually converge to the information theoretic lower bound (i.e. optimal including multiplicative constants). To the best of our knowledge, such a result has not previously been established for any top-$1$ / top-$k$ setting with pairwise comparison under latent utility models, and we prove it for a fairly broad class of parametric distributions.

---

> > ### Author Rebuttal · Reviewer_opv8 · 2026-04-01
> >
> > I still like this paper and think it should be accepted.  I wish there was a simpler interpretation for the question of the loss ofr a fixed $\delta$, but perhaps it is already as simple as possible.

---

### Official Review · Reviewer_RdNJ · 2026-03-12

**Soundness:** 4
**Presentation:** 3
**Significance:** 2
**Originality:** 3
**Overall Recommendation:** 5
**Confidence:** 4

**Summary:**

This paper considers the problem of identifying the top k of n items from pairwise comparisons with high probability. Collection of pairwise comparisons is adaptive; comparisons can be selected sequentially so as to try to minimize the number of samples to identify the top k items with high probability ($1 - \delta$). The main analysis is on the sample complexity of the problem.

The main contribution is an algorithm for allocating samples that is optimal in the limit as the failure probability $\delta \to 0$, in the sense that the expected number of pairwise comparisons collected matches a known information-theoretic lower bound.

**Compliance With Llm Reviewing Policy:**

Affirmed.

**Final Justification:**

Rebuttal and discussion reinforced prior positive assessment

**Key Questions For Authors:**

1. Do the other methods proposed in the literature have optimality guarantees, or are they known to be suboptimal? Could you compare these to your optimality result to theirs (if any) and which regimes seem most practically useful? I’m not sure I see why optimality as $\delta → 0$ is the most relevant regime.

2. From the setup, it seems like if $\theta_{(k)} - \theta_{(k+1)}$ can be arbitrarily small, inference should be harder and stopping time should be larger (since at any fixed sample size I could have picked a gap so that $\theta_{(k)} - \theta_{(k+1)}$ are hard to distinguish and the algorithm would pick the wrong one with probability about 1/2). How does this show up in the stopping time?

**Limitations:**

Yes, discusses the fact that the method performs worse at larger $\delta$.

**Strengths And Weaknesses:**

Strengths:

1. The problem definition, analysis and solution are each clean and satisfying.
2. The overall motivation for inference from pairwise preferences is clear: Ranking from pairwise preference is the foundation for many popular evaluation settings, like (Chatbot/LM) Arena.
3. Formalizing the problem as a two-player zero-sum game provides nice intuition: The algorithm works by choosing the comparison allocation distribution so as to maximize the (estimated) minimum KL distance between the (estimated) true ranking of models and some other ranking that doesn’t have the same $k$ top candidates.
4. The algorithm is asymptotically optimal as the failure probability goes to zero. This makes the theory very neat.

Weaknesses:

1. Choices for comparison and description of prior work:
    1. It would have been nice to outline the techniques used in the baseline methods, and why the intuition for the proposed method is better. Also, please clarify whether existing methods have optimality guarantees. I would revise my review downward if there were preexisting methods that are optimal in the same regime.
    2. Comparison to Chatbot Arena active inference: The most popular of these systems for AI is called (Chatbot/LM) Arena (see Chiang et al “Chatbot Arena: …” https://openreview.net/forum?id=3MW8GKNyzI). In their inference procedure, they allocate samples dynamically to try to minimize the confidence interval around $\max_{i,j} \hat p(y | (i,j))$. First, it seems like the Chatbot Arena strategy (or a minor modification of it) might work well in this setting, and to me it seems simpler. To optimize for top-k identification, one might try to allocate samples so that (adaptive sequences of) confidence intervals to try to exclude 1/2 (so that one model can be concluded to be better than another) until $k$ models are better than all others. It would have been nice to more substantively comment on the status quo approaches of these systems and why the proposed procedure is a better idea. I’m not familiar with the benchmark methods proposed, and the sample allocation scheme from Chatbot Arena seems like it is the nearest popularly implemented strategy (even if it isn’t optimized for top $k$ identification).
2. The motivation for the problem of top $k$ identification could have been made more clear. In what circumstance is this something that someone might want to do? Top-1 identification is easily justified by usage-decision/procurement choices, and intuitively in many cases it is more important to rank the top of the list correctly than it is to rank the bottom of the list. But I imagine the most natural way to encode the importance of ranking the top of the list correctly is via a measure that smooths over top $k$ identification for top $k=1, 2,…$. Fixed top-k identification seems arbitrary to me, and this inferential target wasn’t justified in the text.
3. Empirics: There should be some reporting on the correctness of each of the algorithms in the empirical demonstration. How often is the Algorithm 1 top-k correct? What about the other choices?  I could find this information anywhere in the body or appendix.

Minor comments:

1. The symbol $S$ is used both for the set of top $k$ models and for cumulative scores. I found this initially confusing.

---

> ### Author Rebuttal · Authors · 2026-03-31
>
> Thank you for the careful and constructive feedback. Together with the feedback from the other 3 reviewers, we have spent the last 6 days working to incorporate it and feel that it has substantially strengthened the paper. Since there is some overlap between your feedback and reviewers yJri and opv8, we will refer to some of the responses to their reviews to avoid repetition.
>
> **Weakness 1a:** We use two baselines (SEEKS and Active Ranking). We will describe the first here, and the second in our response “Weakness 1b”. To answer the question of whether there are preexisting methods that are optimal in the same regime, no (to the best of our knowledge). The closest existing baselines are at best within a $\log(n)$ factor of optimal. The strength of their guarantees is that they apply at finite $\delta$ (SEEKS at $\delta < .01$ and Active Ranking at $\delta \leq .14$). Ours is asymptotic, but we match the information-theoretic lower bound exactly (including multiplicative constants).
>
> The following description of Ren et al. (2020) will be added to the literature review.
>
> > SEEKS from Ren et al. (2020) uses an elimination scheme for top-$k$ selection. In each round, they first choose a pivot item that is “close” to the current $k$th item, compare the remaining items to that pivot, and then assign them into three groups: clearly above the pivot, ambiguous, and clearly below it. Items in the first group are “accepted” into the top-$k$, items in the third group are eliminated, and the algorithm repeats on the unresolved items in the second group. It stops once $k$ items have been accepted or $n-k$ items have been eliminated.
>
> Immediately after this, we will insert the paragraph in our response to reviewer yJri (Q2) regarding their guarantees.
>
> **Weakness 1b:** The active sampling rule in Chiang et al. (2024) is a greedy confidence interval reduction rule for improving estimation of the overall ranking. However, they do not have guarantees regarding (near) optimality of the algorithm in terms of sample complexity (or $\delta$-correctness). It serves as a sensible heuristic for learning the entire ranking.
>
> As the reviewer points out, this idea can be turned into a fixed-confidence top-$k$ procedure, and at a high level, is similar to the Active Ranking algorithm of Heckel et al. (2019). Active Ranking essentially counts wins, forms confidence intervals from the current data, and keeps sampling unresolved items until those intervals are sufficiently separated. It can be used for top-$k$ or learning full rankings, and its sample complexity is within $\log n$ factors of optimal.
>
> We agree that those methods are simpler and more interpretable than ours. This mainly arises since our algorithm aims to learn and track the oracle allocation online. That extra complexity is what gives us asymptotic optimality and is likely the reason we outperform the simpler baselines in a range of regimes.
>
> Space permitting we will add a brief discussion of this to the paper.
>
> **Weakness 2:** Thank you for pointing out the need for more motivation. We agree that top-$1$ has many natural applications, which our algorithm covers at $k=1$, but we believe the general top-$k$ setting is also prevalent. For example, in new product development, a company may have $50$ product candidates and use crowdsourced pairwise comparisons to narrow it down to $5$ products launched to the market. Due to character limitations, we can't go into this further. A more detailed motivation for top-$k$ identification has been added to the paper.
>
> **Weakness 3:** This is mentioned briefly at the very beginning of Section 7.1. All algorithms returned the correct top-$k$ set in each simulation.
>
> **Minor Comment:** Thank you for catching this double use of notation. We have changed the gradient score to $G$ and stopped using $G$ for $mD_{max}$.
>
> **Q1:** Most of this question is addressed in our response to Weakness 1a, 1b, reviewer yJri (Q2), and reviewer opv8 (Q2). As for the $\delta \to 0$ regime, we do not claim that it is the most relevant one. It’s a useful regime to study because the quantity $\Gamma^*(\theta)$ governs the asymptotic lower bound, identifying the allocation objective that an optimal procedure should maximize. The asymptotic optimality result guarantees that our allocation rule successfully learns and effectively optimizes this oracle quantity. This will be further discussed in the paper.
>
> **Q2:** Thank you for pointing this out. We did not think about formally demonstrating this until your suggestion: the stopping time does indeed grow as the gap $\Delta_k := \theta_{(k)} - \theta_{(k+1)} > 0$ gets small. In particular, we can show that $\Gamma^\star(\theta) \leq \frac{\sigma^2}{2}\Delta_k^2$ and thus for Algorithm 1, applying Theorem 6.5
> $$\lim_{\delta\to 0}\ \frac{E_\theta[\tau_\delta]}{\log(1/\delta)} = \frac{1}{\Gamma^*(\theta)} \geq  \frac{2}{\sigma^2 \Delta_k^2}.$$
> This will be a valuable addition to the final version of the paper.

---

> > ### Author Rebuttal · Reviewer_RdNJ · 2026-04-03
> >
> > Thanks to the authors for their clarifications. My questions about related work and motivation are resolved and my assessment remains positive.

---

### Official Review · Reviewer_bMRp · 2026-03-12

**Soundness:** 4
**Presentation:** 3
**Significance:** 3
**Originality:** 3
**Overall Recommendation:** 5
**Confidence:** 3

**Summary:**

This paper studies the problem of identifying the top-k items from pairwise comparisons in an online setting. Assuming the fixed-confidence setting under a parametric model, the authors first characterize the optimal sample complexity via an information-theoretic lower bound. They then reformulate the problem as a saddle-point optimization problem. Based on this formulation, the authors propose an online algorithm that adaptively selects which pairs to compare. They show that the algorithm is δ-correct and achieves asymptotically optimal sample complexity, matching the lower bound. Finally, they present numerical experiments under several data-generating models to illustrate the performance of the proposed method and compare it with existing approaches.

**Compliance With Llm Reviewing Policy:**

Affirmed.

**Final Justification:**

A strong paper across multiple dimensions that deserves acceptance.

**Key Questions For Authors:**

1. Could the authors provide more intuition for why this particular choice of threshold in Eq. 4 is the right choice? And how does it compare to stopping rules in related work?

2. The parameter space assumes \|\theta\|_\infty <= R. To what extent is this assumption necessary for the analysis? Would it be possible (or interesting) to extend the results to an unbounded parameter space?

3. The reduction to boundary pairs is elegant, do the authors expect this idea to extend to more general ranking settings, for example, when comparisons are partial over subsets of size greater than two?

**Limitations:**

Yes

**Strengths And Weaknesses:**

Strengths:
- The submission is technically solid, well structured, and well written. The mathematical exposition is thorough, the claims are stated clearly and supported by both theoretical analysis and experimental results.
- Top-k identification from pairwise comparisons is an important problem, and it is significant that this paper presents the first algorithm achieving asymptotic optimality in this setting.
- The main contribution comes from the way the existing ideas are combined and adapted to the top-k setting. In particular, the boundary-based reduction together with the saddle-point formulation seem to be the most novel aspects. While the individual components are not new, their integration is elegant and leads to a strong result. The resulting approach and techniques could be useful for related ranking problems.


Weaknesses:
Mostly presentational.

- Section 5 (Algorithm) is hard to follow. As a first-time reader, it is difficult to understand the pseudocode without going back and forth between Sections 5.1–5.5, since many auxiliary variables used in the pseudocode (such as Pij,Nij,Zij,It) are only defined later.
- One possible improvement would be to first present a simpler version of Algorithm 1 without these auxiliary variables, or at least include a high-level pseudocode broken into blocks with brief description (e.g., sampling, updates, stopping). This would make it easier to form a mental model of the algorithm and relate the pseudocode to the more detailed explanations in Sections 5.1–5.5.
- Relatedly, parts of this section feel quite dense. This is understandable given space constraints, but it might help to expand on some of the technical details in the appendix.
- There is very little intuition provided for the form of the stopping rule, which plays a key role in achieving optimal sample complexity.

---

> ### Author Rebuttal · Authors · 2026-03-31
>
> Thank you for the careful and constructive feedback. Together with the feedback from the other 3 reviewers, we have spent the last 6 days working to incorporate it and feel that it has substantially strengthened the paper.
>
> **Weaknesses:** Thank you for pointing out these presentational issues and the suggestion. While we did make an effort on the exposition of the paper, taking a fresh look at it with your feedback in mind makes it clear that for parts of section 5, our delivery fell short of our ambition. We have now implemented your suggestion regarding a less technical algorithm outline at the beginning of Section 5. This includes much less use of notation, which hopefully eliminates the need to go back-and-forth between the pseudocode and where the notation was formally defined.
>
> **Q1:** We agree that the stopping rule was introduced too abruptly. The following paragraph has been added to the beginning of Section 5.5.
>
> > The stopping threshold in Eq. (4) is obtained from a mixture likelihood-ratio martingale and self-normalized bound, in the same spirit as stopping thresholds used in linear bandits; see, e.g., Abbasi-Yadkori et al. (2011) and Lattimore & Szepesvári (2020). In linear bandits, one forms GLR statistics for each alternative pair of arms $(a,b)$, where arm $b$ is at least as good as arm $a$, and the stopping statistic is the minimum of these pairwise statistics. Here, for each estimated boundary pair $(i,j)$ (where $\theta_i \ge \theta_{(k)} > \theta_j$), we form the statistic $Z_{ij}(t)$ against its alternative (where we suppose $\theta_i \leq \theta_j$). Eq. (5) then stops when the minimum of these boundary pair statistics exceeds the threshold in Eq. (4).
>
> **Q2:** The boundedness assumption is primarily used as a tool in some proofs (also commonly used in linear bandits), but we do not view it as particularly restrictive in practice.  For example, under the Bradley-Terry model, if $R=5$, then $P(i \succ j)$ can be as high as $0.99995$. Thus, in most settings, it is likely that any practically relevant utilities lie within such a bounded range. Thank you for pointing out the lack of explanation for the assumption. The following discussion has been added to the appendix.
>
> > Here, we discuss where the boundedness assumption is used. First, since every pairwise gap $(\theta_i-\theta_j)$ lies in $[-2R,2R]$, the proof can use the curvature bound $$a=\inf_{|\eta|\le 2R} A’‘(\eta)>0,$$
> which is used for the MLE concentration argument. Second, bounded $\Theta$ makes $D_{\max}$ finite, which is the Lipschitz constant used to control how much the oracle value $\Gamma(w;\theta)$ can change with changes to the allocation $w$. Third, it makes $L$ finite, which appears in $$\Delta F_s:=\sup_{w,q}\bigl|F_{\hat\theta_s}(w,q)-F_\theta(w,q)\bigr| \le L\lVert\hat\theta_s-\theta\rVert_2,$$
> used to map estimation error to the error in the achieved information rate. Finally, in the stopping proof, boundedness gives us $$\Delta_t=\sup_{\theta’\in\Theta}\bigl|\bar L_t(\theta’)-L_t(\theta’)\bigr|$$
> finite and tending to zero, and it also gives the bound $$\frac{\lambda}{2}\lVert\hat\theta_t\rVert_2^2 \le \frac{\lambda}{2}nR^2,$$
> which allows us to conclude the stopping threshold grows only logarithmically. If $\Theta$ were unbounded, these bounds would not necessarily be finite constants, so the current proofs of estimation, oracle tracking, and finite stopping would no longer hold.
>
> > An extension to an unbounded parameter space would need a different type of argument: use localized bounds showing that the equations above stay inside a compact region, instead of global bounds over all of $\Theta$. This could be an issue under the Bradley-Terry model, because $A’'(\eta)\to 0$ as $|\eta|\to\infty$, so once $\Theta$ is unbounded, there is no global curvature lower bound to use.
>
> > Thus, the boundedness assumption is primarily used as a tool in some proofs (also commonly used in linear bandits), but we do not view it as particularly restrictive in practice. For example, under the Bradley-Terry model, if $R=5$, then $P(i \succ j)$ can be as high as $0.99995$. Thus in most settings, it is likely that any practically relevant utilities lie within such a bounded range.
>
> **Q3:** The idea of partial rankings as inputs is something that we have explored but so far only with partial results. In regards to the boundary reduction, say for example, that at each step we choose a set of $3$ items to rank (instead of a pairwise comparison). Then the boundary reduction still allows us to view this as a two-player game, and the adversary's strategy remains the same: choose the boundary pair that is weakest to "attack". However, the designer's problem is now harder: they need to choose which of the $\binom{n}{3}$ size-$3$ subsets would be most informative when ranked. For larger subsets, this continues to grow. Thus, viewing the problem from this lens may be helpful, but further work is required to simplify the designer's problem.

---

> > ### Author Rebuttal · Reviewer_bMRp · 2026-04-02
> >
> > The authors provided a detailed response to most of my concerns/questions.

---

### Official Review · Reviewer_yJri · 2026-03-13

**Soundness:** 4
**Presentation:** 3
**Significance:** 2
**Originality:** 2
**Overall Recommendation:** 5
**Confidence:** 4

**Summary:**

The paper studies a pure exploration with fixed confidence setting in dueling bandits, where the goal is to identify as quickly as possible the top k arms by ensuring that a specific confidence in the final decision is given. It is assumed that the feedback model for the pairwise comparisons is an exponential family subsuming popular models such as Bradley-Terry or the Thurstone model. Lower bounds in the spirit of the state-of-art form are derived following the work by Garivier&Kaufmann (ALT 2016). For designing a suitable learning algorithm the Track-and Stop algorithm of the latter work is adapted to the considered setting, where ideas of Degenne et al. (ICML 2020) and Jedra & Proutiere (NeurIPS 2020) are incorporated. This algorithm is investigated theoretically and empirically in experiments on synthetically generated datasets.

**Compliance With Llm Reviewing Policy:**

Affirmed.

**Final Justification:**

I want to thank the authors for the fruitfull discussion. It wasn't until I read the author's response, however, that I realized the feedback value (the $Y$ so to speak) can also be cardinal and isn't necessarily always binary. The specific focus on the literature in the field of Dueling Bandits had actually led me to this conclusion. Although Example 2 actually is an example for this, which, I wrongly set equal to the Thurstone model. In any case, this should be clarified in the paper.

All in all, the paper is a solid contribution and I will upgrade my rating to “Accept” to ensure a unanimous decision. However, I would like to see the promised discussions in the final paper.

**Key Questions For Authors:**

I really appreciate that the authors are also showing a scenario where the algorithm might be not dominating all baselines. Thus, it would be also interesting to see what is meant by the sentence "the early gradient directions can be noisy" which indicates that the choice of $\alpha$ is quite critical, I suppose?

The closest work is by Ren et al. (ICML 2020) where also a lower bound is derived. How does the lower bound shown in the current paper compare to theirs? This is worth a short discussion in the paper.

**Limitations:**

The assumption of an exponential family for the pairwise comparisons might be quite restrictive as it poses quite strong structural assumptions on the arms

**Strengths And Weaknesses:**

## Strengths

- The paper is very well-written and polished.
- The authors also provide good intuitions on certain steps in their construction of the learning algorithm and use a clear notation
- The algorithm is shown to be asymptotically optimal, which is a desired property in this setting
- It is shown in experiments that the algorithm has numerical advantages in comparison to existing baselines and more importantly, that there are also scenarios where it might not be dominating entirely

## Weaknesses

- The overall novelty or originality of the paper is fair, as it follows ideas in the existing literature for the setting considered
- Related to the previous point: The findings of the paper are not surprising and the overall contribution was rather to rigorously formulate and derive the relevant quantities such as the gradient estimates or information rate, etc. as well as investigate the properties of the MLE in this setting
- The assumption of an exponential family for the pairwise comparisons might be quite restrictive as it poses quite strong structural assumptions on the arms
- Related to the previous point: It is true that the SST setting is a misspecification scenario for the derived algorithm. However, it is not too far off. It would be more interesting to have a more drastic violation of the assumption.

---

> ### Author Rebuttal · Authors · 2026-03-31
>
> Thank you for the careful and constructive feedback. Together with the feedback from the other 3 reviewers, we have spent the last 6 days working to incorporate it and feel that it has substantially strengthened the paper.
>
> **Q1:**
> With a fresh look at the paper with your feedback in mind, we realize that the phrase “the early gradient directions can be noisy” is too vague. We have now edited the appendix to add the below explanation. In addition, we will provide simulations showing how the choice of $\alpha$ affects the algorithm’s performance, one of these simulations is provided in the table below.
>
> > There are two mechanisms contributing to this. The first is the stochastic gradient approximation itself. The full $w$-gradient averages over all boundary pairs in $B(\theta)$, whereas our update uses only the sampled pair $I_t$ and the corresponding alternative $\theta_t^*$. Thus, early on depending on which pair is sampled, the update can put too much weight on comparisons that are especially informative for ruling out that one sampled inversion, rather than on those that matter most for the full gradient. For example, if $n=4$ and $k=2$, the most challenging boundary pair is to differentiate ranks $2$ and $3$, but we may draw ranks $(1,4)$ several times early on. The resulting updates overweight comparisons useful for ruling out item $4$ overtaking item $1$, which causes the cumulative scores $S_t^{(w)}$ and $S_t^{(q)}$ to be biased early. Hence, if $\mu_t=t^{-\alpha}$ decays too quickly, later updates will take a long time to undo this bias; if it decays too slowly, then the iterates remain overly sensitive to one-step noise. This is only an issue in early rounds, since later, non-bottleneck boundary pairs receive very little mass under $q_t$, so even if such a pair is sampled, its effect on the $w$-update is small. This is the mechanism that leads to improved performance of the oracle problem ($\theta$ known) when $\alpha$ is moderately small.
>
> > When $\theta$ is not known, there is a second mechanism that arises since the gradients are evaluated at $\hat\theta_t$ rather than at $\theta$. Even if we were to compute the full gradient, it could still be misaligned early, because the estimated boundary set and the associated $\theta_{ij}^*$ may be off. When tuning $\alpha$, we therefore consider both its effect on the oracle problem and on the full problem with estimation error. Empirically, these choices are quite similar. The slower decay that helps mitigate the issue of using a stochastic gradient also gives the MLE and the estimated boundary set time to stabilize.
>
> The table below shows the average stopping time over 100 simulations of the oracle and Algorithm 1, where $n = 50, k = 5, \delta = .01$ as $\alpha$ varies. As the reviewer conjectured, the choice of $\alpha$ is important; too large or too small can result in significantly worse performance. However, as illustrated in the table, for $\alpha$ in the range $[.15,.3]$ the algorithm tends to do well, and this holds robustly across different configurations. So it is not brittle to the exact choice of $\alpha$. We will include plots of this over various combinations of $n,k$ in the paper’s appendix.
>
> | $\alpha$ | 0.1 | 0.15 | 0.2 | 0.25 | 0.3 | 0.4 | 0.5 |
> |---|---|---|---|---|---|---|---|
> | Adaptive | 530K | 76K | 71K | 71K | 75K | 141K | 445K |
> | Oracle (known $\theta$) | 606K | 70K | 67K | 68K | 74K | 139K | 442K |
>
> **Q2:** Thank you for pointing this out. The following discussion will be added to the literature review when discussing Ren et al. (2020).
>
> > Their paper is formulated under SST and STI, but for $\delta$-correct top-$k$ selection, the lower bound is proven under the following assumptions:
> (i) comparisons are generated from a Thurstone model i.e. $p_{ij} = \mathbb P(\theta_i + Z_1 > \theta_j + Z_2)$ with $Z_1$ and $Z_2$ independent Gaussian noise with variance $1$, (ii) $\delta \in (0,1/100)$, (iii) the true utilities $\theta_1,\ldots \theta_n \in [0,1].$
> Under these conditions, they obtain the instance-dependent lower bound on the expected number of samples
> $\Omega\left(\sum_{i=1}^n \Delta_i^{-2}\log(1/\delta) + \log\log \Delta_{r_k}^{-1}\right),$
> where $\Delta_i$ is a measure of distance of item $i$ to the $k$-th (or $k+1$-th) best item. They show their algorithm (SEEKS) has a sample complexity of
> $O\left(\sum_{i=1}^n \Delta_i^{-2}\left(\log(n/\delta) + \log\log \Delta_i^{-1}\right)\right).$
> So under their Thurstone model, there remains an extra $\log n$ factor between the lower and upper bounds, and if comparisons are not according to a Thurstone model, it can potentially be worse.
> By contrast, our guarantee is asymptotic, but we match the lower bound exactly (including multiplicative constants).

---

> > ### Author Rebuttal · Reviewer_yJri · 2026-04-03
> >
> > Thanks for the clarifications. I think the final paper will benefit from such an insightful discussion.
> >
> > I only have to remark that the model considered in your paper automatically fulfills the SST and STI condition, as you are making a stronger assumption with the exponential family and the utility scores.Also in your setting with Gaussian differences you are also in the Thurstone case.The answer reads as if SST,STI and the Thurstone model are something completely unrelated.I think the key difference is rather that they consider a PAC setting.

---

> > > ### Author Response · Authors · 2026-04-05
> > >
> > > Thank you for carefully considering our response and the helpful remark. When observations are binary, the latent utility model automatically satisfies SST, and we will make this clear in the paper. STI is not automatically satisfied, but would hold for many distributions (including Bradley-Terry).
> > >
> > > In regard to the lower bound, we realize our explanation of the distinctions between settings was not fully clear. The Gaussian differences model (where outcomes are cardinal) satisfies our assumptions, but the Thurstone model does not. Under the Thurstone model, outcomes are binary (i.e., the differences are mapped through the probit link), and the model is no longer an exponential family. Thus, the regime in which they have a lower bound to compare their upper bound against is complementary to our work. Space permitting, this will be clarified in the paper.

---

### Decision · Program_Chairs · 2026-04-30

**Decision:**

Accept (regular)

**Comment:**

The paper studies a natural and important top-$k$ identification problem from pairwise comparisons. The reviews were uniformly positive after rebuttal. The main technical contribution is a clean asymptotic-optimality result under a latent-utility/exponential-family model, together with a principled saddle-point formulation and an adaptive allocation strategy that tracks the optimal comparison distribution online. The main concerns raised initially were about presentation, the relation to prior work, the role of the bounded-parameter assumption, and the gap to the lower bound at finite $\delta$. The authors' rebuttal successfully addressed these points.